# Phosphorus transport in a hotter and drier climate: in-channel release of legacy phosphorus during summer low flow conditions

Christine L. Dolph[1], Jacques C, Finlay[1], Brent Dalzell[2,3], Gary W. Feyereisen[2,3,4]

[1]Department of Ecology, Evolution and Behavior, University of Minnesota, 140 Gortner, 1479 Gortner Ave, St. Paul, Minnesota, 55108
[2]USDA-ARS Soil and Water Management Research Unit, St. Paul, Minnesota, 55108
[3]Department of Soil, Water, and Climate, University of Minnesota, 438 Borlaug Hall, 1991 Upper Buford Circle, St. Paul, Minnesota, 55108
[4]Department of Bioproducts and Biosystems Engineering, University of Minnesota, 1390 Eckles Ave., St. Paul, Minnesota, 55108

*Correspondence to*: Christine L. Dolph (dolph008@umn.edu)

**Abstract.** "Legacy phosphorus" is the historical accumulation of phosphorus (P) in soils and sediments due to past human inputs. River networks represent a potential sink and/or source of legacy P, with many in-channel processes potentially governing the storage and mobilization of P over time. The objective of this study was to evaluate the potential contribution of in-channel release of legacy P to bioavailable P transport in streams during summer low flow conditions across a land use gradient in Minnesota, USA. We addressed this objective through synthesis of: 1) water quality and stream flow (Q) data collected for 143 gaged watersheds across the state of Minnesota between 2007-2021 (22,750 total samples); 2) water quality data from 33 additional ditch, stream and river sites in Minnesota sampled under low flow conditions in summer of 2014; and 3) water quality data collected from tile drainage outlets for 10 monitored farm fields between 2011-2021. We used geospatial data and a random forest modeling approach to identify possible drivers of bioavailable P concentrations during summer low flows for gaged watersheds. Between one third to one half of the gaged watersheds we studied exhibited SRP concentrations during low flows in late summer above previously identified thresholds for eutrophication of 0.02 - 0.04 mg/L. For many of these watersheds, stream SRP concentrations in late summer were above those observed in tile drainage outlets. Elevated SRP concentrations during late summer low flows weakened concentration-discharge relationships that would otherwise appear to indicate more strongly mobilizing SRP-Q responses across other seasons and flow conditions. While wastewater discharge likely contributed to elevated P concentrations for watersheds with high densities of treatment plants, many watersheds did not have substantial wastewater impacts. The most important variables for predicting bioavailable P concentrations during late summer low flow conditions in a random forest model were land use in riparian areas (particularly crop cover), soil characteristics including soil erodibility, soil permeability, and soil clay content, agricultural intensity (reflected via higher pesticide use, higher phosphorus uptake by crops, and higher fertilizer application rates), watershed precipitation and stream temperature. These findings suggest that, for stream and river sites heavily impacted by past and current P inputs associated with agriculture and urbanization, biogeochemical processes mediated by climate and geology can result in the release of legacy P from in-channel stores during late summer low flow conditions. As summers become hotter and, at times, drier -- predicted changes in this region -- conditions for the release of legacy P stored in stream and river channels will likely become more prolonged and/or more acute, increasing eutrophication risk.

40 **Short Summary**

"Legacy Phosphorus" is the accumulation of phosphorus (P) in soils and sediments due to past inputs from fertilizer, manure, urban runoff, and wastewater. The release of this P from where it is stored in the landscape can cause poor water quality. Here, we examined whether legacy P is being released from stream and river channels in summer across a large number of watersheds, and we examined what factors (such as climate, land use and soil types) might be

driving that release.

## 1 Introduction

Phosphorus (P) inputs arising from urbanization and industrial/intensive agriculture have resulted in widespread eutrophication of freshwater and marine environments. Excessive inputs of P along with nitrogen (N) have resulted in costly and sometimes dangerous conditions for human society, including increased prevalence of harmful algal blooms, contamination of drinking water supplies, decreased recreational opportunities, loss of critical marine fisheries, and negative impacts to biodiversity (Bennett et al., 2001). This problem is particularly acute in the Midwestern Cornbelt of the United States, which represents a global hotspot for P fertilization (Haque, 2021).

Most progress in reduction of P release to the environment has come from the implementation of improved wastewater infrastructure (Keiser and Shapiro, 2019). However diffuse (nonpoint) sources of P such as those arising from agricultural and urban landscapes have yet to be substantially curtailed and remain largely unregulated. In addition to ongoing P inputs to the environment, water quality is also impacted by the existing supply of "legacy P" in the landscape (Goyette et al., 2018). Legacy P is the historical accumulation of P in soils and sediments due to past land use practices, such as agricultural fertilization, the spreading of manure, and wastewater discharge.

Efforts are underway to understand sources of legacy P in the terrestrial environment including agricultural soils and riparian buffers (e.g., Osterholz et al., 2020). Lentic water bodies (lakes, impoundments and wetlands) are well known for their potential to remobilize stored P and become sources instead of sinks for downstream P, especially at high rates of nutrient inputs (e.g., Vilmin et al., 2022). The river network itself represents another potential sink and/or source of legacy P; with many dynamic in-channel processes potentially governing the storage and mobilization of P over time. For example, benthic redox conditions, in-stream primary productivity, microbial respiration, and sediment adsorption-desorption can all modulate whether P is retained in stream sediments, temporarily immobilized as organic P, or released to the water column as bioavailable P (Records et al., 2016).

We previously observed that concentrations of bioavailable P (i.e., soluble reactive phosphorus, SRP) in agriculturally-dominated streams and rivers of Minnesota were often elevated during low flow conditions in late summer (Dolph et al., 2019). However, questions remained about whether the elevated SRP we observed in late summer was sourced predominantly from tile drainage (i.e., and therefore indicative of legacy and/or current P sources stored in farm soils), from point sources such as wastewater treatment plants, or possibly from legacy sources in the river network itself. Tile drainage is extensive across the agricultural Midwest (Valayamkunnath et al., 2020) and has been found to contribute substantially to and even dominate soluble P export in agricultural watersheds (King et al., 2015; Smith et al., 2015). P concentrations in tile waters have been found to be highest during summer compared to other seasons (King et al., 2015), and therefore represent a possible driver of elevated SRP in streams and rivers receiving tile drainage at this time of year. Comparatively high SRP concentrations during low flows can also be indicative of the dominance of point discharges; these concentrations are often diluted under wetter conditions (Dupas et al., 2023). Alternatively, however, there is some indication that groundwater and/or in-channel processes may drive the release of bioavailable P in river channels at some times of year, such as summer (Schilling et al., 2020; Vissers et al., 2023).

A number of recent papers have examined potential legacy P dynamics in streams and rivers; these studies have typically been deployed at the reach scale (i.e., stream reaches of a few hundred meters or less), or for individual small to medium-sized watersheds (e.g., Bieroza and Heathwaite, 2015; Casquin et al., 2020; Kreling et al., 2023; Siebers et al., 2023; Vissers et al., 2023; Dupas et al., 2023; Rode et al., 2023). These in-depth studies are important and highly useful, as the microscale dynamics governing P mobility in river channels can be complex. However, few studies have examined the potential contribution of in-channel legacy P at larger regional scales, or across a large number of watersheds.

The objective of this analysis was to determine the potential contribution of in-channel legacy P sources to SRP transport under summer low flow conditions across a relatively broad spatial scale (i.e., the state of Minnesota). We hypothesized that in-stream processes contribute to elevated concentrations of bioavailable P during summer in streams with strong agricultural and/or urban influence, in addition to the contribution of tile drainage systems and point source discharges. We addressed this hypothesis through synthesis of three water quality datasets: 1) water quality and stream flow data collected for 143 gaged watersheds across the state of Minnesota between 2007-2021 (22,750 total samples); 2) water quality data from 33 additional ditch, stream and river sites in Minnesota sampled under low flow condition in summer of 2014; and 3) water quality data collected from tile drainage outlets for 10 monitored farm fields between 2011-2021. We also used geospatial data and a machine learning approach to identify possible drivers of elevated SRP concentrations during summer low flows for gaged streams and rivers.

Watersheds across the state of Minnesota span a land use gradient from those dominated by intensive agriculture typical of the Upper Midwest region, to a major metropolitan area, to areas of heavy forest and wetland cover with comparatively fewer historic P inputs. This gradient provides a useful contrast that can potentially be applied to identify differential behavior of streams and rivers strongly impacted by legacy P.

## 2 Methods

### 2.1 Study area

The study area for this research spans the entire state of Minnesota, USA, encompassing approximately 225,163 km$^2$ within the Upper Midwestern region of the United States (Fig. 1). The state includes parts of four major drainage basins: the Upper Mississippi River Basin in the central, south and southeastern portions of the state, the Red River Basin in the northwest, the Great Lakes Basin in the northeast and the Upper Missouri River in the far southwest corner. Gradients in land use, soils, and precipitation vary from north to south and east to west (Fig. 1). The majority of the southern and north-western parts of the state are dominated by industrial row crop agriculture, predominantly corn and soybeans, with a high density of concentrated animal feeding operations (CAFOs) particularly in the south. By contrast, the north and northeastern parts of the state are dominated by forest and wetland cover. The state is also home to a major metropolitan area, encompassing the Twin Cities of Minneapolis and St. Paul and the surrounding seven counties (population of 3.69 million, 2020 US census) characterized by urban and suburban landscapes. Precipitation varies from driest in the northwest (annual average rainfall ~550 mm, 1991-2020) to wettest in the southeast (annual average rainfall ~950 mm, 1991-2020; Johnson et al., 2022). Mean annual temperatures are higher in the south (annual average temperature ~ 7°C, 1991-2020) and lower in the north (annual average temperatures ~ 2-

3°C, 1991-2020). The entire state is characterized by a cold climate, with average winter temperatures well below freezing and with considerable snowfall historically expected most years. Most soils are formed from glacial and peri-glacial deposits. Soil textures range from sandy soils in the central part of the state, clay loam and silty clay loam soils in the south-central and southwest, and outwash till over karst bedrock in the southeast. Many of the soils in the western part of the state are calcareous with high pH. Water quality in the state is characterized by widespread impairments in the agriculturally and urban dominated regions, with the most ubiquitous impairments attributed to turbidity, total phosphorus, fecal coliform, impaired biota, and low dissolved oxygen (MPCA, 2022). Water quality in the northeastern part of the state is comparatively good, with lower levels of nutrient enrichment, although impairments for mercury contamination arising from coal burning and subsequent atmospheric deposition are widespread.

**2.2 Overview of study data**

For this study, we utilized three independent datasets (Fig. 1):

1) SRP concentration and discharge data (total n=22,750 flow-matched water chemistry samples) collected for 143 gaged stream and river watersheds monitored by the state of Minnesota's Watershed Pollutant Load Monitoring Network[1] between 2007-2021. These data were used to evaluate SRP transport behavior and understand drivers of late summer SRP.

2) SRP concentration and discharge data available for 10 tiled farm fields across the state, collected between 2011-2021 by the Discovery Farms Minnesota program[2]. These data were used to estimate seasonal SRP concentrations for tile outlets as a point of comparison with riverine SRP concentrations.

3) SRP grab samples collected during low flow conditions in late summer of 2014 for an additional set of ditch, stream and river sites. (n=33; Dolph et al., 2017a[3]). These data were used to provide additional information about SRP concentrations in smaller order systems that were underrepresented among gaged watersheds.

All data analysis was performed in R (R Core Team, 2023). Study data and R scripts used for data analysis are available at https://doi.org/10.5281/zenodo.13936951.

---

[1] https://public.tableau.com/app/profile/mpca.data.services/viz/WatershedPollutantLoadMonitoringNetworkWPLMNDataViewer/ProgramOverview

[2] https://discoveryfarmsmn.org/

[3] https://conservancy.umn.edu/handle/11299/189907

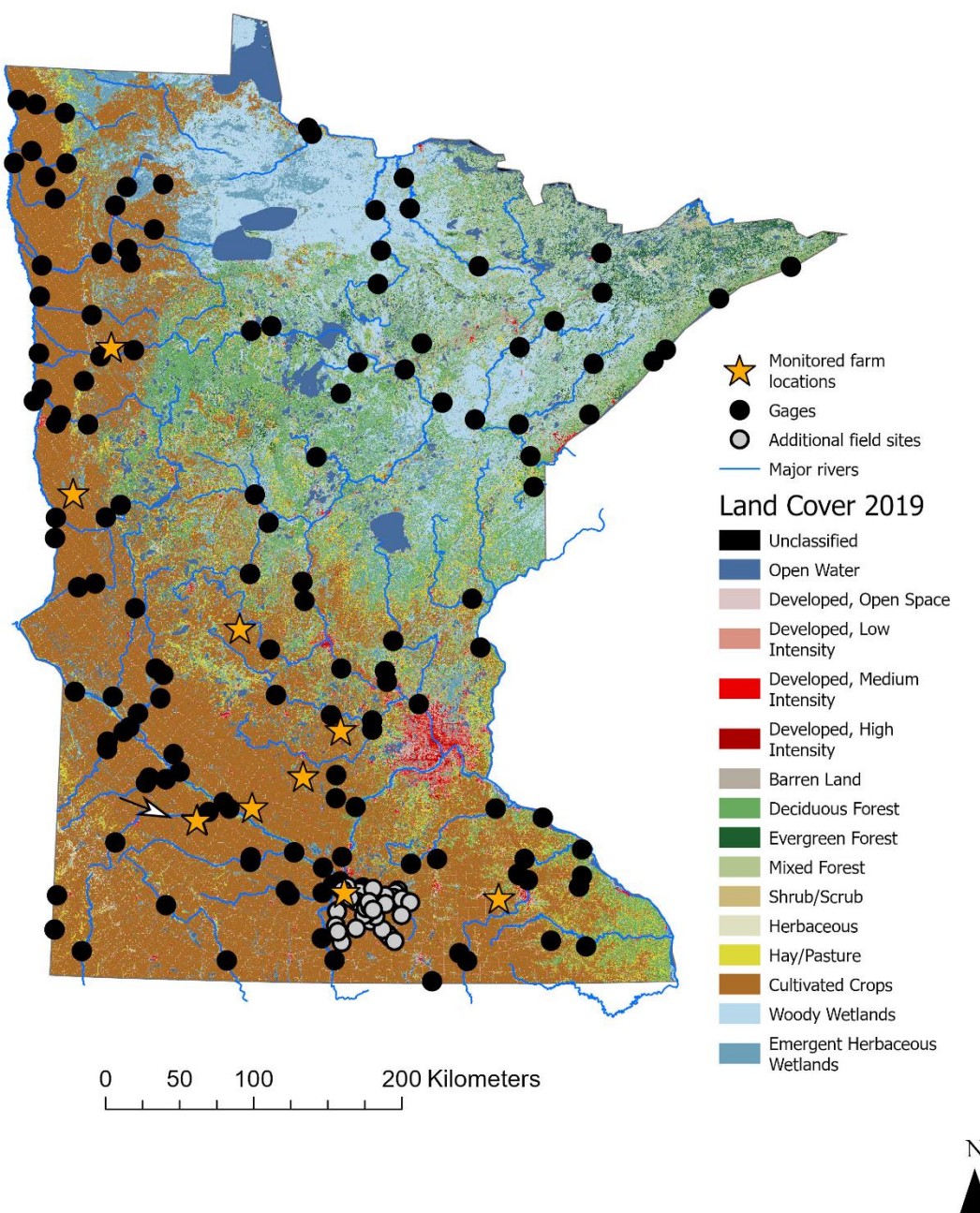

**Figure 1: Locations of 1) 143 gaged stream and river watersheds intensively sampled for SRP and flow (total n=22,750 samples) at the watershed outlet during 2007-2021 (black dots; n=143). Data from these sites were used to evaluate SRP transport behavior and understand drivers of late summer SRP; 2) Farm fields with tile outlet water quality available (collected between 2011-2021; orange stars; n=10) used to estimate seasonal SRP concentrations for tile outlets, as a point of comparison with riverine SRP concentrations; 3) ditch, stream and river sites sampled during summer low flow conditions in 2014 (gray dots; n=33). Data from these sites were used to quantify late summer SRP concentrations in smaller order systems. Note that two farm fields (known as RW1N and RW1S) are located in close proximity, and so appear as one location on the map; this location is noted with a white arrow.**

**2.3 Water quality data from stream and river gages**

We used paired SRP and daily discharge data from 143 gaged stream and river watersheds (Fig. 1) monitored by Minnesota's Watershed Pollutant Load Monitoring Network (WPLMN; note that the WPLMN also refers to SRP as "dissolved orthophosphate"). The total number of samples across all gaged watersheds was 22,750. Periodic water samples and continuous flow data were collected by the Minnesota Pollution Control Agency (MPCA) throughout the year at major watershed sites (watershed areas greater than ~4000 km$^2$ ) and during the period of ice-out through 31 October at smaller subwatershed sites (MPCA, 2015). Water quality sampling efforts were conducted ~biweekly with more intensive sampling focused on snowmelt and storm events, resulting in observations distributed across the range of flows observed at each site (average # of samples per year = 25 for subwatersheds and 35 for major watersheds; MPCA, 2015). The 143 gaged sites we selected for this study had >20 water chemistry samples collected across the sampling period (2007–2021). Median number of water quality samples per site across the whole time period was 120 (min=21, max=478). To determine detection limits, we inspected the data for repeated minimum concentrations and assigned detection limits equal to those minimum concentrations.

Watershed areas for gaged stream and river sites were assembled from multiple sources including existing watershed delineations (n=11 watersheds) available from USGS (2012) and  previously delineated watersheds (n=65 watersheds) from Dolph et al., (2019), or delineated anew as part of the current study (n=68 watersheds). For newly delineated watersheds, we used gage locations provided by Minnesota Department of Natural Resources and Minnesota Pollution Control Agency (2023) as pourpoints, and existing flow direction and flow accumulation rasters available from the NHDv2Plus dataset (USEPA, 2019) to delineate watersheds using the Spatial Analyst toolbox in ArcGIS Pro (ESRI, 2022). Watersheds were inspected visually and manually corrected for inaccuracies in delineation. Across all gaged sites, watershed area ranged from 20 km$^2$ – 29,145 km$^2$ (mean = 1,996 km$^2$)

**2.4 Farm tile outlets**

We used SRP concentration and discharge data from tile outlets draining 10 farm fields across the state, measured between 2011-2021 (Fig. 1). These tile outlets are monitored by the Discovery Farms Minnesota (DFM) program[4]. The DFM is a farmer-led water quality research and educational program with the goal of collecting water quality information under real-world conditions to support the development of better farm management decisions. During the time of data collection, all monitored farm fields were planted in corn and soybean row crops grown in rotation. Two sites (WR1 and ST1) included dairy operations, and two sites (BE1 and DO1) included swine finishing, in addition to row crops. Swine finishing is the final stage of pig farming where young pigs are fed until they reach market weight.

The drainage areas for monitored farm fields ranged from 10-160 acres (mean = 97 acres). Farm field soil textures ranged from poorly drained silty clay loam, to well drained loam. Three of the farms (MC1, RE1, and WR1) each had one surface inlet to the tile drainage system. All other inlets were subsurface. One farm (NO1W), had monitoring data

---

[4] https://discoveryfarmsmn.org/

available for two separate fields (NO1W-N and NO1W-S). Water quality and flow data collection is described in detail by MDA (2021). Briefly, tile outlets were monitored continuously for flow (15 min interval) via area velocity sensors installed in the tile drains that measured both stage and velocity. Water quality samples were collected by ISCO 6712 automatic samplers on an equal-flow increment (EFI) composite basis, whenever tile outlets were flowing. Water quality samples were composited every 125mL. Following a runoff event, water quality samples were collected and promptly transported to a state contract lab and measured for dissolved orthophosphorus (i.e., SRP) along with other water quality constituents. From continuous flow and composited sample SRP concentrations, we calculated a daily flow-weighted SRP concentration (daily C) as follows: 1) multiply composite concentrations by paired continuous flow measures to estimate continuous (15 min) loads; 2) sum composite sample loads into daily loads; 3) divide daily load by summed daily flow to compute a daily flow-weighted concentration in mg/L. Seasonal SRP concentrations were calculated by taking the mean of daily SRP concentrations for each tile outlet during each season (Early winter: Nov-Dec; Late winter: Jan-Mar; Spring: Apr-May; Early summer: Jun-Jul; Late summer: Aug-Sept; Fall: Oct).

### 2.5 Additional field sites

Among gaged stream and river watersheds, small order systems (especially first through third order ditches and streams) are under-represented relative to their prevalence across the landscape. To get a better understanding of SRP concentrations in smaller order systems, we also examined late summer low flow SRP concentrations collected from 33 agriculturally-dominated ditches, streams and mid-sized rivers in the Le Sueur River Basin, Minnesota (Fig. 1). Data for these sites is part of a larger publicly available field dataset[5] for the region and described in detail by Dolph et al. (2019). Briefly, SRP concentrations were determined for grab water samples collected from 33 sites during low flow conditions in August of 2014. Flow conditions at the time of sampling were characterized by flow at the gaged outlet of the major HUC8-scale watershed in which samples were collected (i.e., the Le Sueur River Basin), based on daily discharge data available from MNDR[6]. Although flow at watershed outlets is not precisely representative of flow conditions further upstream in the basin, we have shown previously that discharge conditions across study sites scaled reasonably well with drainage area (Dolph et al., 2017b). We sampled on August 14, 17, 20, and 26 of 2014, during which flow conditions at the watershed outlet ranged between 19-25th percentile of all daily flows available for this watershed. Sites were categorized as ditches, perennial streams and rivers, or intermittent streams and rivers according to their designation in the NHDPlusv2 (USEPA, 2019).

### 2.6 Low flow conditions

Part of our aim in this study was to identify whether in-channel dynamics, such as instream release of legacy P, may affect stream and river SRP concentrations and transport behavior. Thus, we sought to identify low flow conditions where we assumed in-channel processes were likely to dominate P dynamics. We identified 'low flow' conditions as those falling within the lowest 25% of all daily discharge conditions measured for each watershed during the period

---

[5] https://doi.org/10.13020/D6FH44
[6] https://www.dnr.state.mn.us/waters/csg/index.html

of record for that gage. We defined seasons as follows: Early winter (Nov-Dec); Late winter (Jan-Mar); Spring (Apr-May); Early summer (Jun-Jul); Late summer: (Aug-Sept); Fall (Oct). We defined these seasons in approximate relation to the agricultural growing seasons in our study region, with spring corresponding to when crops are planted, summer corresponding to when crops are growing rapidly, fall corresponding to when dominant crops (corn, soybeans) are harvested, and winter corresponding to when crops are dormant. We divided winter into early winter when snow is accumulating and generally not melting, and late winter, which is associated with snowmelt. We divided summer into early summer when conditions are generally wetter and crops are experiencing rapid growth, and late summer when climate conditions are generally drier and warm season crops mature rapidly.

We calculated mean SRP during low flow conditions for each gaged watershed in each season, for gages that had a minimum of three SRP samples collected during low flow conditions in that season. Note that not all gaged watersheds had three or more SRP samples collected during low flows in each season (Table A4); thus, the number of gaged watersheds with mean low flow SRP values available for analysis was different during each season (this parallels the availability of low flow conditions across seasons, with low flows being most common during late summer compared to other seasons).

We hypothesized that low flow SRP concentrations could be substantially affected by one or all of the following: tile outlet concentrations, wastewater treatment plant discharges, or riverine legacy P stores. To help discern these influences, we compared low flow riverine SRP concentrations to tile outlet concentrations. In addition, we evaluated low flow riverine SRP concentrations for gaged watersheds relative to wastewater treatment plant density (sites/km$^2$) in the watershed. Wastewater treatment plant density estimates were obtained from the US EPA StreamCat dataset (Hill et al., 2016; see additional details about StreamCat below), and were based on wastewater treatment plants listed in EPA's Facility Registry Services and National Pollutant Discharge Elimination System (NPDES)[7]. We also evaluated low flow riverine SRP concentrations relative to % cropland land use in gaged watersheds, to examine the assumption that agricultural land use and the associated past and current P inputs might drive the supply of riverine P. Cropland land use estimates were also obtained from StreamCat and were based on the 2019 National Land Cover Database (Dewitz, 2021).

**2.7 Influence of late summer low flows on concentration-discharge relationships**

We evaluated the relationship between SRP concentration (C) and discharge (Q) using the power law relationship in Eq. 1:

$C=aQ^b$ (1)

where the curve's coefficient (*a*) and exponent (*b*) are representative of the degree, direction, and rate at which SRP is transported as a function of stream flow. This equation can alternatively be expressed in log-log scale as Eq. 2:

$\log(C) = b \log(Q) + \log(a)$ (2)

---

[7] https://catalog.data.gov/dataset/epa-facility-registry-service-frs-wastewater-treatment-plants

where $b$ is the slope of the linear log-log relation, and $\log(a)$ is the y-intercept. Normalizing Q by the geometric mean of discharge ($Q_{GM}$) shifts the center of mass of the log-transformed Q data to the y-intercept, allowing for comparison of rating curves among different watersheds (Warrick et al., 2015). We performed linear regression of log-transformed SRP concentrations on log-transformed normalized discharge using Eq. 3:

$$\log(C) = b \log(Q/Q_{GM}) + \log(a) \tag{3}$$

All regressions were performed in R (R Core Team, 2023). We evaluated the fit of the power law relationship for all gaged watersheds using the significance value $p$, slope $b$ and $R^2$ of the linear regression.

The slope $b$ of this relationship describes the per unit increase in concentration as discharge increases. Concentrating relationships ($b > 0$) imply higher flows are mobilizing more of a waterborne constituent, particularly through erosion
or greater landscape connectivity. Diluting relationships ($b < 0$) suggest that constituents are source-limited or that relatively consistent inputs are diluted by greater discharge (Godsey et al., 2009). When $b$ is near 0, $C$-$Q$ relationships may be either chemostatic (i.e., relatively constant concentrations across the range of discharge conditions), or chemodynamic (i.e., concentrations are highly variable across the range of discharge conditions but not linearly related to flow). Chemostatic behavior has been observed for mineral weathering products or for constituents with large legacy
sources like nitrate (Godsey et al., 2009; Basu et al., 2010; Musolff et al., 2015), whereas chemodynamic behavior may indicate that biogeochemical processes such as sorption/desorption, biotransformation or oxidation/reduction strongly affect nutrient transport behavior (e.g., Wanner et al., 1989). To distinguish between these two behaviors, we evaluated the coefficient of variation of $C$ relative to the coefficient of variation of $Q$ ($CV_C/CV_Q$). A $CV_C/CV_Q << 1$ suggests that concentrations are relatively constant compared to variability in flow, indicating chemostatic behavior.
By contrast, a larger $CV_C/CV_Q$ indicates chemodynamic behavior (i.e., comparatively large variations in concentration relative to variation in flow). Thompson et al. (2011) suggested that $CV_C/CV_Q$ values ≈0.3 could be used as a threshold to identify chemostatic vs chemodynamic behavior.

To determine the influence of low flow conditions in late summer on the nature of the $C$-$Q$ relationships for all watersheds, we refit power law relationships to all watersheds after excluding SRP samples that were collected during
late summer low flow conditions. We compared regression parameters ($p$, slope $b$ and $R^2$) before and after withholding samples collected during late summer low flow conditions, to determine if these samples had a widespread effect on $C$-$Q$ relationships for SRP across gaged watersheds.

### 2.8 Regression analysis

### 2.8.1 Random Forest models

We used random forest modeling to identify possible predictors of SRP during low flow conditions in late summer for gaged stream and river watersheds. Random forest regression is a nonparametric ensemble learning method that utilizes predictions from multiple decision trees to improve model accuracy. Each tree is composed of branches ("nodes") representing yes–no questions where features (i.e., predictive variables) are used to split the dependent variable into two groups that minimize in-group variability and maximize between group variability. We selected a
random forest approach because these models require few assumptions about data structure (i.e., data need not conform

to assumptions of classical statistics such as linearity, normality, and constant variance), are robust to outliers, and generally perform as well or better than other data intensive approaches (Hagenauer et al., 2019). The use of random forest models also allows for the identification of predictors that are important to model accuracy, using measures such as condition permutation importance and post-hoc partial dependence plots (see additional details below).

### 2.8.2 Predictor Variables

Predictor variables for random forest (RF) models were assembled from the U.S. EPA StreamCat dataset[8]. StreamCat contains information for over 600 different environmental metrics linked to individual stream reaches in the NHDv2Plus dataset (Hill et al., 2016). These metrics summarize diverse geospatial attributes—including aspects of land cover, impervious surfaces and road density, soil type, point source and nutrient inputs, and climatic factors (temperature and precipitation), among others—at the catchment and watershed scale draining into each reach. "Catchments" (i.e., local drainage areas) include the immediate land area draining into each individual stream reach in the NHD excluding areas draining to upstream reaches; "watersheds" include the entire land area draining into each stream reach. StreamCat contains land use data for catchments and watersheds summarized from the National Land Cover Database (NLCD) for multiple years. We used land cover attributes only from the 2019 iteration of the NLCD (DeWitz, 2021). To supplement this dataset, we derived estimates of tile density (i.e., area tiled per area watershed) for each gaged watershed using estimates of tiled areas (30 m resolution) from Valayamkunnath et al. (2020). Prior to developing a random forest model, we excluded predictors from the StreamCat dataset that did not contain useful information (i.e., all rows=0). We also excluded attributes where information was missing ('NA') for >20% sites. Some of the remaining attributes still contained some missing values. Because random forest models cannot handle missing values in predictor variables, we used the *missRanger* package in R (Mayer, 2023) to impute the remaining missing values for the training and testing datasets. In total, we used 253 predictor variables in the model, after excluding variables that did not provide useful information (i.e., were all 0s or had too many missing values), that did not match the timing of our dataset (i.e., land cover data from years other than 2019), or were not especially relevant (e.g., variables describing forest fire intensity or extent). Prior to random forest modeling, we normalized (i.e., centered and scaled) numeric attributes to have a mean of zero and a standard deviation of one.

### 2.8.3 Model Tuning and Selection

We developed the random forest model to predict mean SRP during late summer low conditions, based on data for 127 gaged watersheds. Only 128 of the 143 total gaged watersheds in the study had >=3 SRP samples collected during late summer low flow conditions and were therefore used to calculate mean SRP values. Prior to model development, we excluded one additional site from the testing dataset (Buffalo Creek near Glencoe, MN) that had a mean SRP value for late summer that exceeded the range of SRP values in the training dataset (see Appendix Fig. S1). To develop the RF model, we used the same general approach to random forest modeling described in detail by Dolph et al. (2023). Data were split randomly into independent model training (70%, n=88) and model testing (30%, n=39) datasets. Using the training dataset and the *ranger* package in R (Wright and Ziegler, 2017), we applied tenfold cross validation to

---

[8] https://www.epa.gov/national-aquatic-resource-surveys/streamcat-dataset

tune model hyperparameters across a range of possible values. K-fold cross validation can assist in avoiding model over-fitting and works by partitioning training data into K equal sized "folds" (in our case 10). The model is iteratively trained on various combinations of tuning hyperparameters across K-1 folds, leaving the remaining fold to evaluate model performance for each combination. The hyperparameters selected for tuning were: mtry (i.e., number of variables randomly sampled as candidates at each split) and min_n (i.e., the minimum number of data points in a

node). The trees hyperparameter (i.e., number of trees) was set to 1000 across all models. We defined a grid of 20 potential combinations of hyper-parameters using the *tune_grid*() function from the *tidymodels* collection of packages in R (Kuhn and Wickham, 2020). This approach draws hyperparameter values semi-randomly from parameter space such that the various combinations cover the whole space of potential values. We selected hyperparameter values using out-of-bag (OOB) RMSE and $R^2$ for the associated models. Once hyperparameter values were tuned, we reran

the random forest model using the *randomForest* package (Liaw and Weiner, 2002), to create a *randomForest* object that was compatible with our selected measure of predictive variable importance (conditional permutation importance, see next paragraph). We evaluated overall model performance using $R^2$ and RMSE between predicted and observed SRP values in the independent test dataset (comprising 30% of the original dataset).

**2.8.4 Variable importance**

We used Conditional Permutation Importance (CPI) to evaluate the importance of predictors to model performance. CPI aims to capture the dependence between a predictor and the response variable, conditionally on the values of all other predictors. CPI can be used to assess how much each variable contributes to accurately predicting the response variable, given what we know from all other predictive variables. We implemented the CPI approach from the *permimp* package in R (Debeer and Stobl, 2021). In *permimp*, a threshold value, equal to 1- the *p*-value for the

association between predictor variables, is used to determine whether to include a predictor in the conditioning for the predictor of interest. We used the default value for the threshold parameter in permimp (0.95; Debeer and Strobl, 2021).

While the CPI method can rank predictors in terms of their importance to model accuracy, it does not convey information about the nature of the relationship between predictor variables and late summer SRP concentrations. To

visualize these relationships, we created partial dependence plots (PDPs) using the *partialPlot* function in R (part of the *randomForest* package, Liaw and Weiner, 2002). These plots illustrate the change in predicted SRP concentration when the values of one predictor are changed while all other predictors are kept constant at their original values (Greenwell, 2017). We generated PDPs for the top 15 predictor variables identified as most important by the measure of CPI.

**3 Results**

**3.1 SRP concentrations at gaged watersheds during low flow**

Across gaged watersheds, we expected SRP concentrations at low flow conditions to differ depending on the extent of historic and current P inputs associated with anthropogenic land use. Most gaged watersheds in our study region (90%, n=128) were substantially impacted by either agricultural or urban land use (defined here as watersheds with

355 >=50% crop cover and/or >=10% high intensity urban land use). The remaining watersheds (n=15) were characterized as 'less impacted'.

Among watersheds with substantial agricultural or urban influences, mean low flow SRP concentrations were highest in late winter, lowest in spring, and then increased progressively through early summer, late summer, fall and early winter (Table 1).

However, there was large variation (3–4 orders of magnitude) in low flow SRP concentrations across sites in any given season (range across all samples = 0.001–3.9 mg/L). For less impacted sites, seasonal low flow SRP concentrations were also highest on average during late winter, although the absolute concentrations were much lower than more heavily impacted sites. By contrast to more heavily impacted sites, mean low flow SRP concentrations at less impacted sites dropped in spring and stayed steady through summer, and dropped slightly again in fall. Less
impacted sites showed comparatively low SRP concentrations and lower variability in low flow SRP concentrations across sites or seasons (range 0.001-0.046 mg/L).

**Table 1: Mean, minimum and maximum low flow SRP concentrations (mg/L) for more heavily impacted gaged watersheds ( >=50% crop cover and/or >=10% high intensity urban land use) and less impacted gaged watersheds<50% crop cover and < 10% high intensity urban land use), across seasons.**

| Degree of anthropogenic disturbance | Season | Mean SRP | Min SRP | Max SRP |
|---|---|---|---|---|
| More impacted | Late Winter | 0.129 | 0.002 | 3.550 |
| | Spring | 0.030 | 0.001 | 0.384 |
| | Early Summer | 0.039 | 0.001 | 0.526 |
| | Late Summer | 0.055 | 0.001 | 1.350 |
| | Fall | 0.069 | 0.002 | 1.595 |
| | Early Winter | 0.117 | 0.002 | 3.900 |
| Less Impacted | Late Winter | 0.008 | 0.002 | 0.019 |
| | Spring | 0.007 | 0.002 | 0.031 |
| | Early Summer | 0.006 | 0.001 | 0.028 |
| | Late Summer | 0.005 | 0.001 | 0.046 |
| | Fall | 0.004 | 0.002 | 0.008 |
| | Early Winter | 0.008 | 0.002 | 0.030 |

**3.2 Influences of wastewater treatment facilities (point sources) on riverine SRP concentrations at low flow**

Mean SRP concentrations at low flow for gaged watersheds were significantly related to the density of wastewater treatment plants in the watershed during early winter, late winter, late summer, and fall but not in spring or early summer (Fig. 2). Part of the discrepancy across seasons may have been caused by the fact that few watersheds with a high density of wastewater treatment plants were sampled during low flows in spring and early summer. The relationship between mean low flow SRP and wastewater treatment plant density was strongest in early winter (though still somewhat weak overall; $R^2$=0.26) and comparatively weaker in other seasons (Appendix Table A1). These relationships were largely driven by watersheds where density of wastewater treatment plants was comparatively high (>0.005 sites/km$^2$). When watersheds with wastewater treatment plant density > 0.005 sites/km$^2$ were excluded, we observed a persevering very weak significant positive relationship between wastewater treatment plant density and mean lowflow SRP during late summer and late winter ($R^2$=0.06 and 0.10, respectively), but not during any other season. (see Appendix Table A1).

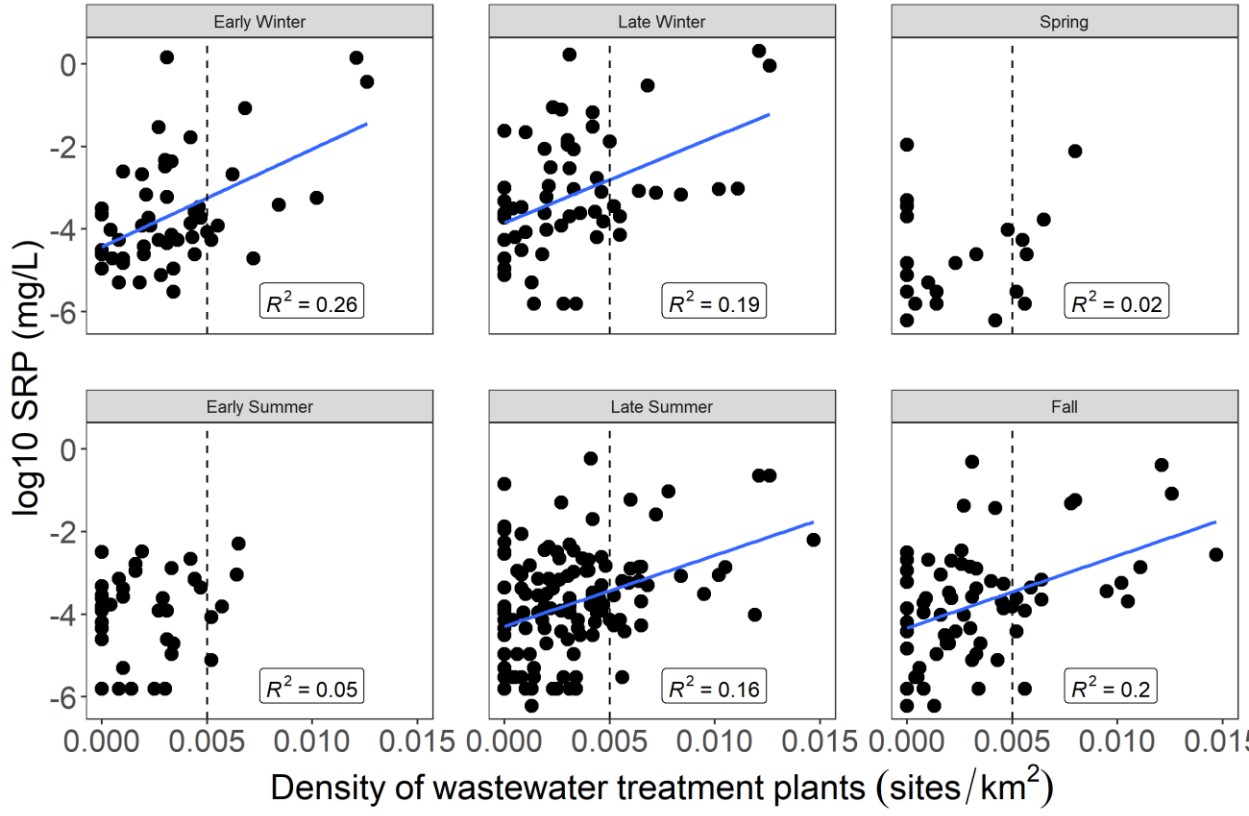

**Figure 2: Relationship of SRP concentrations at low flows (log scale) in gaged watersheds to the density of wastewater treatment plants (sites/km$^2$) in the watershed, by season. Blue lines indicate statistically significant linear regressions ($p <$ 0.05). Linear regression statistics are shown in Appendix Table A1. Dashed line indicates wastewater treatment plant density of 0.005 sites/km$^2$. Note that not all gaged watersheds had sufficient samples collected during low flows in each**

**season to generate mean values; thus, the number of gaged watersheds with low flow mean SRP values available was different during each season.**

### 3.3 Riverine SRP at low flows in relation to agricultural land use

We observed consistent and positive relationships between agricultural land use (% cropland) and mean low flow SRP

concentrations across gaged watersheds during all seasons, with the strongest relationships occurring during late summer and late winter (Fig. 3; Appendix Table A2). When we examined only sites without wastewater treatment plant influence, these relationships appeared even stronger, as evidenced by increased $R^2$ values (Fig. 4; Appendix Table A2). The strongest correlations were evident in late summer and early winter ($R^2$ of 0.69 and 0.86, respectively; Fig. 4). However, it should be noted that the sample size for early winter was small (n=5), which may have inflated

the $R^2$ value for this season.

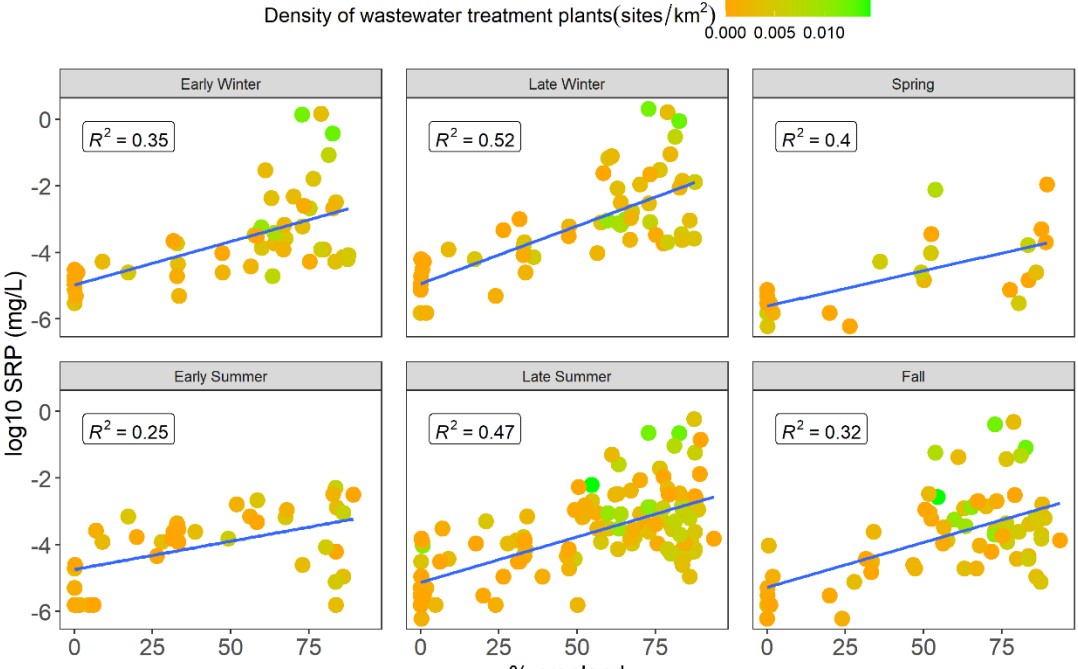

**Figure 3: Mean low flow SRP concentrations across gages (log scale), in relation to % crop cover, by season. Color scale indicates density of wastewater treatment plants in the watershed. Relationships in all seasons were significant and positive. Linear regression statistics are shown in Appendix Table A2.**

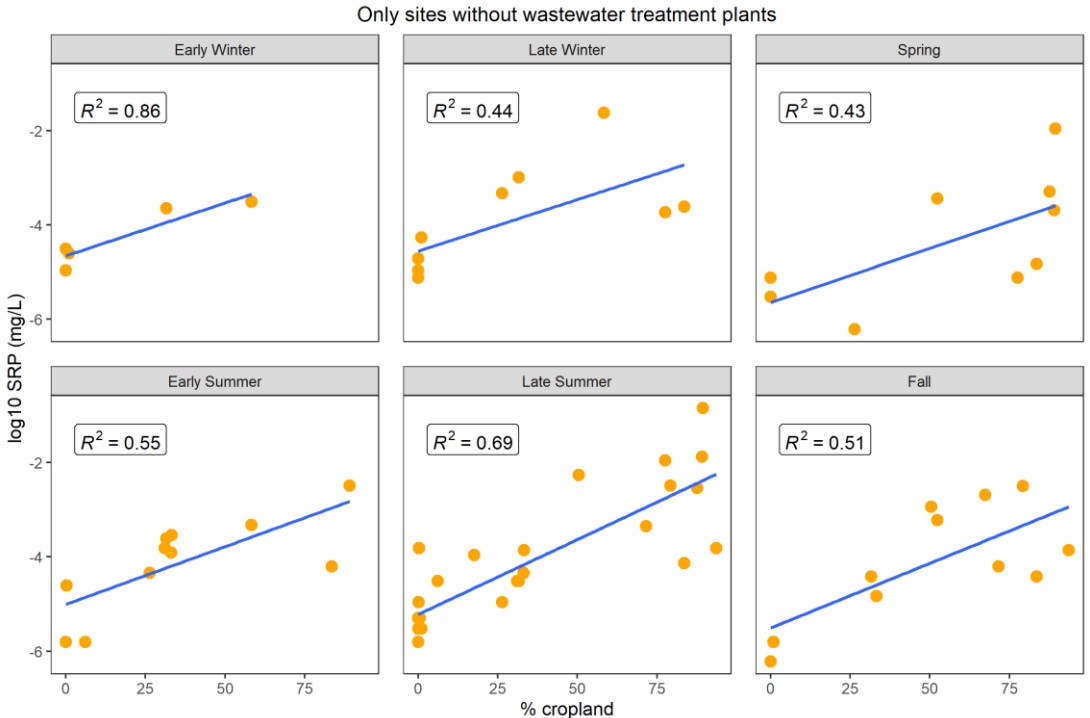

**Figure 4: Mean low flow SRP concentrations across gages (log scale), in relation to % crop cover, by season, for sites with *no wastewater treatment plant influence*. Relationships in all seasons were significant and positive. Linear regression statistics are shown in Appendix Table A2.**

### 3.4 SRP concentrations at tile outlets

Across tile outlets for 10 conventionally farmed fields (corn-soybean rotation), mean SRP concentration of tile drainage was highest in late winter (mean SRP = 0.131 mg/L) and lowest in early, late summer and early winter (mean
SRP = 0.03 mg/L; Table 2). Two sites (WR1 and ST1) included dairy operations, and two sites (BE1 and DO1) included swine finishing, in addition to row crops. The two dairy-influenced farm fields (WR1 and ST1) had notably higher tile SRP concentrations across all seasons relative to other sites. Three sites (MC1 RE1, and WR1) had one surface inlet to the tile system (all other inlets were subsurface). These sites appeared to have higher mean SRP concentrations in late winter (coinciding with snowmelt) and early summer (in the case of WR1), but of the surface
inlet sites only WR1 (the dairy farm site) had higher mean SRP concentrations in late summer.

**Table 2: Mean flow-weighted daily SRP concentrations (mg/L) from farm tile outlets, by season. Tile outlet data were collected from 10 farms between 2011-2021. Note that one farm field (NOW1) had two tile outlets (NOW1-N and NOW1-S) that drained different areas of the same field. Early winter: Nov-Dec; Late winter: Jan-Mar; Spring: Apr-May; Early summer: Jun-Jul; Late summer: Aug-Sept; Fall: Oct. [a]Farms included dairy operations. [b]Farms included a surface inlet to**
**tile drainage system. [c]Farms included swine finishing.**

| Site | Early Winter | Late Winter | Spring | Early Summer | Late Summer | Fall | Annual mean |
|---|---|---|---|---|---|---|---|
| BE1[c] | 0.064 | 0.053 | 0.031 | 0.017 | 0.023 | 0.132 | 0.036 |
| DO1[c] | 0.012 | 0.029 | 0.036 | 0.015 | 0.018 | 0.010 | 0.023 |
| MC1[b] | 0.019 | 0.139 | 0.023 | 0.012 | 0.022 | 0.024 | 0.037 |
| NO1W-N | 0.008 | 0.011 | 0.018 | 0.015 | 0.016 | 0.008 | 0.014 |
| NO1W-S | 0.017 | 0.025 | 0.024 | 0.012 | 0.018 | 0.015 | 0.018 |
| RE1[b] | 0.014 | 0.070 | 0.045 | 0.075 | 0.022 | 0.043 | 0.049 |
| RW1N | 0.023 | 0.231 | 0.025 | 0.014 | 0.011 | 0.033 | 0.061 |
| RW1S | 0.011 | 0.278 | 0.059 | 0.019 | 0.017 | 0.045 | 0.069 |
| ST1[a] | 0.053 | 0.164 | 0.091 | 0.062 | 0.073 | 0.048 | 0.084 |
| WI1 | 0.011 | 0.008 | 0.009 | 0.008 | 0.005 | 0.010 | 0.008 |
| WR1[a,b] | 0.055 | 0.307 | 0.156 | 0.119 | 0.157 | 0.105 | 0.151 |
| **All sites** | **0.029** | **0.131** | **0.051** | **0.036** | **0.033** | **0.035** | **0.052** |

**3.5 Riverine SRP at low flows compared to tile concentrations**

We evaluated SRP during low flow conditions for each gaged watershed in each season (Table A3), and compared these riverine SRP values to SRP concentrations in monitored tile outlets. Note that not all gaged watersheds had sufficient samples collected during low flows in each season to generate mean values (Table A4); thus, the number of gaged watersheds with low flow mean SRP values available was different during each season. Comparisons for late winter, spring and late summer are shown in Fig. 5 (only a subset of seasons are shown for improved clarity in data visualization; similar figures for early winter, early summer and fall are shown in Appendix Fig. A2).

In early winter, 36% of gaged watersheds (n=18/50 sites for which low flow data was available) exhibited SRP concentrations at low flows that were higher than mean tile SRP concentration. Six of these watersheds were characterized by comparatively high wastewater treatment plant density (defined as >0.005 sites/km$^2$). In late winter when tile SRP concentrations were highest, 23% of gaged watersheds (n=13/57) exhibited SRP concentrations at low flows that were higher than mean tile concentrations. A minority of these sites (3/13) had considerable wastewater treatment plant influence (wastewater treatment plant density > 0.005 sites/km$^2$). In spring, SRP concentrations during low flow conditions were uniformly low across nearly all gaged watersheds. SRP samples collected during low flow conditions were fairly uncommon, with only 23 watersheds having >=3 SRP samples collected during spring low flows. Of these, two sites (9%) had SRP concentrations at low flows that were higher than mean tile concentrations. In early summer (Jun-Jul), 28% of sites (n=11/40) had SRP concentrations at low flow that were higher than mean tile concentrations. Two of these sites had considerable wastewater treatment influence. In late summer (Aug-Sep), 39% of gaged watersheds (n=50/128) had SRP concentrations at low flow that were higher than mean tile concentrations, and 16 of these sites had considerable wastewater treatment influence. In fall (Oct), 35% of gaged watersheds (n=24/68) sites had SRP concentrations at low flow that were higher than mean tile concentrations; eight of these sites had comparatively higher wastewater treatment plant density.

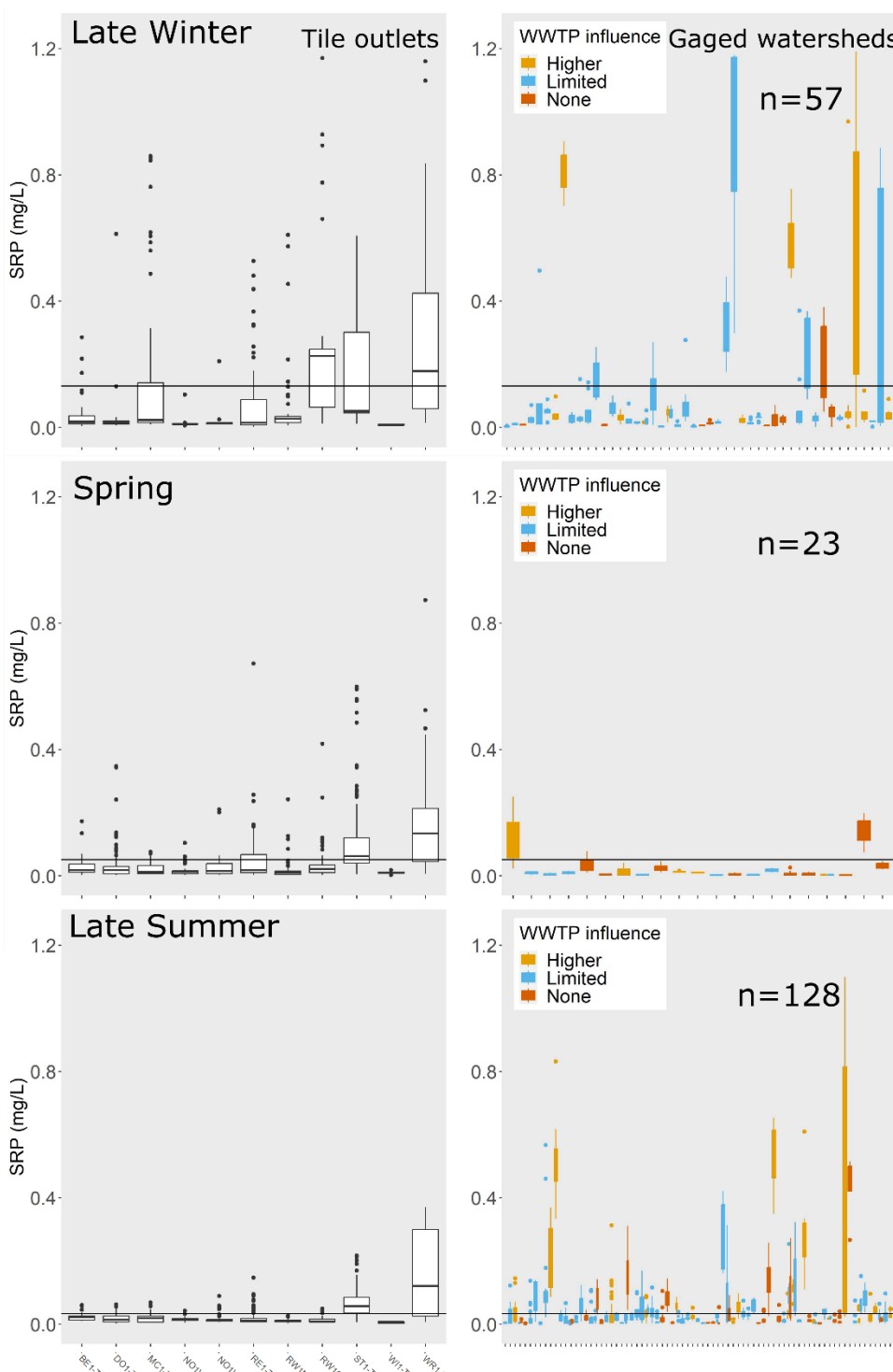

**Figure 5: SRP concentrations (mg/L) for tile outlets and during low flow conditions for gaged watersheds, by season. Only a subset of seasons is shown for improved clarity in data visualization; similar figures for remaining seasons are shown in Appendix Fig. A2. The horizontal line in each plot is the mean SRP concentration among tile outlets for that season. For gaged watersheds, color of boxplots indicates degree of influence from wastewater treatment plants: light orange: wastewater treatment plant density >0.005 sites/km$^2$; blue = wastewater treatment plant density <0.005 sites/km$^2$ but greater than zero; dark orange: no wastewater treatment plant sites in watershed. The number of gages for which low flow data was available in each season is printed on the plot. To improve data visibility, the y-axis for SRP was limited to a maximum of 1.25 mg/L, which eliminated a small number of outliers from the plots for tile outlets (n=34 out of 11,079 records) and gaged watersheds (n=16 out of 2,696 low flow records).**

## 3.6 Low flow SRP concentrations from additional field sites

Among gaged stream and river sites, small order systems (especially first through third order ditches and streams) were under-represented relative to their prevalence across the landscape. These smaller order systems are also less likely to have substantial point source discharges. To get a better understanding of SRP conditions in smaller order

systems, we examined SRP concentrations collected from 33 agriculturally dominated ditches, streams and mid-sized rivers in southern Minnesota during low flow conditions in August of 2014 (Dolph et al., 2017). During this sampling event, SRP concentrations at most sites were higher than mean SRP concentrations from farm tile outlets (Fig. 6). Mean SRP concentrations in late summer were highest in ditches (0.19 mg/L) and intermittent streams (0.19-0.30 mg/L).

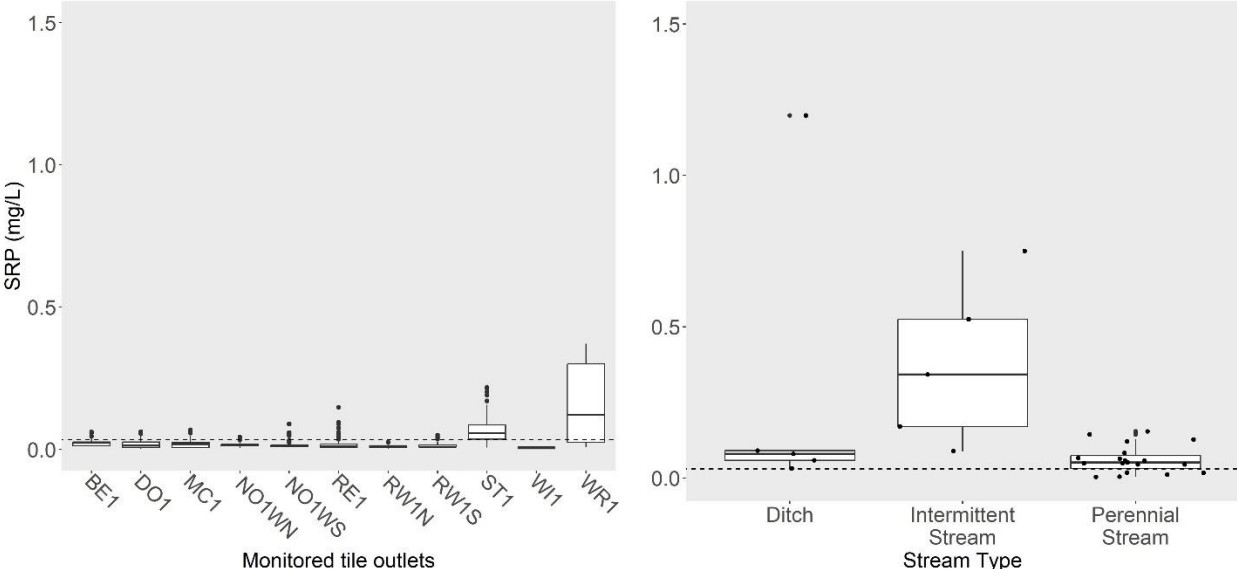

**Figure 6: SRP concentrations (mg/L) among tile outlets (left panel) compared to sampled ditches (n=5), intermittent streams and rivers (n=6), and perennial streams/rivers (n=22) (right panel) during late summer low flow conditions. Dashed line shows mean SRP concentration for tile outlets in late summer.**

**3.7 *C-Q* relationships at stream and river gages**

When *C-Q* relationships were evaluated using all flow data for each gage, the majority of gaged watersheds (72%, n=103) showed mobilizing behavior for SRP in relation to stream flow (i.e., significant positive slopes for the *C-Q* power law relationship and $CV_C/CV_Q > 0.3$; Fig. A3). Mobilizing behavior for bioavailable P ranged from very weak ($R^2=0.01$) to comparatively strong ($R^2=0.68$). Watersheds with positive SRP-Q relationships were located predominantly in the agriculturally dominated regions of the state (the southern and western parts of the state corresponding to the southern part of the Upper Mississippi River Basin, the Minnesota River Basin, the Driftless areas in the southeast, and the Red River Basin; Fig. 7). Chemodynamic behavior (non-significant slopes for the *C-Q* power law relationship and $CV_C/CV_Q > 0.3$ was observed for 24% (n=34) of sites, most located in the central and northeastern parts of the state dominated by forest and wetland cover (Fig. 7). A small number of sites (n=4, 3%) showed diluting behavior for SRP, as defined by significant negative slopes for the *C-Q* power law relationship and $CV_C/CV_Q > 0.3$. Two of these sites showed considerable wastewater treatment plant influences (wastewater treatment plant density > 0.005 sites/km[2;] Table A3). Three sites (2%) showed chemostatic behavior for SRP transport, as defined by a $CV_C/CV_Q <= 0.3$.

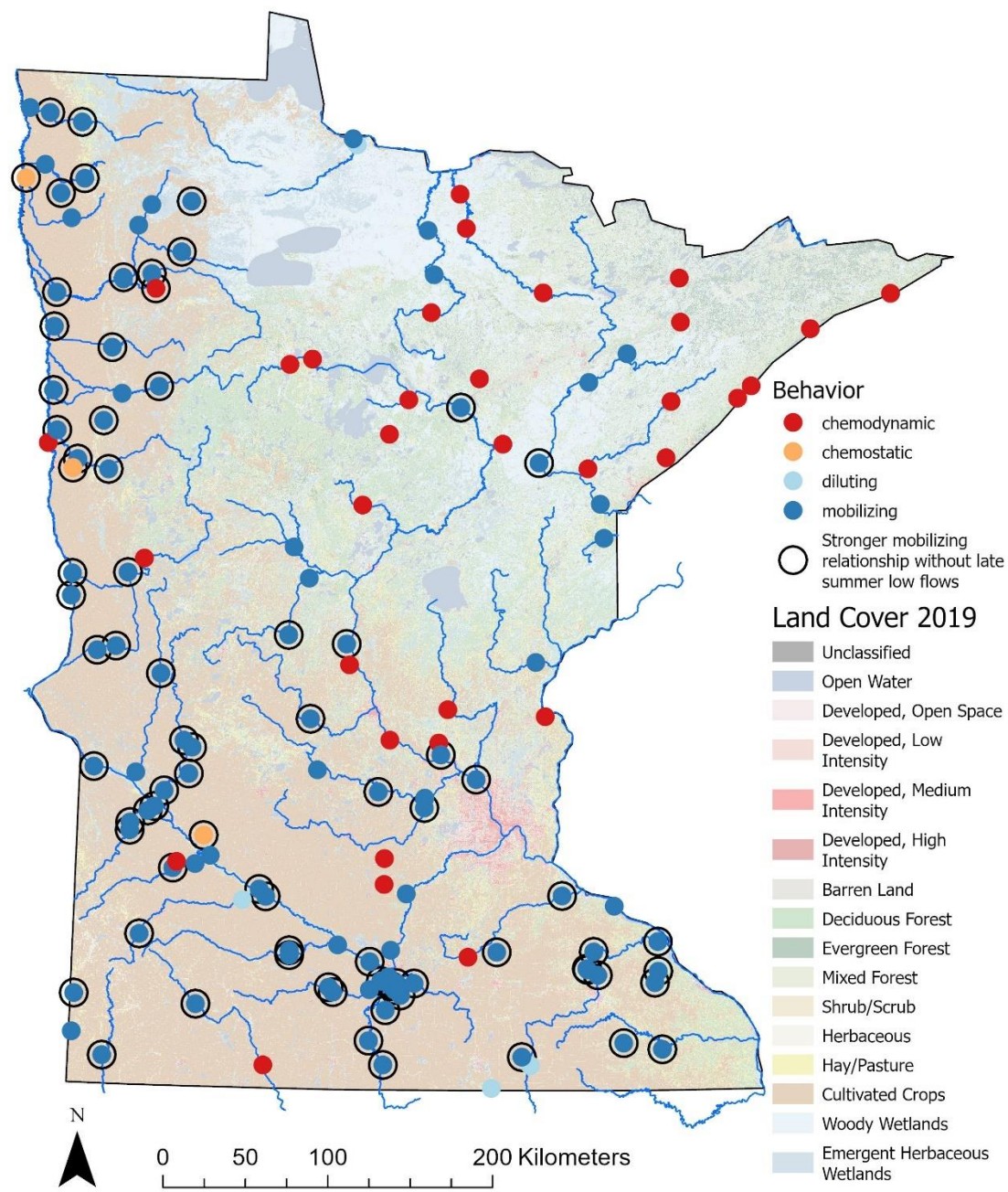

**Figure 7:**

**Transport behavior of SRP in relation to flow (Q) for gaged watersheds (n=143). Dots indicate gage locations. Color of dots indicates transport behavior diagnosed by slope *b* of the *C-Q* power law relationship and $CV_C/CV_Q$. Mobilizing = black dots (n=103); Chemodynamic = light orange dots (n=34); Diluting = purple dots (n=4); Chemostatic = dark orange (n=3). Sites with a stronger mobilizing relationship (i.e., an increase in slope *b*) when late summer low flows were excluded are shown as open circles (n=78). Land cover is based on the 2019 National Land Cover Database.**

When low flow samples from late summer were removed, 54% of gaged watersheds (n=78) exhibited an increase in the slope of the mobilizing relationship between SRP and Q (Fig. 7; Appendix Table A5). For these sites, slopes of the *C-Q* relationships increased by 23%, on average, after late summer low flow samples were excluded (range in percent slope increase was 0.1%-273%). In other words, mobilizing behavior for SRP was stronger when these late summer low flow conditions were excluded. Examples of this phenomenon for four different gaged watersheds are

shown in Fig. 8, where the slope of the C-Q relationship is steeper when late summer low flow samples were excluded, and comparatively flatter when they are included. Watersheds where late summer low flows modulated (flattened) the slope of the C-Q relationship for SRP were again located predominantly in the agriculturally dominated regions of the state (Fig. 7).

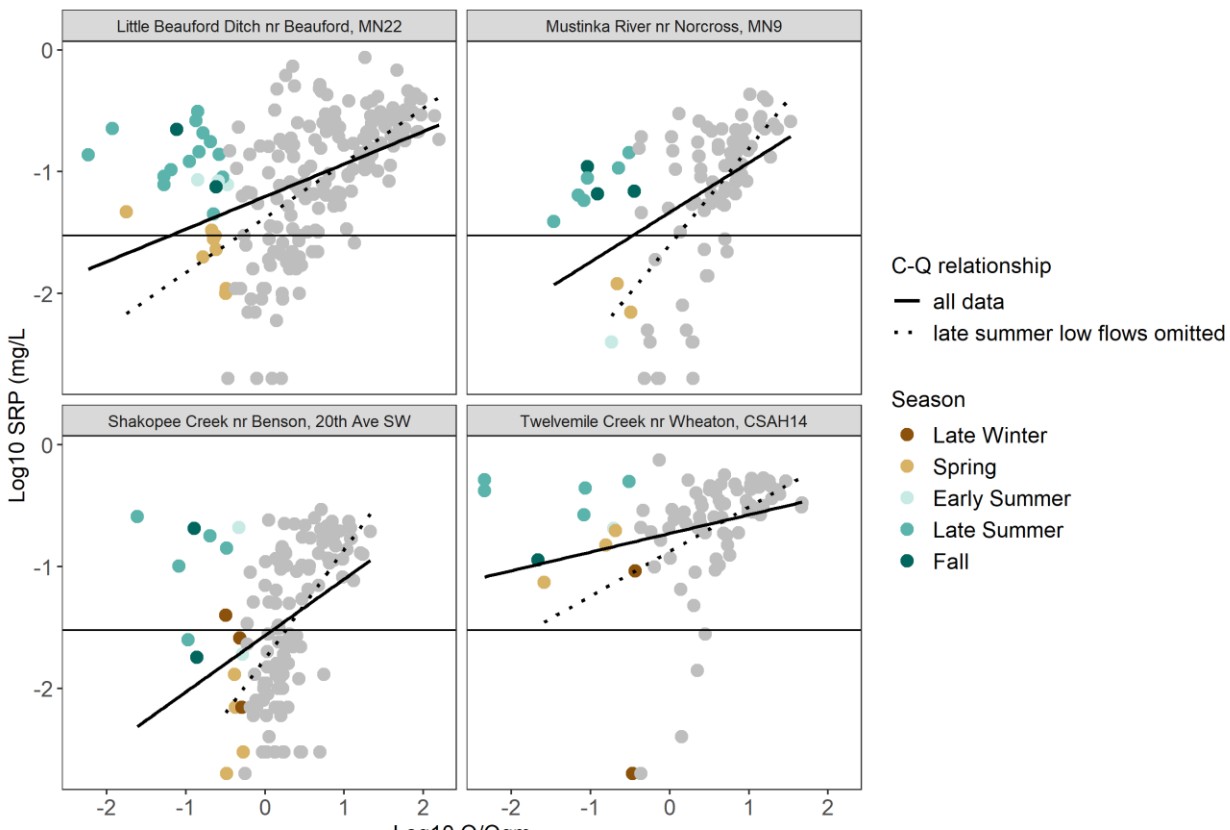

**Figure 8: Example watersheds where low flows in late summer modulate the slope of the *C-Q* relationship for SRP. Low flow samples are shown as colored points where color indicates the season in which they were collected. All other samples are shown in gray. When all data are included, the slope of the overall *C-Q* relationship is reduced (solid line) compared to slopes for analyses with late summer low flow samples omitted (dashed line), indicating stronger mobilizing behavior.**

### 3.8 Regression analysis to identify drivers of elevated SRP concentrations in late summer

The final selected hyperparameters for the random forest model based on model tuning with tenfold cross validation for this dataset were mtry = 7, trees = 1000, min_n = 6. Evaluation of predicted vs. actual late summer SRP for the independent test dataset indicated a model RMSE of 0.10, and an $R^2$ of 0.41 (Fig. A4). On average, the random forest

model underestimated actual mean SRP concentrations during late summer low flows compared to actual measured

values among test sites (mean of estimated - true values = -0.005). However, this negative bias was driven by poor predictive model performance for three sites with exceptionally high mean SRP concentrations of 0.293, 0.525, and 0.526 mg/L (Figure A4). For the remaining test sites (all with mean SRP <0.2 mg/L), predictive bias (mean of estimate - true values =0.02) was actually positive. For most sites, in other words, the random forest model over-estimated mean late summer SRP concentrations compared to measured values.

The top 15 predictors to model performance are shown in Fig. 9. Importance values for all predictors are shown in Table A6. Partial dependence plots for these top predictors (Fig. 10) showed that higher SRP during late summer low flow conditions was associated with: higher cropland land use in riparian areas, various soil characteristics (higher soil erodibility, lower soil permeability, higher soil clay content), greater agricultural intensity (higher pesticide use, higher phosphorus uptake by crops, higher fertilizer application rates), more urban land use in riparian areas, lower woody wetland, and mixed forest in riparian areas, lower grassland land use in watersheds, lower surplus precipitation in the watershed (precipitation minus evaporation) and higher stream temperatures. Table 3 summarizes possible mechanisms linking these attributes to riverine SRP concentrations.

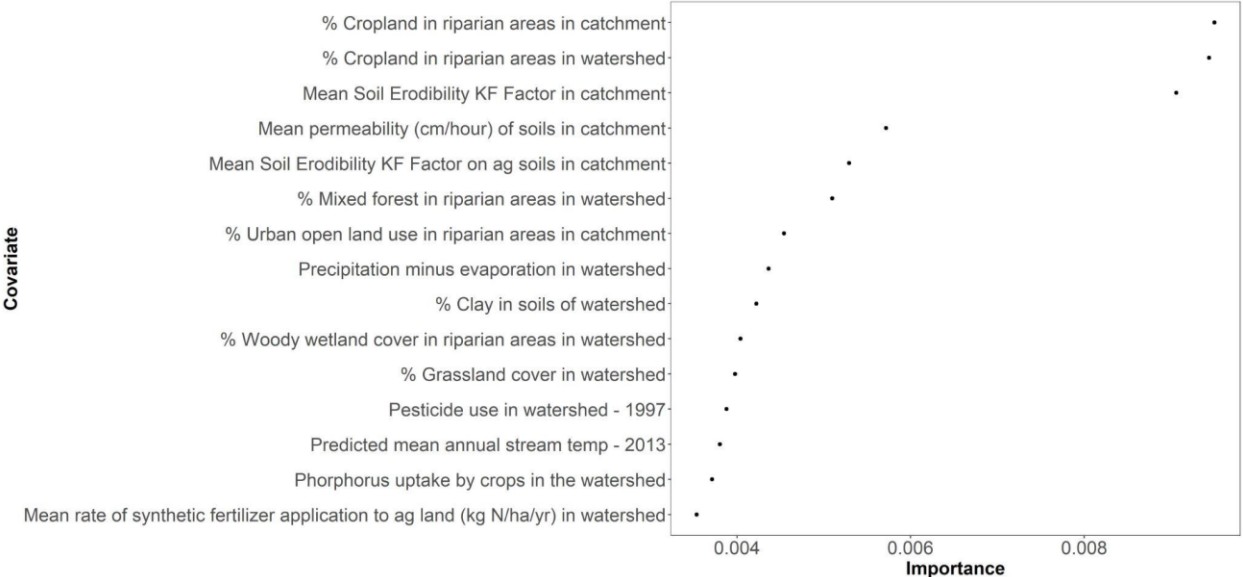

**Figure 9: Conditional Permutation Importance (CPI) values for the top 15 predictors in the random forest model for late summer SRP during low flows and stream and river gages. CPI is a measure that can be used to assess how much each variable 'adds' to accurately predicting the response variable, given what we know from all other covariates. Importance values for all attributes are given in Table A6.**

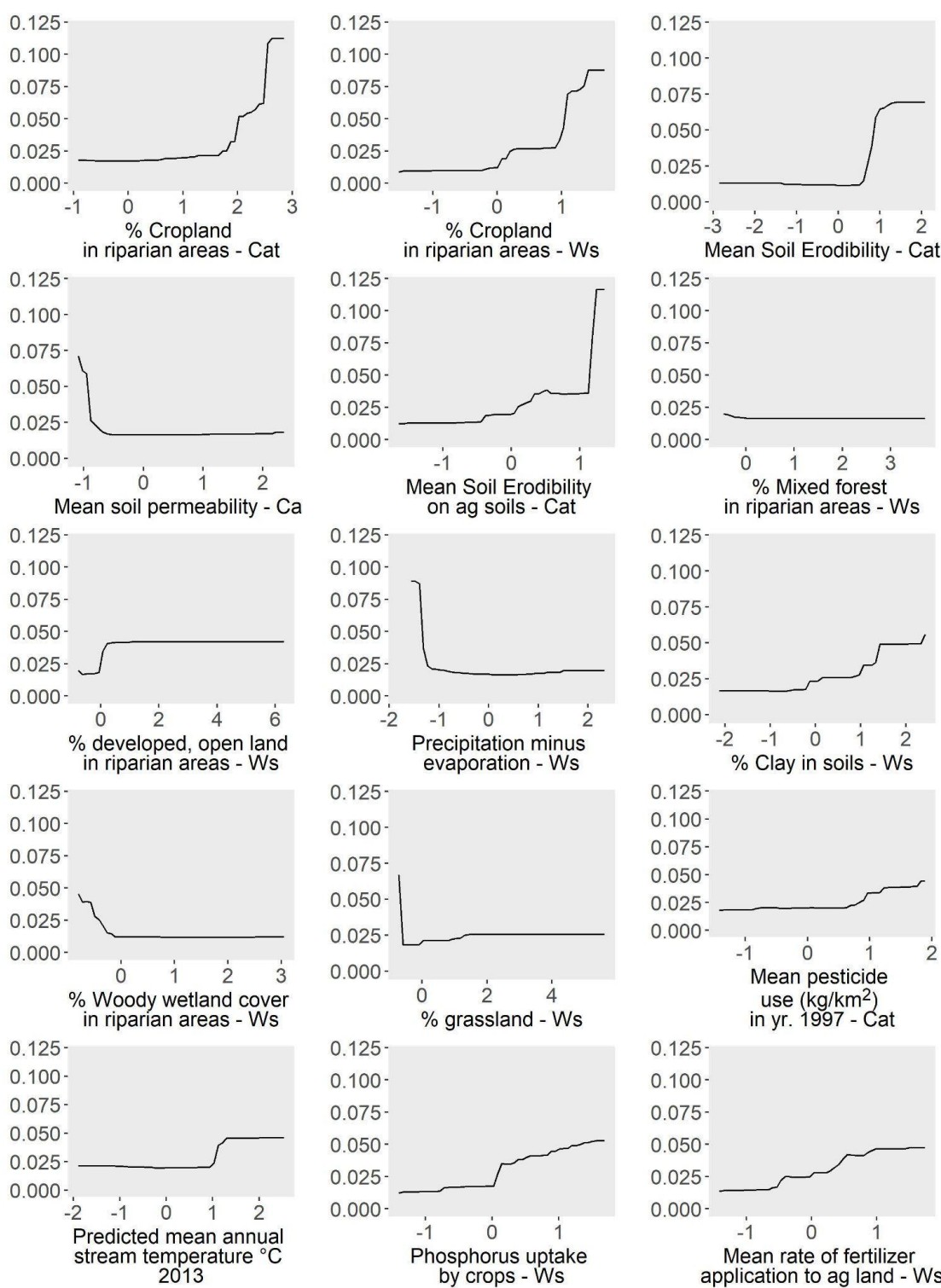

**Figure 10: Partial dependence (y axis = change in predicted SRP value) for each of the 15 most important predictors to model performance. Partial dependence shows the change in the response variable (late summer low flow SRP) when each predictor of interest is varied while all other predictors stay constant. "Ws"=predictors summarized at watershed scale, "Ca"=predictors summarized at catchment scale. All predictor variables are from the U.S. EPA StreamCat dataset (Hill et al., 2016).**

**Table 3: Possible mechanisms linking random forest model predictor variables to late summer SRP concentrations during low flow conditions, for the top 15 attributes identified as most important to the performance of the random forest model. All attributes are from the U.S. EPA StreamCat dataset.**

| Attribute | Relationship | General mechanism category | Potential specific mechanism(s) linked to elevated late summer SRP |
|---|---|---|---|
| Percent of local NHD catchment classified as crop land use (NLCD 2019) within a 100-m buffer of NHD streams | Increasing SRP with increasing crop cover grown in proximity to river network | Direct inputs and/or legacy P supply | Indicator of current and historic P inputs from ag land use, especially in local riparian areas |
| Percent of watershed classified as crop land use (NLCD 2019) within a 100-m buffer of NHD streams | Increasing SRP with increasing crop cover grown in proximity to river network | Direct inputs and/or legacy P supply | Indicator of current and historic P inputs from ag land use, especially in local and upstream riparian areas |
| Mean soil erodibility factor (kffact, via STATSGO) within local NHD catchment; represents a relative index of susceptibility of bare, cultivated soil to particle detachment and transport by rainfall | Increasing SRP with increasing soil erodibility | Direct inputs and/or legacy P supply | Inputs of soil-associated P into river networks, either current or historic |
| Mean permeability (cm/hour) of soils (STATSGO) within local NHD catchment | Higher SRP at very low soil permeability | Mediates biogeochemical processes in near channel environment | Low permeability impedes oxygen exchange to the hyporheic zone, facilitating redox-mediated P release |
| Percent of watershed classified as mixed forest land cover (NLCD 2019) within a 100-m buffer of NHD streams | Higher SRP with low mixed forest cover in riparian areas | Direct inputs and/or legacy P and/or mediates biogeochemical processes | Fewer current or historic inputs of P to river networks with forested riparian areas; potential for forested riparian areas to trap P; shading of river channel alters stream productivity/stream metabolism |
| Percent of local NHD catchment classified as developed, open space land use (NLCD 2019) within a 100-m buffer of NHD streams | Higher SRP with higher open urban land use in riparian buffer | Direct inputs and/or legacy P supply | Indicator of urban land use and associated P inputs; could also mediate biogeochemical processes by altering stream temperature or flow conditions (i.e., higher stream temps in urban areas; stormwater infrastructure contributing to low flow conditions during dry periods, etc) |
| Precipitation (mm) minus potential evaporation within | Higher SRP at low and high ends of | Mediates biogeochemical | Dry conditions contribute to higher stream temperatures |

| watershed | precipitation range | processes in near channel environment | and lower discharge, influencing biologically-mediated P release |
|---|---|---|---|
| Mean % clay content of soils within watershed | Higher SRP with greater clay content in soils | Mediates biogeochemical processes in near channel environment | Fe-containing clay sediments can adsorb P and release it via redox reactions |
| Percent of watershed classified as woody wetland land cover (NLCD 2019) within a 100-m buffer of NHD streams | Higher SRP with low woody wetland cover in riparian buffers | Mediates biogeochemical processes in near channel environment | Woody wetlands acting as sinks for SRP |
| Percent of local catchment classified as grassland/ herbaceous land cover (NLCD 2019) | Higher SRP at low grassland cover | Direct inputs and/or legacy P supply | Fewer historic and ongoing P inputs in grasslands vs ag/urban lands |
| Mean pesticide use (kg/km2) in yr. 1997 within watershed | Increasing SRP with increasing pesticide use | Direct inputs and/or legacy P supply | Indicator of agricultural intensity and degree of historic/current P inputs |
| Predicted mean annual stream temperature for 2013 | Higher SRP with higher stream temperatures | Mediates biogeochemical processes in near channel environment; Proxy for direct inputs and/or legacy P supply | Indicator of temperature-mediated biological activity, or of land use differences across climate gradients associated with P inputs (e.g., more agriculture & associated legacy P supplies in warmer climates) |
| Phosphorus uptake by crops in the watershed | Increasing SRP with increasing P uptake | Direct inputs and/or legacy P supply | Indicator of agricultural intensity and degree of historic/current P inputs |
| Mean rate of synthetic nitrogen fertilizer application to agricultural land in kg N/ha/yr, within the watershed | Increasing SRP with increasing fertilizer inputs | Direct inputs and/or legacy P supply | Indicator of agricultural intensity and degree of historic/current P inputs |

## 4 Discussion

In this study we observed that between one third to one half of the gaged watersheds in Minnesota exhibited SRP concentrations during late summer low flows that were above previously identified thresholds for eutrophication of 0.02 - 0.04 mg/L for freshwater environments (Zeng et al., 2016; Poikane et al., 2021; (34% were above a threshold of 0.04 mg/L and 53% of watersheds were above 0.02 mg/L). One avenue for future research is to investigate how the timing and duration of elevated summer SRP concentrations affect local and downstream eutrophication outcomes. On the one hand, the large majority of annual P export by load likely occurs under high flow conditions in late winter and spring (Dolph et al., 2019; Schilling et al., 2020). However, the release of highly bioavailable P during hot, dry summer periods when conditions are optimal for algal growth in lakes and rivers may also drive increased eutrophication risk, resulting in outcomes such as increased occurrence of harmful algal blooms (Paerl and Huisman, 2008). As climate change is expected to result in increased prolonged periods of drought and heat during summers in the Upper Midwest (Wilson et al., 2023), the effects of elevated bioavailable P at low flows could be extended for longer parts of the season.

We also observed that, formore than half of the gaged watersheds we studied (54%), elevated SRP concentrations during low flows in late summer dampened $C$-$Q$ relationships which would have otherwise appeared more strongly mobilizing across other seasons and flow conditions. Strongly mobilizing relationships are indicative of landscape connectivity as a key driver for SRP export (Musloff et al., 2015), with flow accumulation and riparian areas identified as critical source areas for SRP (Casquin et al., 2020; Dupas et al., 2023). Thus, for many of the sites we studied, connectivity appears important to SRP export during winter, spring and early summer and during moderate to high flow conditions at all times of year. During late summer low flows, by contrast, other in-channel dynamics may cause riverine $C$-$Q$ patterns to deviate from linear relationships (Meybeck and Moatar, 2012). Below we discuss possible mechanisms that may contribute to comparatively high SRP concentrations during late summer low flow conditions in streams and rivers of our study region.

### 4.1 Drivers of SRP during late summer low flows

Overall, our analysis shows that landscape drivers that govern diffuse P inputs and legacy P supply in the river network, as well as wastewater inputs and biogeochemical processes, are associated with high late summer SRP concentrations during low flows at many anthropogenic-ly-dominated sites. Crop cover was strongly and directly related to SRP concentrations during low flow conditions in all seasons, and crop cover in riparian areas at the local catchment and watershed scales were the top two most important variables to the performance of the random forest model used to predict late summer low flow SRP concentrations. Other top variables to model performance included aspects of agricultural intensity at the watershed or catchment scale (pesticide use, phosphorus uptake by crops, and fertilizer application), as well as urban land use in riparian areas. The importance of these variables points to historic and ongoing inputs of P arising from intensive/industrial agriculture and urban land use that have resulted in the accumulation of legacy P in riverine channels, which can potentially be released under environmental conditions such as warm temperatures, low oxygen and variable moisture. Conversely, greater mixed forest or woody wetland land use in riparian areas was associated with lower SRP concentrations during late summer low flows, perhaps because

these environments may act as sinks for bioavailable P (Ury et al., 2023). Overall, it is notable that land use in riparian areas showed up as top variables of importance to model performance, suggesting that near channel environments (and therefore potentially near channel management practices) may be important in regulating elevated SRP during late summer low flows. Lastly, both geologic and climatic variables (soil erodibility, soil permeability, clay content of soils, mean annual stream temperatures, and precipitation minus evaporation) were also identified as important in the random forest model predicting late summer low flow SRP, suggesting that environmental factors which mediate biogeochemical processes also likely play an important role in driving late summer riverine SRP concentrations.

Interestingly, SRP concentrations during late summer low flow conditions in anthropogenic-ly-dominated watersheds often exceeded tile SRP concentrations. Although tile drainage is known to represent a key input of P to river networks (Smith et al., 2015), it may be that in-channel dynamics beyond tile concentrations drive variability in SRP concentration during summer low flows. However, it is also important to note that the two tile outlets draining farms with dairy operations exhibited much higher SRP concentrations during late summer (and all times of year), compared to tile outlets draining fields characterized only by corn and soybean row crops. Thus, the prevalence of CAFOs and other animal agriculture operations is likely to strongly influence the contribution of tile drainage to riverine SRP concentrations. Three sites also had a surface inlet to their tile drainage system. These tile systems had comparatively high SRP during winter, spring or early summer (depending on the site), which can likely be explained by the additional loss of sediment and nutrients to surface inlets during snowmelt on frozen and thawing soils (Feyereisen et al., 2015). However, the two surface inlet-influenced sites without dairy operations exhibited SRP concentrations during late summer similar to other nondairy impacted sites.

Wastewater treatment plant density did not rank among the most important predictors in our random forest model performance, and a substantial portion of sites (38%) exhibited elevated SRP concentrations (above 0.02 mg/L) during late summer low flow conditions despite having limited or no wastewater treatment plant influence in their watersheds. However, the influence of wastewater on summer low flow SRP was evident in elevated SRP concentrations at sites with strong wastewater influence throughout most seasons (apart from spring, when low flow SRP concentrations were nearly universally low, presumably due to rapid in-stream uptake or abiotic immobilization). We also observed direct (though weak) correlations with low flow SRP in late summer and wastewater treatment plant density in gaged watersheds. However, a sizeable number of the streams and rivers we studied (38%)

**4.2 Biogeochemical processes and riverine SRP**

Previous studies have identified a number of biogeochemical processes that can affect riverine concentrations of SRP at low flows. These processes include: 1) the concentration of legacy P entering the stream via groundwater and/or streambed pore water, 2) redox-driven release of P from stream sediments, and 3) release of P resulting from mineralization of organic matter.

During low flow conditions, groundwater and/or pore water can become proportionately dominant components of flow, with stores of legacy P in these sources contributing more strongly to overall riverine SRP. These groundwater

sources can include tile drainage (Schilling et al., 2020; Rode et al., 2023), but can also include streambed pore water entering from the hyporheic zone via upwelling flow paths with P concentrations that are distinct and potentially higher than that of tile drainage (Vissers et al., 2023). Upwelling of P-rich pore water can be patchy and is likely controlled by hyper-local spatial and temporal conditions operating at the reach scale, such as the availability and extent of reducing vs oxic conditions (e.g., Vissers et al., 2023).

SRP can also be released into river channels from stream sediments. Stream sediments often have the potential to buffer stream SRP concentrations by adsorbing P (Simpson et al., 2021). However, this buffering capacity will depend on sediment and stream characteristics, including sorption affinity, stream pH, exchangeable P concentration, sediment particle sizes, and seasonal variation in temperature, light, discharge, redox, primary productivity, stream respiration and sediment inputs (Simpson et al., 2021). Seasonal release of SRP is commonly thought to occur via the reduction of Fe-, Mn- or Al- oxyhydroxide-containing sediments under anoxic conditions, releasing $PO_4^{3-}$. These anoxic conditions typically arise when flow velocities are low, water and sediment temperatures increase, and oxygen becomes depleted due to increased microbial activity. For example, Smolders et al. (2017), showed that high summer concentrations of bioavailable P for rivers in Belgium were likely explained by internal loading from legacy P that was released from sediments when dissolved oxygen concentrations were low and P:Fe molar ratios in sediment were large.

Lastly, Jarvie et al., (2020) showed that, in a wetland-pond system, microbial respiration and the resulting mineralization of organic matter can also represent a source of bioavailable P under low flow conditions in summer and fall. They found that SRP release was potentially related to drier and hotter conditions that could facilitate both higher rates of biomass accumulation and its subsequent breakdown via microbial processes. Presumably, this dynamic could also be at play for slow moving ditches and streams in parts of our study region, where water is sometimes nearly stagnant in summer. Under low flow conditions and warm temperatures, ditches and streams may operate in ways that are similar to wetlands or other lentic water bodies. The stream network is also populated with in-channel and riparian wetlands that may further affect ambient SRP concentrations. Felton et al. (2023) found elevated dissolved P concentrations along a longitudinal stream gradient where the channel intersected wetlands and concluded that locally elevated SRP could reflect P release from decomposition of organic matter in wetland environments; however, in that study elevated P concentrations did not persist downstream and were assumed to be rapidly assimilated or adsorbed to sediments.

Our findings provide some insight as to the relative importance of these potential in-channel processes in determining seasonally elevated SRP concentrations at low flow. The importance of climate and geologic variables in the random forest model we used to predict late summer low flow SRP suggests that characteristics of stream sediments and/or climate-mediated biotic activity may play an important role in elevated SRP concentrations in late summer. Partial dependence plots indicated that increased SRP during late summer low flows was associated with the drier conditions (lower precipitation minus evaporation in gaged watersheds) and warmer conditions (higher predicted mean stream

temperatures)[9]. This finding could be consistent with an important role for biologically mediated processes such as microbial respiration that are affected by temperature and stream discharge. Microbial activity is important both in the decomposition of organic matter (i.e., mineralization), as well as in the reduction of redoximorphic sediments (i.e., sediments containing Fe, Al, Mn, etc.), both of which can result in the release of SRP. The predicted mean stream temperature values used in this study were derived from Hill et al. (2013), and were themselves influenced by air

temperature, soil permeability, agricultural and urban land use, stream slope, the influence of reservoirs, and watershed area. The positive relationship between stream temperature and late summer SRP at low flow needs further investigation but could also be related to greater influence of groundwater or to climate gradients that correspond to variation in biological activity or in land use and associated P inputs.

Soil erodibility was also identified as one of the most important variables to random forest model performance, with

700 partial dependence plots showing higher SRP during late summer low flows corresponding with greater soil erodibility in the local catchment. Eroded soils have long been understood as a primary vector by which P enters river networks (Berhe et al., 2018). Recently, this understanding has expanded to include eroded stream bank sediments as an additional driver of downstream P transport (Margenot et al., 2023). Sediment-associated P may be temporarily stored in river channels, with desorption of P occurring under certain environmental conditions, as described above.

Partial dependence plots showed that late summer low flow SRP concentrations were highest where soil permeability of soils in local catchments was low. This finding is also consistent with release of P from stream sediments. Low soil permeability is characteristic of fine sediments (Ren and Santamarina, 2018). If broader watershed soil types are indicative of in-stream sediment, very low permeability of fine-grained stream sediments could impede oxygen exchange to the hyporheic zone, potentially creating anoxic conditions to facilitate redox-mediated P release

(Mendoza-Lera and Datry, 2017).

Partial dependence plots also indicated that increased SRP during late summer low flow conditions was associated with increased clay content of soils in gaged watersheds. Clay particles are small in size, providing greater P adsorption potential (Simpson et al., 2021). Clay sediments also typically contain iron (Fe) that can bind P and can therefore provide a substrate for microbially mediated redox reactions (Pentrakova et al., 2013). Our findings are consistent

with a mechanism whereby clay sediments bind considerable P under some conditions, and then release it via redox reactions during late summer when oxic conditions are low due to microbial decomposition of organic matter.

Environmental conditions in large parts of our study region are consistent with those previously reported to foster situational SRP release from sediments. Previous studies have observed release of SRP from stream sediments when SRP to Fe ratios in sediments are high and when dissolved oxygen concentrations are low (Inamdar et al., 2020; van

Dael et al., 2020; Diamond et al., 2023). These conditions are characteristic of slow-moving lowland streams with large legacy P stores arising from current and historic P inputs, and may be especially common in headwater streams

---

[9] Note that stream temperature data in the U.S. EPA StreamCat dataset is derived from Hill et al. (2013) and takes into account natural factors and certain aspects of anthropogenic influence (i.e., reservoirs, urban land use and agricultural land use) but does not account for wastewater effluent.

(Diamond et al., 2023). Such conditions are widespread across our study region. Ditches, streams, and rivers in the flat to gently rolling landscapes of southern and northwestern Minnesota are characterized by relatively low gradients, high current and historic P loading from agriculture and urban land use (Boardman et al., 2019), and high rates of instream primary productivity (Dolph et al., 2017b). These conditions are likely to coincide during warm late summer conditions in high rates of microbial respiration, anoxic conditions, and P release.

Overall, our findings agree with previous studies that have identified the importance of biogeochemical processes in seasonally modulating nutrient concentrations during low flows in lowland lotic systems (e.g., Smolders et al., 2017) and in many ways parallels findings for eutrophic lakes (Søndergaard et al., 2001). Further study is needed to parse the importance of pore water, stream sediment dynamics, and mineralization to the elevated SRP concentrations we observed at various stream and river sites during late summer low flow conditions.

**Limitations and future study**

A major limitation of this study is that we did not have direct information about legacy P supply in stream and river channels. Efforts to quantify the P content of stream sediments, and the potential of stream sediments to adsorp/desorb SRP in different geographic and seasonal and hydrologic contexts, could provide additional valuable information about the extent to which in-channel release of legacy P is affecting downstream water quality.

It is also important to note that performance of the random forest model we used to predict late summer SRP concentrations was middling ($R^2=0.41$). We speculate that improved model performance will depend on reach-scale variables that may strongly determine SRP dynamics, such as channel morphology, characteristics and volume of bed sediment, and stream productivity and respiration. Future research could aim to incorporate both reach scale and broader scale variables into a more precise understanding of in channel SRP dynamics. For example, the sampling platform described by Felton et al. (2023) presents the intriguing possibility of monitoring stream conditions intensively along longitudinal gradients and could be refined to include measures of dissolved oxygen, $CO_2$ (as a proxy for respiration), temperature, sorption capacity, and/or to identify P inputs associated with tile and point discharges or certain aspects of channel morphology.

**5 Conclusions**

In this study we observed widespread elevation of SRP concentrations during late summer low flow conditions among anthropogenically dominated ditches, streams and rivers in Minnesota. These elevated SRP concentrations altered *C-Q* transport behavior for more than half (54%) of the gaged watersheds we studied, weakening what was otherwise more strongly mobilizing behavior during higher flow conditions and other times of year.

While wastewater discharge likely contributed to elevated SRP concentration at low flow for some sites, most sites exhibiting elevated SRP concentrations during late summer low flow conditions did not have substantial wastewater treatment impacts. Moreover, elevated SRP concentrations during low flow at these sites typically exceeded tile drainage SRP concentrations from corn and soy planted farm fields during late summer. We found that late summer low flow SRP concentrations were related to land use, soil characteristics, measures of agricultural intensity, and climate. Taken together, these findings suggest that climate and geologically mediated biogeochemical processes

likely result in the release of in-channel stores of legacy P during late summer low flow conditions in a substantial number of stream and river sites that have been heavily impacted by past and current P inputs associated with industrial/intensive agriculture and urbanization. As summers become hotter and drier – predicted climate changes in our region – conditions for the release of legacy P stored in stream and river channels will likely become more prolonged and/or more acute, contributing to the increased occurrence of adverse events such as harmful algal blooms. Further study is needed to determine the duration, fate and dominant mechanisms associated with riverine release of bioavailable P during late summer and other times of year.

Our findings suggest that efforts to reduce the impacts of bioavailable P to freshwaters will need to address both 1) mobilization of dissolved P from the landscape during high flow conditions and 2) in-channel environments that result in the release of accumulated legacy P from streams and rivers during summer low flows when freshwater systems are especially vulnerable to eutrophication outcomes. With regards to management, the association of land use in riparian areas with SRP during late summer low flows suggests that practices targeting near channel and riparian environments may be important in regulating elevated SRP. In Minnesota, state law implemented in 2015 has already mandated vegetative 50-foot riparian buffers along the state's water bodies to protect water quality (Riparian Protection and Water Quality Practices Act, 2015[10]). Our findings suggest that perhaps, specific vegetative conditions within these riparian buffers – for example whether mixed forest or woody wetlands are present – could additionally enhance legacy P retention and mitigate release. Further study is needed to examine the way these habitat types interact with legacy P over time and under different conditions.

Controlling ongoing P inputs will also be instrumental to reducing riverine P loading. For example, in Minnesota, additional phosphorus regulations added to National Pollutant Discharge Elimination System (NPDES) permits since the year 2000 have resulted in substantial reductions of P loading arising from wastewater facilities[11]. Policies and management approaches to substantially reduce inputs of fertilizer, manure and wastewater, as well their losses via surface, tile and other groundwater pathways, remain critical to achieving societal water quality goals.

**Code availability**
All R scripts used for data analysis are available in a permanent data repository at: https://doi.org/10.5281/zenodo.13936951

**Data availability**
All input data used in this paper are available in a permanent data repository at: https://doi.org/10.5281/zenodo.13936951

**Author contribution**
CD, BD, GF and JF conceived study design, identified available datasets, and identified research questions. CD performed data analysis and developed R scripts. CD prepared the manuscript with contributions from all co-authors.

**Competing interests**
The authors declare that they have no conflict of interest.

---

[10] https://www.revisor.mn.gov/statutes/cite/103F.48
[11] https://www.pca.state.mn.us/business-with-us/phosphorus-in-wastewater

**Acknowledgements**

This project was supported through the U.S. Department of Agriculture's Conservation Effects Assessment Project (CEAP) National Legacy Phosphorus Assessment, with funding provided by USDA Natural Resources Conservation Service (NRCS) through an Interagency Agreement (Agreement # NRC21IRA0010879). CEAP (https://www.nrcs.usda.gov/ceap) is a multi-agency effort led by the Natural Resources Conservation Service and the

Agricultural Research Service to quantify the effects of voluntary conservation and strengthen data-driven management decisions across the nation's private lands.

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

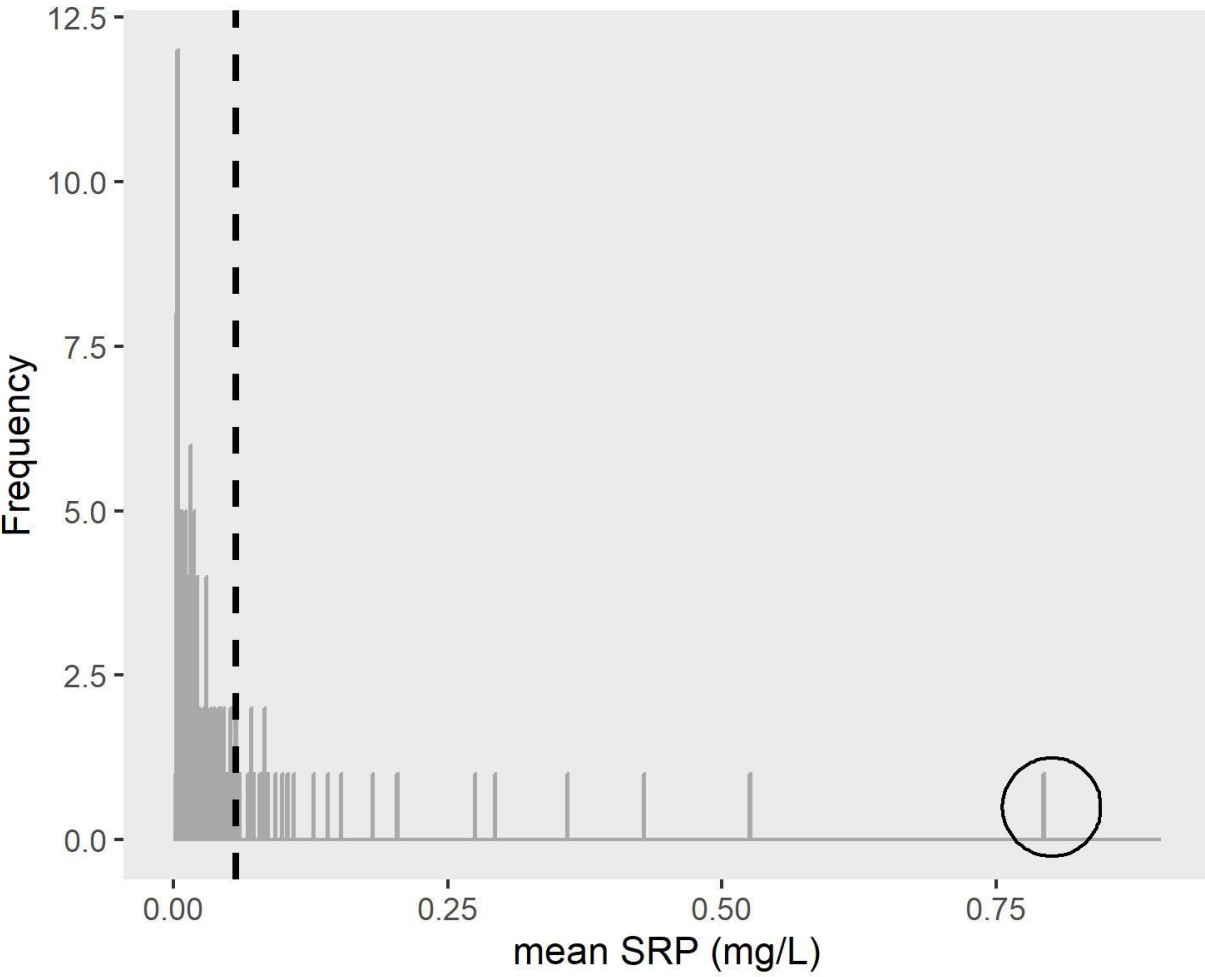

**Figure A1.** Distribution of mean SRP concentrations (mg/L) during late summer low flow conditions for 128 gaged watersheds with >=3 SRP samples collected during late summer low flows. Note that one outlier (circled value) was excluded prior to model development. The dashed line indicates the mean of all mean SRP values.

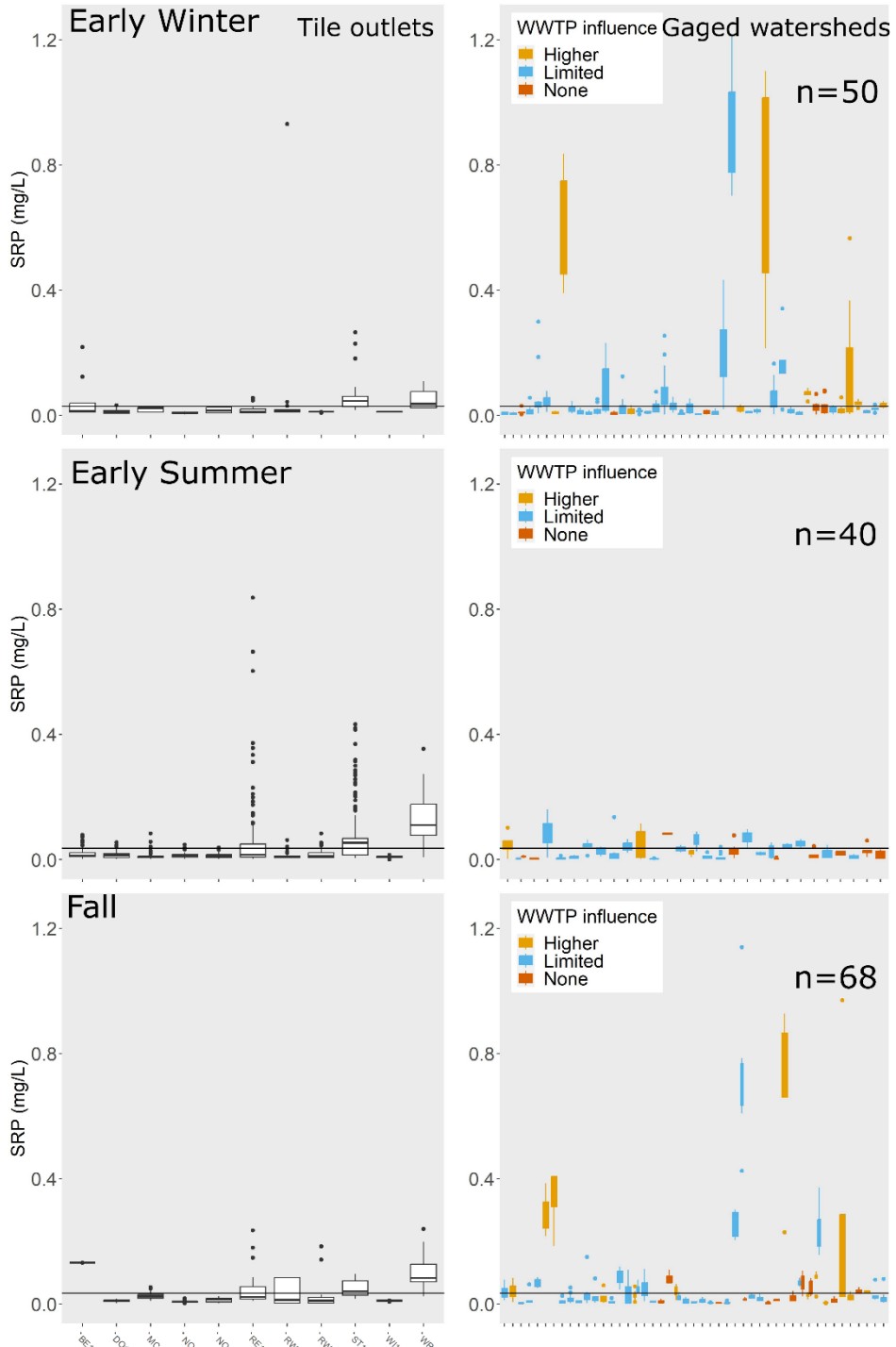

**Figure A2.** SRP concentrations (mg/L) for tile outlets and during low flow conditions for gaged watersheds, for early winter, early summer, and fall. See main text for similar plots for other seasons. The horizontal line in each plot is the mean SRP concentration among tile outlets for that season. The number of gages for which low flow data was available in each season is printed on the plot. For gaged watersheds, color of boxplots indicates degree of influence from wastewater treatment plants: light orange: wastewater treatment plant density >0.005 sites/km$^2$; blue = wastewater treatment plant density <0.005 sites/km$^2$ but greater than zero; dark orange: no wastewater treatment plant sites in watershed. To improve data visibility, the y-axis for SRP was limited to a maximum of 1.25 mg/L, which eliminated a small number of outliers from the plots for tile outlets (n=34 out of 11,079 records) and gaged watersheds (n=16 out of 2,696 low flow records).

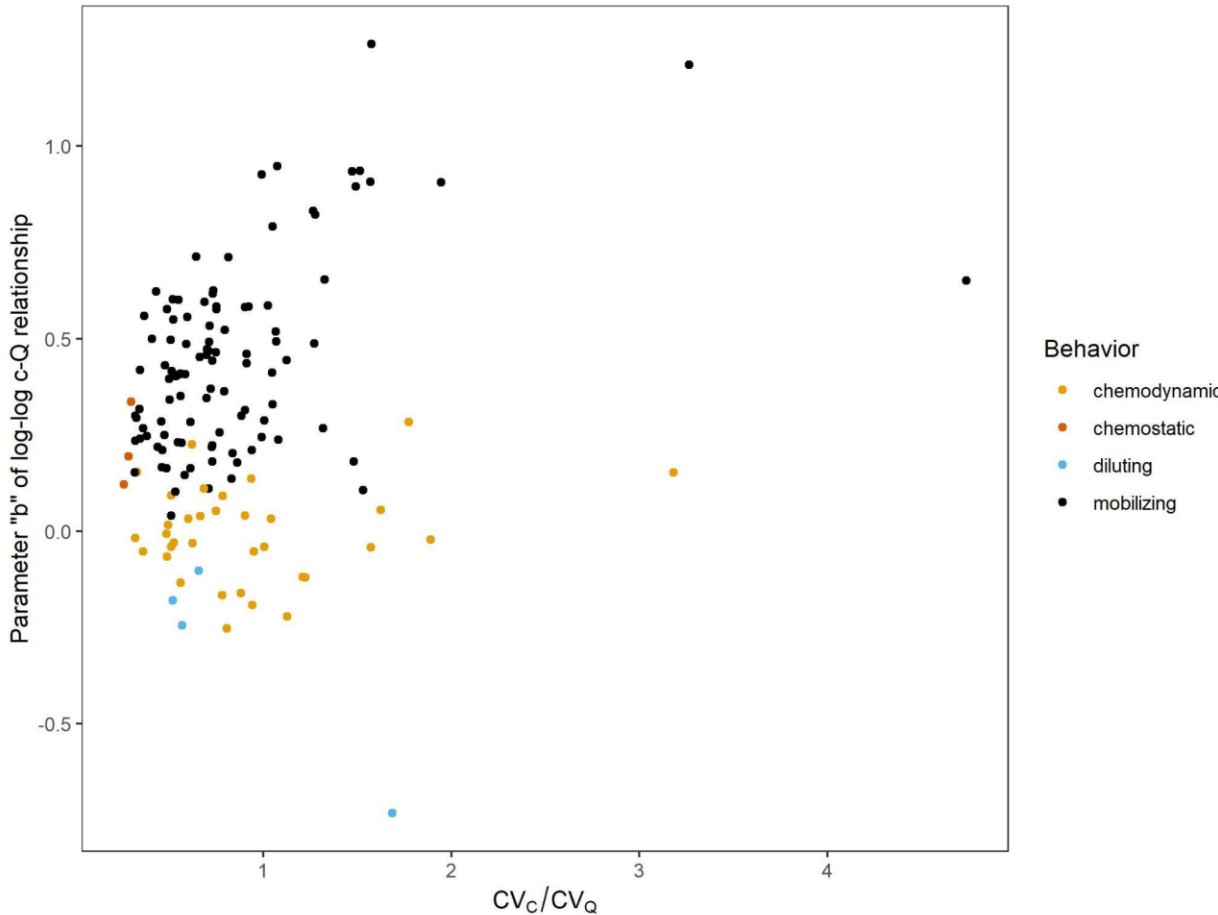

**Figure A3.** Parameter b (slope) of the event-scale log-log C-Q relationship for soluble reactive phosphorus (SRP, mg/L) in relation to CVC/CVQ for 143 gaged watersheds. Color indicates export behavior based on criteria defined for b and CVC/CVQ: Chemostatic: CVC/CVQ <= 0.3 (*sensu* Thompson et al., 2011); chemodynamic: CVC/CVQ > 0.3 and no significant b (p>0.05); diluting: significant b <0 ; mobilizing: significant b > 0.

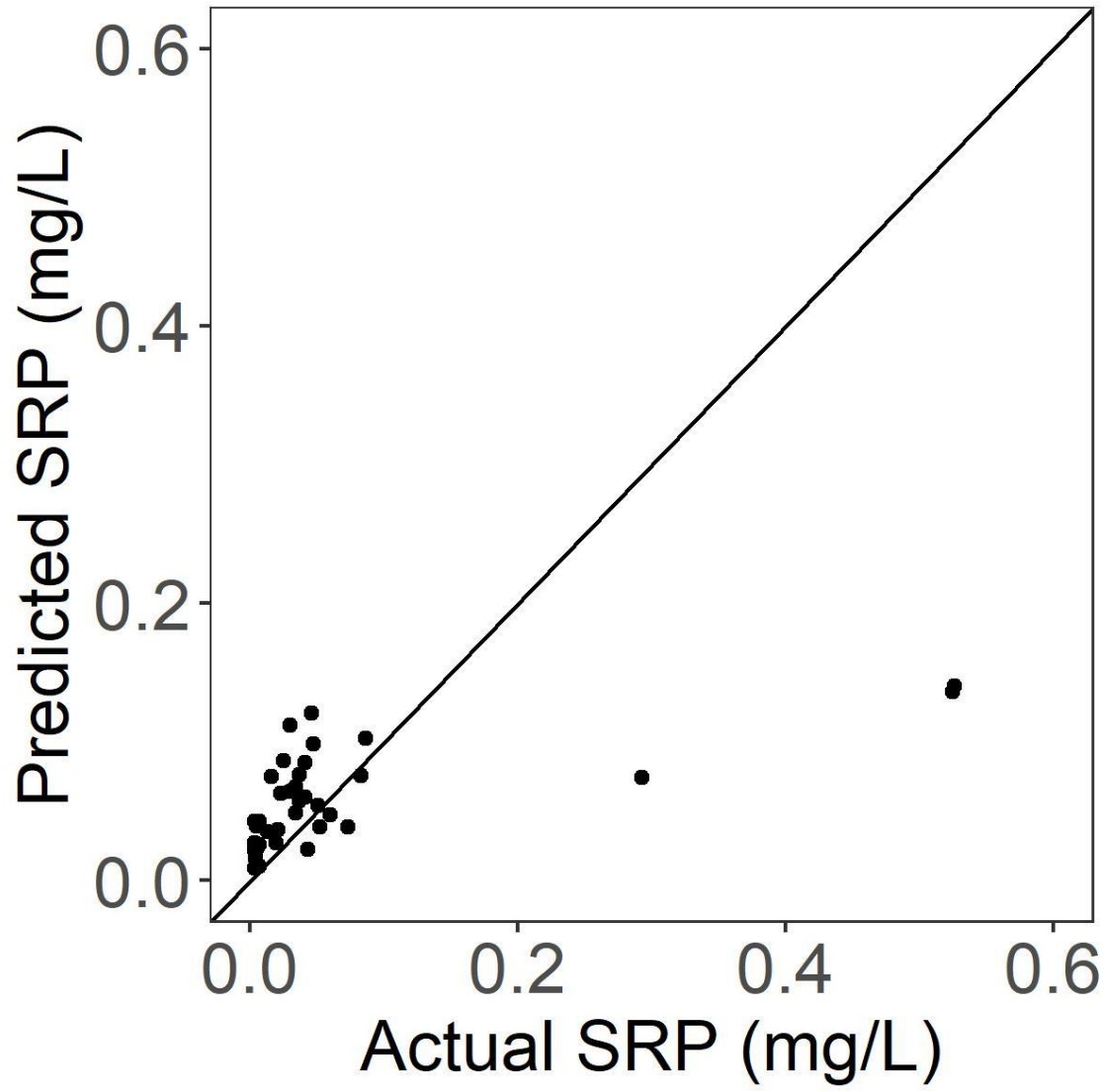

**Figure A4.** Actual vs predicted late summer SRP concentrations (mg/L) for gaged streams and rivers in the independent test dataset (i.e., not used to build the model). $R^2$=0.41, RMSE=0.010, p <0.0001. Solid line shows 1-1 relationship.

**Table A1.** Linear regression statistics for mean SRP (log scale) during low flow conditions in relation to the density of wastewater treatment plants (sites/km$^2$) across gaged watersheds, by season. Linear regression were calculated with all sites, and recalculated where sites with density of wastewater treatment plants >0.005 sites/km$^2$ were excluded.

| Season | Slope | T statistic | p | R2 | n | WWTP influence |
|---|---|---|---|---|---|---|
| Early Winter | 102.77 | 4.06 | <0.001 | 0.26 | 50 | all watersheds |
| Early Winter | 87.12 | 1.61 | 0.12 | 0.06 | 40 | gages with WWTP >0.005 sites/km2 excluded |
| Late Winter | 90.57 | 3.62 | <0.001 | 0.19 | 57 | all watersheds |
| Late Winter | 121.36 | 2.14 | 0.04 | 0.1 | 45 | gages with WWTP >0.005 sites/km2 excluded |
| Spring | 29.47 | 0.7 | 0.49 | 0.02 | 23 | all watersheds |
| Spring | -55.71 | -0.7 | 0.5 | 0.03 | 17 | gages with WWTP >0.005 sites/km2 excluded |
| Early Summer | 51.93 | 1.4 | 0.17 | 0.05 | 40 | all watersheds |
| Early Summer | 47.84 | 0.91 | 0.37 | 0.02 | 35 | gages with WWTP >0.005 sites/km2 excluded |
| Late Summer | 74.76 | 4.84 | <0.001 | 0.16 | 128 | all watersheds |
| Late Summer | 80.66 | 2.44 | 0.02 | 0.06 | 102 | gages with WWTP >0.005 sites/km2 excluded |
| Fall | 76.03 | 4.07 | <0.001 | 0.2 | 68 | all watersheds |
| Fall | 95.19 | 1.85 | 0.07 | 0.07 | 51 | gages with WWTP >0.005 sites/km2 excluded |

**Table A2.** Linear regression statistics for mean SRP (log scale) during low flow conditions in relation to % crop cover across gaged watersheds, by season. Linear regression were calculated with all sites, and recalculated for sites with no wastewater treatment plant influence in their watersheds.

| Season | Slope | T statistic | p | R2 | n | WWTP influence |
|--------|-------|-------------|------|------|-----|----------------|
| Early Winter | 0.01 | 5.09 | <0.001 | 0.35 | 50 | all watersheds |
| Early Winter | 0.01 | 4.25 | 0.02 | 0.86 | 5 | no WWTP present in watershed |
| Late Winter | 0.02 | 7.65 | <0.001 | 0.52 | 57 | all watersheds |
| Late Winter | 0.01 | 2.5 | 0.04 | 0.44 | 10 | no WWTP present in watershed |
| Spring | 0.01 | 3.75 | <0.001 | 0.4 | 23 | all watersheds |
| Spring | 0.01 | 2.27 | 0.06 | 0.43 | 9 | no WWTP present in watershed |
| Early Summer | 0.01 | 3.56 | <0.001 | 0.25 | 40 | all watersheds |
| Early Summer | 0.01 | 3.32 | 0.01 | 0.55 | 11 | no WWTP present in watershed |
| Late Summer | 0.01 | 10.66 | <0.001 | 0.47 | 128 | all watersheds |
| Late Summer | 0.01 | 7.39 | <0.001 | 0.69 | 26 | no WWTP present in watershed |
| Fall | 0.01 | 5.63 | <0.001 | 0.32 | 68 | all watersheds |
| Fall | 0.01 | 3.09 | 0.01 | 0.51 | 11 | no WWTP present in watershed |

**Table A3.** Mean SRP during low flow conditions (flow conditions <= lowest 25th percentile of flows on record) for each gaged watershed during each season. Note means were only calculated where gaged watersheds had >= three low flow samples collected during a season. 'NA' indicates < three low flow samples were collected during that season. 'WS Area' refers to the watershed land area draining into each gage in km$^2$. 'WWTP Density' is wastewater treatment plant density for each watershed in sites/km$^2$.

| Site name | WS Area (km$^2$) | Human Impact | WWTP Density | Spring | Early Summer | Late Summer | Fall | Early Winter | Late Winter |
|---|---|---|---|---|---|---|---|---|---|
| Brule River nr Hovland, MN61 | 544.30 | Less Impacted | 0 | NA | NA | NA | NA | NA | NA |
| Poplar River nr Lutsen, 0.2mi US of MN61 | 293.29 | Less Impacted | 0.003 | NA | NA | 0.003 | 0.003 | 0.004 | 0.003 |
| Baptism River nr Beaver Bay, MN61 | 358.44 | Less Impacted | 0.003 | NA | NA | 0.004 | NA | 0.006 | 0.003 |
| Beaver River nr Beaver Bay, 1.2mi us of MN61 | 318.73 | Impacted | 0.003 | NA | NA | 0.003 | NA | NA | NA |
| Big Sucker Creek nr Palmers, CR258 | 93.48 | Less Impacted | 0 | NA | 0.003 | 0.003 | NA | 0.011 | 0.009 |
| St. Louis River at Floodwood, CSAH8 | 4978.62 | Impacted | 0.004 | 0.002 | NA | 0.011 | NA | NA | NA |
| St. Louis River nr Forbes, US53 | 1793.51 | Impacted | 0.006 | 0.003 | NA | 0.004 | 0.003 | NA | NA |
| Second Creek nr Aurora, 0.6mi us of CSAH110 | 56.66 | Impacted | 0 | NA | NA | 0.004 | 0.002 | NA | 0.006 |
| St. Louis River at Scanlon, MN | 8893.21 | Impacted | 0.003 | NA | 0.009 | 0.004 | NA | 0.007 | NA |
| Cloquet River nr Brimson, CSAH44 | 330.99 | Less Impacted | 0 | NA | NA | 0.004 | NA | NA | NA |
| Cloquet River nr Burnett, CR694 | 2032.29 | Less Impacted | 0.001 | 0.005 | 0.005 | 0.003 | NA | 0.008 | NA |
| Nemadji River nr Pleasant Valley, MN23 | 324.41 | Impacted | 0 | NA | NA | 0.005 | NA | NA | NA |
| Mississippi River nr Bemidji, CSAH11 | 958.60 | Impacted | 0.001 | NA | 0.028 | 0.03 | NA | NA | NA |
| Mississippi River nr Bemidji, MN | 1627.93 | Impacted | 0.002 | NA | 0.003 | 0.003 | NA | NA | NA |
| Leech Lake River nr Ball Club, CR139 | 3464.53 | Impacted | 0.001 | 0.004 | 0.003 | 0.004 | NA | NA | NA |
| Boy River nr Boy River, CSAH53 | 750.13 | Less Impacted | 0.001 | NA | NA | 0.002 | NA | NA | NA |
| Prairie River nr Taconite, MN | 79.80 | Impacted | 0 | 0.006 | NA | 0.005 | NA | NA | 0.006 |

| | | | | | | | | | |
|---|---|---|---|---|---|---|---|---|---|
| Mississippi River at Grand Rapids, MN | 8483.32 | Impacted | 0.001 | 0.003 | 0.003 | 0.005 | 0.007 | NA | 0.003 |
| Swan River nr Jacobson, CR438 | 758.11 | Impacted | 0.012 | NA | NA | 0.018 | NA | NA | NA |
| Swan River nr Sobieski, MN238 | 462.60 | Impacted | 0.002 | NA | NA | 0.021 | NA | NA | NA |
| Pine River nr Jenkins, CSAH15 | 743.11 | Impacted | 0 | NA | NA | NA | NA | NA | NA |
| Leaf River nr Staples, CSAH29 | 2232.34 | Impacted | 0.003 | NA | NA | 0.022 | NA | 0.013 | 0.025 |
| Long Prairie River at Philbrook, 313th Ave | 2346.60 | Impacted | 0.005 | NA | 0.035 | 0.023 | NA | 0.024 | 0.022 |
| Long Prairie River at Long Prairie, MN | 1271.78 | Impacted | 0.005 | 0.014 | NA | NA | NA | NA | 0.016 |
| Two Rivers nr Bowlus, 40th St | 386.26 | Impacted | 0.008 | NA | NA | NA | NA | NA | NA |
| Sauk River nr St. Martin, CR12 | 2887.54 | Impacted | 0.005 | 0.018 | NA | 0.059 | NA | NA | NA |
| Clearwater River nr Clearwater, CR145 | 436.24 | Impacted | 0.002 | 0.008 | NA | 0.003 | NA | NA | NA |
| Elk River nr Big Lake, MN | 1373.83 | Impacted | 0.003 | NA | 0.027 | NA | NA | NA | NA |
| St. Francis River nr Big Lake, 164th St | 530.84 | Impacted | 0 | NA | 0.02 | 0.013 | NA | NA | NA |
| Middle Fork Crow River nr Manannah, CSAH30 | 702.11 | Impacted | 0.006 | 0.01 | 0.022 | 0.012 | NA | NA | NA |
| North Fork Crow River nr Cokato, CSAH4 | 2638.30 | Impacted | 0.004 | NA | 0.07 | 0.031 | NA | NA | NA |
| North Fork Crow River nr Rockford, Farmington Ave | 3444.89 | Impacted | 0.005 | NA | NA | 0.02 | 0.021 | 0.031 | 0.045 |
| South Fork Crow River at Delano, Bridge Ave | 3295.67 | Impacted | 0.004 | NA | NA | 0.182 | 0.239 | 0.168 | 0.218 |
| Buffalo Creek nr Glencoe, CSAH1 | 969.28 | Impacted | 0.004 | NA | NA | 0.793 | NA | NA | NA |
| West Branch Rum River nr Princeton, CR102 | 482.91 | Impacted | 0.006 | NA | NA | NA | NA | NA | NA |
| Rum River at Anoka, headwater side of dam | 4113.49 | Impacted | 0.004 | NA | 0.043 | NA | NA | 0.01 | 0.015 |

| | | | | | | | | | |
|---|---|---|---|---|---|---|---|---|---|
| Minnesota River nr Lac qui Parle, MN | 10546.93 | Impacted | 0.003 | NA | NA | 0.085 | 0.055 | 0.094 | 0.126 |
| Yellow Bank River nr Odessa, CSAH40 | 1191.21 | Impacted | 0.004 | NA | NA | 0.03 | NA | 0.021 | 0.31 |
| Pomme De Terre River at Appleton, MN | 2227.38 | Impacted | 0.002 | NA | NA | NA | NA | 0.024 | 0.082 |
| Pomme de Terre River nr Hoffman, CR76 | 1026.90 | Impacted | 0.002 | NA | NA | 0.013 | 0.01 | NA | NA |
| Lac qui Parle River nr Lac qui Parle, CSAH31 | 2487.56 | Impacted | 0.004 | NA | 0.042 | 0.019 | NA | 0.028 | 0.063 |
| Lac qui Parle River nr Providence, CSAH23 | 983.91 | Impacted | 0.004 | NA | NA | 0.023 | NA | NA | NA |
| West Branch Lac qui Parle River at Dawson, Diagonal St | 1220.43 | Impacted | 0.002 | NA | NA | 0.083 | NA | NA | NA |
| Hawk Creek nr Maynard, MN23 | 1226.46 | Impacted | 0.003 | NA | 0.056 | 0.051 | 0.034 | NA | NA |
| Hawk Creek nr Granite Falls, CR52 | 1321.42 | Impacted | 0.003 | NA | 0.003 | 0.01 | 0.013 | 0.083 | 0.158 |
| Beaver Creek nr Beaver Falls, CSAH2 | 495.39 | Impacted | 0.004 | NA | NA | 0.053 | 0.041 | NA | NA |
| Yellow Medicine River nr Granite Falls, MN | 1716.39 | Impacted | 0.002 | NA | NA | 0.034 | 0.012 | 0.02 | 0.35 |
| Yellow Medicine River nr Hanley Falls, CR18 | 1171.62 | Impacted | 0.001 | NA | NA | 0.019 | 0.027 | NA | NA |
| Spring Creek nr Hanley Falls, 480th St | 332.81 | Impacted | 0.006 | NA | NA | 0.293 | NA | NA | NA |
| Chippewa River nr Clontarf, CSAH22 | 1731.39 | Impacted | 0.004 | NA | NA | 0.016 | 0.009 | NA | NA |
| Shakopee Creek nr Benson, 20th Ave SW | 786.32 | Impacted | 0 | 0.006 | NA | 0.141 | NA | NA | 0.024 |
| Chippewa River nr Milan, MN40 | 4626.36 | Impacted | 0.002 | NA | NA | 0.022 | 0.009 | 0.02 | 0.027 |
| East Branch Chippewa River nr Benson, CR78 | 1305.27 | Impacted | 0.001 | NA | 0.043 | 0.048 | 0.019 | NA | NA |

| | | | | | | | | | |
|---|---|---|---|---|---|---|---|---|---|
| Redwood River nr Redwood Falls, MN | 1619.96 | Impacted | 0.003 | NA | NA | 0.099 | 0.731 | 1.175 | 1.247 |
| Redwood River at Russell, CR15 | 599.04 | Impacted | 0 | NA | NA | 0.035 | 0.015 | NA | NA |
| Minnesota River at Morton, MN | 23008.88 | Impacted | 0.003 | NA | NA | 0.046 | 0.058 | 0.098 | 0.141 |
| Minnesota River at Judson, CSAH42 | 28865.06 | Impacted | 0.003 | NA | 0.01 | 0.027 | 0.028 | 0.04 | 0.08 |
| Seven Mile Creek nr St. Peter, 0.6mi us of US169 | 94.16 | Impacted | 0 | 0.008 | 0.015 | 0.016 | 0.012 | NA | 0.027 |
| Cottonwood River nr New Ulm, MN68 | 3374.02 | Impacted | 0.003 | 0.01 | 0.007 | 0.007 | 0.007 | 0.016 | 0.048 |
| Sleepy Eye Creek nr Cobden, CR8 | 706.81 | Impacted | 0 | NA | NA | 0.022 | 0.021 | NA | NA |
| Cottonwood River nr Leavenworth, CR8 | 2281.68 | Impacted | 0.004 | NA | NA | 0.013 | NA | NA | NA |
| Blue Earth River nr Winnebago, CSAH12 | 3592.84 | Impacted | 0.004 | NA | NA | 0.027 | 0.025 | NA | NA |
| East Branch Blue Earth River at Blue Earth, CSAH16 | 764.22 | Impacted | 0.003 | NA | NA | 0.071 | 0.062 | NA | NA |
| Blue Earth River nr Rapidan, MN | 6274.80 | Impacted | 0.004 | NA | NA | 0.015 | 0.006 | 0.015 | 0.028 |
| South Fork Watonwan River nr Madelia, CSAH13 | 515.74 | Impacted | 0.004 | NA | NA | 0.052 | NA | NA | NA |
| Watonwan River nr La Salle, CSAH3 | 875.15 | Impacted | 0.005 | NA | NA | 0.073 | NA | NA | NA |
| Watonwan River nr Garden City, CSAH13 | 2205.66 | Impacted | 0.005 | NA | NA | 0.016 | 0.022 | 0.017 | 0.152 |
| Maple River nr Sterling Center, CR18 | 801.21 | Impacted | 0.004 | NA | NA | 0.071 | NA | NA | NA |
| Big Cobb River nr Beauford, CSAH16 | 787.09 | Impacted | 0.006 | NA | 0.048 | 0.041 | 0.042 | NA | NA |
| Maple River nr Rapidan, CR35 | 877.79 | Impacted | 0.005 | NA | NA | 0.02 | 0.038 | NA | NA |

| | | | | | | | | | |
|---|---|---|---|---|---|---|---|---|---|
| Little Beauford Ditch nr Beauford, MN22 | 22.03 | Impacted | 0 | 0.025 | 0.083 | 0.153 | NA | NA | NA |
| Le Sueur River nr Rapidan, CR8 | 1156.71 | Impacted | 0.005 | 0.004 | 0.017 | 0.029 | 0.027 | NA | NA |
| Le Sueur River nr Rapidan, MN | 2873.06 | Impacted | 0.005 | NA | 0.006 | 0.014 | 0.012 | 0.014 | 0.032 |
| Le Sueur River at St. Clair, CSAH28 | 919.12 | Impacted | 0.006 | 0.023 | 0.101 | 0.058 | NA | NA | NA |
| High Island Creek nr Arlington, CR9 | 428.96 | Impacted | 0 | 0.037 | NA | 0.079 | NA | NA | NA |
| High Island Creek nr Henderson, CSAH6 | 624.30 | Impacted | 0.002 | NA | NA | 0.019 | NA | NA | NA |
| Snake River nr Pine City, MN | 2563.68 | Impacted | 0.003 | NA | 0.02 | 0.012 | NA | 0.014 | 0.02 |
| Sunrise River at Sunrise, CR88 | 870.50 | Impacted | 0.005 | NA | NA | 0.037 | NA | NA | NA |
| Wells Creek nr Frontenac, US61 | 176.12 | Impacted | 0 | 0.032 | NA | NA | 0.04 | NA | NA |
| Cannon River at Welch, MN | 3477.94 | Impacted | 0.007 | NA | NA | 0.204 | NA | 0.009 | 0.044 |
| Cannon River at Morristown, CSAH16 | 627.90 | Impacted | 0.008 | 0.121 | NA | NA | 0.29 | NA | NA |
| Straight River nr Faribault, MN | 1129.24 | Impacted | 0.006 | NA | NA | NA | NA | 0.069 | NA |
| Whitewater River nr Beaver, CSAH30 | 689.62 | Impacted | 0.01 | NA | NA | 0.047 | 0.039 | 0.039 | 0.048 |
| North Fork Whitewater River at Elba, Whitewater Dr | 269.37 | Impacted | 0.011 | NA | NA | NA | 0.057 | NA | 0.049 |
| North Fork Zumbro River nr Mazeppa, CSAH7 | 621.81 | Impacted | 0.006 | NA | NA | 0.057 | 0.026 | NA | 0.046 |
| Zumbro River at Kellogg, US61 | 3677.89 | Impacted | 0.008 | NA | NA | 0.046 | NA | 0.033 | 0.042 |
| South Fork Zumbro River nr Oronoco, CR121 | 885.55 | Impacted | 0.015 | NA | NA | 0.11 | 0.077 | NA | NA |
| South Branch Middle Fork Zumbro River nr Oronoco,5th St | 569.86 | Impacted | 0.011 | NA | NA | 0.057 | 0.025 | NA | NA |
| Middle Fork Zumbro River nr Oronoco,5th St | 534.52 | Impacted | 0.006 | NA | NA | 0.041 | 0.02 | NA | NA |
| South Branch Root River at | 738.36 | Impacted | 0.01 | NA | NA | 0.03 | 0.032 | NA | NA |

| | | | | | | | | | |
|---|---|---|---|---|---|---|---|---|---|
| Lanesboro, Rochelle Ave N | | | | | | | | | |
| Middle Branch Root River nr Fillmore, CSAH5 | 506.94 | Impacted | 0.006 | NA | NA | 0.039 | 0.035 | NA | NA |
| Cedar River nr Austin, MN | 1030.86 | Impacted | 0.013 | NA | NA | 0.525 | 0.338 | 0.65 | 0.954 |
| Turtle Creek at Austin, 43rd St | 382.45 | Impacted | 0.008 | NA | NA | 0.359 | 0.268 | NA | NA |
| Shell Rock River nr Gordonsville, CSAH1 | 496.65 | Impacted | 0.012 | NA | NA | 0.526 | 0.68 | 1.159 | 1.368 |
| West Fork Des Moines River nr Avoca, CSAH6 | 1233.33 | Impacted | 0.006 | NA | NA | 0.014 | NA | NA | NA |
| West Fork Des Moines River at Jackson, River St | 3216.99 | Impacted | 0.007 | NA | NA | 0.037 | NA | 0.342 | 0.592 |
| Bois de Sioux River nr Doran, MN | 4713.44 | Impacted | 0.002 | NA | 0.084 | 0.086 | NA | 0.069 | 0.128 |
| Mustinka River nr Norcross, MN9 | 486.58 | Impacted | 0 | NA | NA | 0.083 | 0.082 | NA | NA |
| Twelvemile Creek nr Wheaton, CSAH14 | 1006.00 | Impacted | 0 | 0.141 | NA | 0.429 | NA | NA | NA |
| Pelican River nr Fergus Falls, MN210 | 1294.46 | Impacted | 0.003 | NA | 0.02 | 0.019 | 0.006 | NA | NA |
| Otter Tail River nr Elizabeth, MN | 2974.42 | Impacted | 0.001 | NA | NA | 0.003 | 0.002 | NA | 0.005 |
| Otter Tail River at Breckenridge, CSAH16 | 5116.19 | Impacted | 0.002 | NA | NA | 0.016 | 0.011 | 0.005 | 0.01 |
| Red River of the North nr Kragnes, CSAH26 | 28555.84 | Impacted | 0.003 | NA | NA | 0.275 | 0.254 | 0.217 | 0.332 |
| Buffalo River nr Georgetown, CR108 | 2820.97 | Impacted | 0.002 | NA | NA | 0.093 | 0.067 | 0.042 | 0.052 |
| Buffalo River nr Glyndon, CSAH19 | 998.93 | Impacted | 0.004 | NA | NA | 0.068 | NA | NA | NA |
| South Branch Buffalo River nr Glyndon, 28th Ave S | 1314.84 | Impacted | 0.001 | NA | NA | 0.128 | NA | NA | NA |
| Buffalo River nr Hawley, MN | 837.03 | Impacted | 0.001 | NA | NA | 0.06 | NA | NA | NA |
| Wild Rice River nr Mahnomen, CSAH25 | 1473.79 | Impacted | 0 | NA | NA | 0.019 | NA | NA | NA |

| Location | | | | | | | | | |
| --- | --- | --- | --- | --- | --- | --- | --- | --- | --- |
| Wild Rice River at Twin Valley, MN | 2405.45 | Impacted | 0.001 | NA | NA | 0.007 | NA | NA | NA |
| Wild Rice River at Hendrum, MN | 4091.49 | Impacted | 0.002 | NA | NA | 0.03 | 0.031 | 0.012 | 0.018 |
| S Br. Wild Rice River at CR27 nr Felton, MN | 499.54 | Impacted | 0.006 | NA | NA | 0.055 | NA | NA | NA |
| Sandhill River nr Fertile, 450th St SW | 620.95 | Impacted | 0.002 | NA | 0.052 | 0.029 | 0.018 | NA | NA |
| Sand Hill River at Climax, MN, US-75 | 1209.33 | Impacted | 0.001 | NA | NA | 0.034 | 0.024 | 0.014 | 0.031 |
| Red Lake River at High Landing nr Goodridge, MN | 5963.65 | Impacted | 0 | NA | 0.003 | 0.011 | NA | NA | NA |
| Red Lake River at Red Lake Falls, CR13 | 9413.51 | Impacted | 0 | 0.003 | 0.023 | 0.004 | 0.004 | NA | NA |
| Red Lake River at Fisher, MN | 14466.12 | Impacted | 0.001 | NA | 0.034 | 0.013 | NA | 0.009 | 0.017 |
| Thief River nr Thief River Falls, MN | 2515.45 | Impacted | 0 | NA | 0.027 | 0.011 | 0.012 | 0.026 | 0.05 |
| Thief River downstream of CSAH 7, 6 mi E of Holt | 1658.23 | Impacted | 0 | 0.002 | 0.013 | 0.007 | NA | NA | 0.036 |
| Mud River nr Grygla, MN89 | 481.28 | Impacted | 0 | NA | 0.029 | 0.021 | 0.008 | NA | NA |
| Clearwater River at Plummer, MN | 1454.55 | Impacted | 0.002 | NA | NA | 0.043 | 0.027 | NA | NA |
| Lost River nr Brooks, CR119 | 772.13 | Impacted | 0.003 | NA | NA | 0.042 | 0.085 | NA | NA |
| Clearwater River at Red Lake Falls, MN | 3474.65 | Impacted | 0.002 | NA | NA | 0.009 | 0.009 | 0.01 | 0.04 |
| Snake River nr Big Woods, MN220 | 2006.98 | Impacted | 0.001 | NA | NA | NA | 0.068 | 0.074 | 0.191 |
| Middle River at Argyle, MN | 643.64 | Impacted | 0.002 | NA | 0.062 | 0.043 | 0.048 | NA | NA |
| Snake River above Warren, MN | 491.56 | Impacted | 0 | NA | NA | NA | 0.068 | NA | NA |
| Tamarac River nr Florian, CSAH1 | 464.17 | Impacted | 0 | NA | 0.022 | 0.011 | NA | NA | NA |
| Tamarac River nr Stephen, CSAH22 | 909.20 | Impacted | 0 | NA | 0.036 | NA | NA | 0.03 | 0.198 |
| Two Rivers nr Hallock, CSAH16 | 2660.25 | Impacted | 0 | NA | NA | 0.016 | NA | 0.018 | 0.03 |

| | | | | | | | | | |
|---|---|---|---|---|---|---|---|---|---|
| South Branch Two Rivers at Hallock, MN175 | 1669.83 | Impacted | 0.001 | NA | NA | 0.052 | NA | NA | NA |
| South Branch Two Rivers at Lake Bronson, MN | 1380.24 | Impacted | 0 | NA | NA | 0.104 | 0.053 | NA | NA |
| Kawishiwi River nr Winton, CSAH18 | 3415.83 | Less Impacted | 0 | 0.004 | NA | 0.004 | NA | 0.007 | 0.007 |
| Stony River nr Babbitt, Tomahawk Rd | 550.10 | Less Impacted | 0 | NA | NA | 0.003 | NA | NA | NA |
| Little Fork River nr Linden Grove, TH73 | 736.54 | Impacted | 0.003 | NA | NA | 0.02 | 0.018 | NA | NA |
| Little Fork River at Little Fork, MN | 4382.42 | Less Impacted | 0 | NA | NA | 0.004 | 0.004 | 0.009 | 0.015 |
| Little Fork River nr Littlefork, MN65 | 3492.09 | Less Impacted | 0.001 | NA | NA | 0.007 | 0.005 | NA | NA |
| Big Fork River nr Bigfork, MN6 | 1567.06 | Less Impacted | 0 | NA | 0.01 | 0.007 | NA | NA | NA |
| Big Fork River at Big Falls, MN | 3888.36 | Less Impacted | 0.001 | NA | 0.003 | 0.004 | 0.003 | 0.005 | 0.011 |
| Big Fork River nr Craigville, MN6 | 2555.10 | Less Impacted | 0.001 | NA | NA | 0.004 | NA | NA | NA |
| East Fork Rapid River nr Clementson, CSAH18 | 709.40 | Less Impacted | 0 | NA | NA | 0.022 | NA | NA | NA |
| Rapid River at Clementson, MN11 | 2486.55 | Impacted | 0 | NA | NA | 0.004 | 0.003 | 0.01 | 0.014 |
| Split Rock Creek nr Jasper, 201st St | 822.62 | Impacted | 0.004 | NA | NA | 0.011 | NA | 0.014 | 0.027 |
| Pipestone Creek nr Pipestone, CSAH13 | 308.55 | Impacted | 0.006 | NA | NA | 0.025 | NA | NA | NA |
| Rock River at Luverne, CR4 | 1084.55 | Impacted | 0.005 | NA | NA | 0.016 | NA | 0.02 | 0.025 |

**Table A4.** Number of SRP samples collected during low flow conditions in each season for each gaged watershed. Note that mean SRP values (Table S3) were only calculated in seasons where >=3 samples had been collected during low flow conditions for that gaged watershed.

| Site name | Early Winter | Late Winter | Spring | Early Summer | Late Summer | Fall |
|---|---|---|---|---|---|---|
| Baptism River nr Beaver Bay, MN61 | 4 | 5 | NA | 2 | 9 | NA |
| Beaver Creek nr Beaver Falls, CSAH2 | NA | 1 | 1 | 1 | 6 | 5 |
| Beaver River nr Beaver Bay, 1.2mi us of MN61 | NA | NA | NA | 2 | 4 | 1 |
| Big Cobb River nr Beauford, CSAH16 | NA | 2 | 2 | 4 | 18 | 4 |
| Big Fork River at Big Falls, MN | 6 | 10 | 1 | 4 | 14 | 3 |
| Big Fork River nr Bigfork, MN6 | NA | NA | NA | 4 | 14 | NA |
| Big Fork River nr Craigville, MN6 | NA | NA | NA | 2 | 10 | 1 |
| Big Sucker Creek nr Palmers, CR258 | 6 | 4 | NA | 3 | 4 | NA |
| Blue Earth River nr Rapidan, MN | 11 | 7 | 2 | 2 | 19 | 3 |
| Blue Earth River nr Winnebago, CSAH12 | NA | 1 | NA | NA | 8 | 3 |
| Bois de Sioux River nr Doran, MN | 10 | 5 | 2 | 6 | 12 | 2 |
| Boy River nr Boy River, CSAH53 | NA | NA | 1 | 2 | 5 | 2 |
| Brule River nr Hovland, MN61 | NA | NA | NA | NA | 1 | 2 |
| Buffalo Creek nr Glencoe, CSAH1 | NA | 1 | 2 | 1 | 3 | 1 |
| Buffalo River nr Georgetown, CR108 | 8 | 7 | NA | NA | 11 | 4 |
| Buffalo River nr Glyndon, CSAH19 | NA | NA | NA | NA | 3 | 2 |
| Buffalo River nr Hawley, MN | NA | 2 | 1 | NA | 4 | 2 |
| Cannon River at Morristown, CSAH16 | NA | 1 | 3 | NA | 2 | 3 |
| Cannon River at Welch, MN | 8 | 5 | 1 | NA | 6 | 1 |
| Cedar River nr Austin, MN | 10 | 4 | NA | 1 | 9 | 4 |
| Chippewa River nr Clontarf, CSAH22 | NA | 2 | 1 | 2 | 5 | 4 |
| Chippewa River nr Milan, MN40 | 11 | 8 | NA | NA | 9 | 4 |
| Clearwater River at Plummer, MN | NA | NA | 2 | NA | 5 | 6 |

| | | | | | |
|---|---|---|---|---|---|
| Clearwater River at Red Lake Falls, MN | 14 | 9 | 2 | 1 | 16 | 9 |
| Clearwater River nr Clearwater, CR145 | NA | 2 | 3 | 1 | 3 | 1 |
| Cloquet River nr Brimson, CSAH44 | NA | NA | 1 | 1 | 7 | NA |
| Cloquet River nr Burnett, CR694 | 3 | 2 | 4 | 4 | 13 | 2 |
| Cottonwood River nr Leavenworth, CR8 | NA | NA | 2 | 1 | 6 | 2 |
| Cottonwood River nr New Ulm, MN68 | 14 | 8 | 3 | 3 | 14 | 6 |
| East Branch Blue Earth River at Blue Earth, CSAH16 | NA | 1 | NA | 1 | 8 | 3 |
| East Branch Chippewa River nr Benson, CR78 | NA | NA | NA | 6 | 6 | 3 |
| East Fork Rapid River nr Clementson, CSAH18 | NA | NA | 1 | 2 | 8 | NA |
| Elk River nr Big Lake, MN | NA | 1 | 2 | 5 | 2 | 1 |
| Hawk Creek nr Granite Falls, CR52 | 13 | 7 | 1 | 3 | 12 | 5 |
| Hawk Creek nr Maynard, MN23 | NA | 1 | 2 | 3 | 5 | 4 |
| High Island Creek nr Arlington, CR9 | NA | 1 | 4 | 1 | 5 | 1 |
| High Island Creek nr Henderson, CSAH6 | NA | 1 | 2 | 1 | 8 | 1 |
| Kawishiwi River nr Winton, CSAH18 | 3 | 6 | 3 | 1 | 5 | 1 |
| Lac qui Parle River nr Lac qui Parle, CSAH31 | 9 | 4 | 2 | 3 | 5 | 2 |
| Lac qui Parle River nr Providence, CSAH23 | NA | NA | NA | NA | 3 | 2 |
| Le Sueur River at St. Clair, CSAH28 | NA | 1 | 3 | 4 | 14 | 2 |
| Le Sueur River nr Rapidan, CR8 | NA | 2 | 3 | 6 | 17 | 4 |
| Le Sueur River nr Rapidan, MN | 16 | 7 | 1 | 3 | 19 | 4 |
| Leaf River nr Staples, CSAH29 | 7 | 9 | NA | 1 | 7 | NA |
| Leech Lake River nr Ball Club, CR139 | 1 | 2 | 3 | 5 | 8 | 1 |

| Location | | | | | | |
|---|---|---|---|---|---|---|
| Little Beauford Ditch nr Beauford, MN22 | NA | NA | 8 | 3 | 14 | 2 |
| Little Fork River at Little Fork, MN | 7 | 9 | NA | 1 | 11 | 4 |
| Little Fork River nr Linden Grove, TH73 | NA | NA | NA | NA | 4 | 3 |
| Little Fork River nr Littlefork, MN65 | NA | NA | NA | 1 | 8 | 6 |
| Long Prairie River at Long Prairie, MN | NA | 3 | 4 | 1 | 1 | 2 |
| Long Prairie River at Philbrook, 313th Ave | 14 | 11 | 1 | 3 | 5 | 2 |
| Lost River nr Brooks, CR119 | NA | NA | NA | NA | 12 | 3 |
| Maple River nr Rapidan, CR35 | NA | NA | NA | 2 | 11 | 3 |
| Maple River nr Sterling Center, CR18 | NA | 1 | NA | 2 | 15 | 2 |
| Middle Branch Root River nr Fillmore, CSAH5 | NA | NA | NA | NA | 8 | 5 |
| Middle Fork Crow River nr Manannah, CSAH30 | NA | 1 | 3 | 3 | 3 | 2 |
| Middle Fork Zumbro River nr Oronoco,5th St | NA | 1 | NA | NA | 8 | 5 |
| Middle River at Argyle, MN | NA | NA | NA | 3 | 3 | 4 |
| Minnesota River at Judson, CSAH42 | 12 | 6 | NA | 4 | 14 | 5 |
| Minnesota River at Morton, MN | 12 | 5 | NA | 1 | 7 | 8 |
| Minnesota River nr Lac qui Parle, MN | 5 | 5 | NA | 2 | 3 | 4 |
| Mississippi River at Grand Rapids, MN | 2 | 5 | 8 | 8 | 13 | 4 |
| Mississippi River nr Bemidji, CSAH11 | NA | NA | NA | 3 | 3 | 1 |
| Mississippi River nr Bemidji, MN | NA | NA | 1 | 3 | 4 | 1 |
| Mud River nr Grygla, MN89 | NA | NA | NA | 8 | 15 | 3 |
| Mustinka River nr Norcross, MN9 | NA | NA | 2 | 1 | 6 | 3 |
| Nemadji River nr Pleasant Valley, MN23 | NA | NA | NA | 1 | 6 | 1 |

| Station | | | | | | |
|---|---|---|---|---|---|---|
| North Fork Crow River nr Cokato, CSAH4 | NA | 1 | 2 | 3 | 3 | 1 |
| North Fork Crow River nr Rockford, Farmington Ave | 5 | 7 | NA | 2 | 4 | 3 |
| North Fork Whitewater River at Elba, Whitewater Dr | NA | 3 | 2 | 1 | 2 | 3 |
| North Fork Zumbro River nr Mazeppa, CSAH7 | NA | 3 | 2 | NA | 4 | 4 |
| Otter Tail River at Breckenridge, CSAH16 | 10 | 7 | NA | 1 | 11 | 8 |
| Otter Tail River nr Elizabeth, MN | NA | 7 | 1 | 1 | 7 | 3 |
| Pelican River nr Fergus Falls, MN210 | NA | NA | 2 | 4 | 5 | 3 |
| Pine River nr Jenkins, CSAH15 | NA | NA | NA | 1 | 2 | NA |
| Pipestone Creek nr Pipestone, CSAH13 | NA | 1 | 1 | 1 | 7 | NA |
| Pomme De Terre River at Appleton, MN | 9 | 7 | NA | 1 | 2 | 2 |
| Pomme de Terre River nr Hoffman, CR76 | NA | NA | NA | 2 | 7 | 3 |
| Poplar River nr Lutsen, 0.2mi US of MN61 | 3 | 7 | 1 | 1 | 7 | 4 |
| Prairie River nr Taconite, MN | NA | 3 | 3 | NA | 6 | 2 |
| Rapid River at Clementson, MN11 | 8 | 11 | 1 | 2 | 13 | 3 |
| Red Lake River at Fisher, MN | 13 | 8 | NA | 4 | 17 | 2 |
| Red Lake River at High Landing nr Goodridge, MN | NA | NA | 2 | 4 | 6 | 2 |
| Red Lake River at Red Lake Falls, CR13 | NA | NA | 3 | 4 | 8 | 5 |
| Red River of the North nr Kragnes, CSAH26 | 14 | 8 | NA | NA | 7 | 4 |
| Redwood River at Russell, CR15 | NA | 1 | 2 | NA | 7 | 4 |
| Redwood River nr Redwood Falls, MN | 12 | 7 | 2 | 1 | 11 | 6 |
| Rock River at Luverne, CR4 | 7 | 6 | 1 | 1 | 4 | 2 |

| | | | | | | |
|---|---|---|---|---|---|---|
| Rum River at Anoka,headwater side of dam | 10 | 14 | 1 | 6 | 1 | NA |
| S Br. Wild Rice River at CR27 nr Felton, MN | NA | NA | NA | NA | 4 | 1 |
| Sand Hill River at Climax, MN, US-75 | 10 | 11 | 2 | 1 | 9 | 3 |
| Sandhill River nr Fertile, 450th St SW | NA | NA | NA | 3 | 10 | 4 |
| Sauk River nr St. Martin, CR12 | NA | NA | 3 | 2 | 4 | 1 |
| Second Creek nr Aurora, 0.6mi us of CSAH110 | NA | 3 | 2 | 1 | 9 | 4 |
| Seven Mile Creek nr St. Peter, 0.6mi us of US169 | NA | 3 | 9 | 8 | 22 | 5 |
| Shakopee Creek nr Benson, 20th Ave SW | NA | 3 | 4 | 2 | 5 | 2 |
| Shell Rock River nr Gordonsville, CSAH1 | 9 | 6 | NA | NA | 3 | 5 |
| Sleepy Eye Creek nr Cobden, CR8 | NA | NA | 1 | NA | 8 | 3 |
| Snake River above Warren, MN | NA | NA | 2 | 1 | 1 | 3 |
| Snake River nr Big Woods, MN220 | 13 | 3 | NA | 1 | 2 | 3 |
| Snake River nr Pine City, MN | 4 | 8 | NA | 5 | 7 | NA |
| South Branch Buffalo River nr Glyndon, 28th Ave S | NA | NA | NA | 1 | 7 | 2 |
| South Branch Middle Fork Zumbro River nr Oronoco,5th St | NA | 1 | NA | NA | 3 | 5 |
| South Branch Root River at Lanesboro, Rochelle Ave N | NA | 2 | NA | 1 | 7 | 4 |
| South Branch Two Rivers at Hallock, MN175 | NA | 1 | NA | 1 | 4 | 2 |
| South Branch Two Rivers at Lake Bronson, MN | NA | 1 | NA | 2 | 3 | 4 |
| South Fork Crow River at Delano, Bridge Ave | 5 | 7 | 1 | NA | 6 | 4 |
| South Fork Watonwan River nr Madelia, CSAH13 | NA | 1 | NA | NA | 6 | 1 |
| South Fork Zumbro River nr Oronoco, CR121 | NA | 1 | NA | NA | 6 | 5 |

| | | | | | |
|---|---|---|---|---|---|
| Split Rock Creek nr Jasper, 201st St | 6 | 7 | NA | NA | 6 | 2 |
| Spring Creek nr Hanley Falls, 480th St | NA | 1 | 2 | 1 | 7 | 2 |
| St. Francis River nr Big Lake, 164th St | NA | 1 | NA | 6 | 5 | 1 |
| St. Louis River at Floodwood, CSAH8 | NA | NA | 3 | 2 | 5 | 1 |
| St. Louis River at Scanlon, MN | 3 | 1 | 1 | 4 | 11 | 2 |
| St. Louis River nr Forbes, US53 | NA | NA | 3 | 1 | 9 | 5 |
| Stony River nr Babbitt, Tomahawk Rd | NA | 1 | 2 | 1 | 6 | 1 |
| Straight River nr Faribault, MN | 5 | 2 | NA | NA | 2 | NA |
| Sunrise River at Sunrise, CR88 | 2 | 2 | NA | NA | 4 | NA |
| Swan River nr Jacobson, CR438 | NA | NA | 2 | 2 | 4 | 1 |
| Swan River nr Sobieski, MN238 | NA | NA | NA | NA | 3 | 2 |
| Tamarac River nr Florian, CSAH1 | NA | NA | 1 | 4 | 10 | 2 |
| Tamarac River nr Stephen, CSAH22 | 11 | 6 | NA | 3 | 2 | NA |
| Thief River downstream of CSAH 7, 6 mi E of Holt | NA | 3 | 4 | 5 | 6 | 1 |
| Thief River nr Thief River Falls, MN | 12 | 12 | 1 | 3 | 11 | 3 |
| Turtle Creek at Austin, 43rd St | NA | NA | NA | 1 | 9 | 4 |
| Twelvemile Creek nr Wheaton, CSAH14 | NA | 2 | 3 | 1 | 5 | 1 |
| Two Rivers nr Bowlus, 40th St | NA | NA | NA | 2 | NA | NA |
| Two Rivers nr Hallock, CSAH16 | 11 | 9 | NA | 2 | 9 | 2 |
| Watonwan River nr Garden City, CSAH13 | 11 | 8 | 1 | 1 | 15 | 3 |
| Watonwan River nr La Salle, CSAH3 | NA | 1 | 1 | NA | 6 | 2 |
| Wells Creek nr Frontenac, US61 | NA | NA | 4 | 1 | 2 | 3 |
| West Branch Lac qui Parle River at Dawson, Diagonal St | NA | NA | NA | NA | 4 | 2 |
| West Branch Rum River nr Princeton, CR102 | NA | NA | NA | NA | NA | 1 |
| West Fork Des Moines River at Jackson, River St | 8 | 8 | 1 | NA | 5 | 2 |

| | | | | | | |
|---|---|---|---|---|---|---|
| West Fork Des Moines River nr Avoca, CSAH6 | NA | NA | 1 | 1 | 5 | 1 |
| Whitewater River nr Beaver, CSAH30 | 7 | 5 | 2 | 1 | 5 | 3 |
| Wild Rice River at Hendrum, MN | 8 | 11 | 2 | 1 | 14 | 5 |
| Wild Rice River at Twin Valley, MN | NA | 1 | NA | NA | 4 | 1 |
| Wild Rice River nr Mahnomen, CSAH25 | NA | NA | NA | NA | 5 | NA |
| Yellow Bank River nr Odessa, CSAH40 | 6 | 5 | NA | NA | 4 | 2 |
| Yellow Medicine River nr Granite Falls, MN | 7 | 5 | 2 | NA | 8 | 7 |
| Yellow Medicine River nr Hanley Falls, CR18 | NA | 1 | 2 | NA | 7 | 5 |
| Zumbro River at Kellogg, US61 | 7 | 6 | NA | NA | 9 | 2 |
| **Total number of sites with >=3 samples collected during low flows** | **50** | **57** | **23** | **40** | **128** | **68** |

**Table A5.** Linear regression statistics for the log-log relationship between SRP concentrations (mg/L) and normalized flow ($Q/Q_{GM}$). The regressions were run twice. The first regressions (denoted with (1) in the table) included all samples collected for a given site. The second set of regressions (denoted with (2) in the table) excluded samples collected during late summer low flows. '% change in slope' indicates the change in slope between the first and second regression for each site. $CV_C/CV_Q$ is reported using all samples collected for each site. Statistics in bold indicate statistically significant ($p < 0.05$) relationships.

| Site name | Behavior | $CV_C/CV_Q$ | Slope(1) | p(1) | $R^2$(1) | n(1) | Slope(2) | p(2) | $R^2$(2) | n(2) | % slope change |
|---|---|---|---|---|---|---|---|---|---|---|---|
| Baptism River nr Beaver Bay, MN61 | chemodynamic | 0.66 | 0.04 | 0.21 | 0.01 | 184 | 0.05 | 0.14 | 0.01 | 175 | 30.8 |
| Beaver Creek nr Beaver Falls, CSAH2 | mobilizing | 0.34 | **0.42** | **<0.01** | **0.36** | **113** | **0.45** | **<0.01** | **0.37** | **107** | 7.7 |
| Beaver River nr Beaver Bay, 1.2mi us of MN61 | chemodynamic | 0.51 | 0.09 | 0.20 | 0.04 | 48 | 0.07 | 0.42 | 0.02 | 44 | -24.03 |
| Big Cobb River nr Beauford, CSAH16 | mobilizing | 0.56 | **0.35** | **<0.01** | **0.20** | **243** | **0.47** | **<0.01** | **0.20** | **225** | 32.66 |
| Big Fork River at Big Falls, MN | mobilizing | 0.61 | **0.16** | **<0.01** | **0.06** | **264** | **0.12** | **0.01** | **0.03** | **250** | -26.35 |

| | | | | | | | | | | |
|---|---|---|---|---|---|---|---|---|---|---|
| Big Fork River nr Bigfork, MN6 | chemodynamic | 1.01 | -0.04 | 0.62 | 0.00 | 116 | -0.01 | 0.91 | 0.00 | 102 | -70.13 |
| Big Fork River nr Craigville, MN6 | mobilizing | 0.73 | **0.22** | **<0.01** | **0.07** | **112** | 0.15 | 0.10 | 0.03 | 102 | -32.38 |
| Big Sucker Creek nr Palmers, CR258 | chemodynamic | 0.6 | 0.03 | 0.40 | 0.00 | 162 | 0.01 | 0.71 | 0.00 | 158 | -54.47 |
| Blue Earth River nr Rapidan, MN | mobilizing | 0.74 | **0.62** | **<0.01** | **0.45** | **437** | **0.64** | **<0.01** | **0.41** | **418** | 1.69 |
| Blue Earth River nr Winnebago, CSAH12 | mobilizing | 0.71 | **0.47** | **<0.01** | **0.27** | **139** | **0.50** | **<0.01** | **0.22** | **131** | 6.3 |
| Bois de Sioux River nr Doran, MN | mobilizing | 0.32 | **0.23** | **<0.01** | **0.26** | **426** | **0.24** | **<0.01** | **0.25** | **414** | 2.88 |
| Boy River nr Boy River, CSAH53 | chemodynamic | 1.77 | 0.28 | 0.09 | 0.06 | 49 | 0.29 | 0.14 | 0.05 | 44 | 0.7 |
| Brule River nr Hovland, MN61 | chemodynamic | 0.94 | 0.14 | 0.17 | 0.10 | 21 | 0.14 | 0.20 | 0.09 | 20 | 2.09 |
| Buffalo Creek nr Glencoe, CSAH1 | chemodynamic | 0.95 | -0.05 | 0.56 | 0.00 | 71 | 0.09 | 0.39 | 0.01 | 68 | -260.73 |
| Buffalo River nr Georgetown, CR108 | mobilizing | 0.32 | **0.30** | **<0.01** | **0.25** | **438** | **0.32** | **<0.01** | **0.26** | **427** | 7.98 |
| Buffalo River nr Glyndon, CSAH19 | mobilizing | 0.50 | **0.40** | **<0.01** | **0.28** | **73** | **0.46** | **<0.01** | **0.32** | **70** | 15.25 |
| Buffalo River nr Hawley, MN | mobilizing | 0.53 | **0.40** | **<0.01** | **0.27** | **97** | **0.43** | **<0.01** | **0.28** | **93** | 6.76 |
| Cannon River at Morristown, CSAH16 | chemodynamic | 0.81 | -0.25 | 0.07 | 0.06 | 58 | -0.20 | 0.18 | 0.03 | 56 | -20.19 |
| Cannon River at Welch, MN | mobilizing | 1.07 | **0.49** | **<0.01** | **0.09** | **119** | **0.68** | **<0.01** | **0.17** | **113** | 37.33 |
| Cedar River nr Austin, MN | diluting | 0.57 | **-0.25** | **<0.01** | **0.19** | **265** | **-0.22** | **<0.01** | **0.14** | **256** | -12.34 |
| Chippewa River nr Clontarf, CSAH22 | mobilizing | 1.51 | **0.94** | **<0.01** | **0.28** | **123** | **1.03** | **<0.01** | **0.29** | **118** | 10.1 |
| Chippewa River nr Milan, MN40 | mobilizing | 1.33 | **0.65** | **<0.01** | **0.29** | **304** | **0.68** | **<0.01** | **0.28** | **295** | 4.66 |
| Clearwater River at Plummer, MN | mobilizing | 0.73 | **0.44** | **<0.01** | **0.15** | **99** | **0.55** | **<0.01** | **0.19** | **94** | 24.76 |

| | | | | | | | | | | |
|---|---|---|---|---|---|---|---|---|---|---|
| Clearwater River at Red Lake Falls, MN | mobilizing | 0.55 | **0.41** | **<0.01** | **0.18** | 396 | **0.37** | **<0.01** | **0.14** | 380 | -10.15 |
| Clearwater River nr Clearwater, CR145 | chemodynamic | 1.22 | -0.12 | 0.44 | 0.01 | 59 | -0.28 | 0.10 | 0.05 | 56 | 137.88 |
| Cloquet River nr Brimson, CSAH44 | chemodynamic | 0.62 | -0.03 | 0.75 | 0.00 | 80 | -0.13 | 0.29 | 0.02 | 73 | 325.35 |
| Cloquet River nr Burnett, CR694 | chemodynamic | 1.04 | 0.03 | 0.59 | 0.00 | 142 | -0.02 | 0.77 | 0.00 | 129 | -165.87 |
| Cottonwood River nr Leavenworth, CR8 | mobilizing | 0.91 | **0.46** | **<0.01** | **0.35** | 136 | **0.47** | **<0.01** | **0.27** | 130 | 1.75 |
| Cottonwood River nr New Ulm, MN68 | mobilizing | 0.73 | **0.62** | **<0.01** | **0.49** | 389 | **0.60** | **<0.01** | **0.44** | 375 | -3.08 |
| East Branch Blue Earth River at Blue Earth, CSAH16 | mobilizing | 0.84 | **0.20** | **<0.01** | **0.06** | 140 | **0.29** | **<0.01** | **0.08** | 132 | 41.42 |
| East Branch Chippewa River nr Benson, CR78 | mobilizing | 0.91 | **0.44** | **<0.01** | **0.13** | 119 | **0.69** | **<0.01** | **0.24** | 113 | 57.36 |
| East Fork Rapid River nr Clementson, CSAH18 | diluting | 0.52 | **-0.18** | **<0.01** | **0.18** | 93 | **-0.14** | **0.01** | **0.07** | 85 | -24.78 |
| Elk River nr Big Lake, MN | mobilizing | 1.12 | **0.44** | **0.02** | **0.09** | 63 | **0.50** | **0.01** | **0.11** | 61 | 11.55 |
| Hawk Creek nr Granite Falls, CR52 | mobilizing | 0.51 | **0.50** | **<0.01** | **0.37** | 375 | **0.44** | **<0.01** | **0.31** | 363 | -11.18 |
| Hawk Creek nr Maynard, MN23 | chemostatic | 0.3 | **0.34** | **<0.01** | **0.30** | 120 | **0.34** | **<0.01** | **0.29** | 115 | 2.35 |
| High Island Creek nr Arlington, CR9 | chemodynamic | 0.78 | 0.09 | 0.12 | 0.02 | 100 | 0.14 | 0.07 | 0.03 | 95 | 54.04 |
| High Island Creek nr Henderson, CSAH6 | mobilizing | 0.79 | **0.36** | **<0.01** | **0.20** | 96 | **0.33** | **<0.01** | **0.11** | 88 | -10.11 |
| Kawishiwi River nr Winton, CSAH18 | chemodynamic | 1.21 | -0.12 | 0.08 | 0.02 | 143 | **-0.16** | **0.02** | **0.04** | 138 | 34.45 |
| Lac qui Parle River nr Lac qui Parle, CSAH31 | mobilizing | 0.71 | **0.49** | **<0.01** | **0.33** | 186 | **0.49** | **<0.01** | **0.31** | 181 | 0.31 |
| Lac qui Parle River nr Providence, CSAH23 | mobilizing | 0.49 | **0.58** | **<0.01** | **0.40** | 72 | **0.70** | **<0.01** | **0.44** | 69 | 20.68 |
| Le Sueur River at St. Clair, CSAH28 | mobilizing | 0.34 | **0.32** | **<0.01** | **0.21** | 179 | **0.38** | **<0.01** | **0.20** | 165 | 21.25 |

| | | | | | | | | | | | |
|---|---|---|---|---|---|---|---|---|---|---|---|
| Le Sueur River nr Rapidan, CR8 | mobilizing | 0.43 | **0.62** | **<0.01** | **0.50** | 269 | **0.67** | **<0.01** | **0.45** | 252 | 7.11 |
| Le Sueur River nr Rapidan, MN | mobilizing | 0.59 | **0.56** | **<0.01** | **0.46** | 478 | **0.55** | **<0.01** | **0.42** | 459 | -0.76 |
| Leaf River nr Staples, CSAH29 | mobilizing | 0.54 | **0.23** | **<0.01** | **0.15** | 181 | **0.23** | **<0.01** | **0.14** | 174 | -0.59 |
| Leech Lake River nr Ball Club, CR139 | chemodynamic | 1.63 | 0.05 | 0.68 | 0.00 | 77 | 0.05 | 0.76 | 0.00 | 69 | -13.76 |
| Little Beauford Ditch nr Beauford, MN22 | mobilizing | 0.36 | **0.27** | **<0.01** | **0.18** | 198 | **0.42** | **<0.01** | **0.32** | 184 | 57.82 |
| Little Fork River at Little Fork, MN | chemodynamic | 0.49 | 0.02 | 0.62 | 0.00 | 251 | -0.04 | 0.27 | 0.01 | 240 | -336.91 |
| Little Fork River nr Linden Grove, TH73 | chemodynamic | 0.36 | -0.05 | 0.35 | 0.02 | 50 | -0.02 | 0.79 | 0.00 | 46 | -65.46 |
| Little Fork River nr Littlefork, MN65 | chemodynamic | 0.49 | -0.01 | 0.88 | 0.00 | 113 | -0.05 | 0.36 | 0.01 | 105 | 598.71 |
| Long Prairie River at Long Prairie, MN | mobilizing | 1.05 | **0.41** | **<0.01** | **0.14** | 60 | **0.44** | **<0.01** | **0.16** | 59 | 7.12 |
| Long Prairie River at Philbrook, 313th Ave | mobilizing | 0.71 | **0.11** | **0.02** | **0.02** | 279 | **0.10** | **0.04** | **0.02** | 274 | -5.65 |
| Lost River nr Brooks, CR119 | chemodynamic | 0.68 | 0.11 | 0.24 | 0.02 | 79 | **0.42** | **<0.01** | **0.15** | 67 | 273.75 |
| Maple River nr Rapidan, CR35 | mobilizing | 0.37 | **0.56** | **<0.01** | **0.50** | 216 | **0.62** | **<0.01** | **0.49** | 205 | 11.2 |
| Maple River nr Sterling Center, CR18 | mobilizing | 0.34 | **0.24** | **<0.01** | **0.22** | 269 | **0.33** | **<0.01** | **0.26** | 254 | 36.51 |
| Middle Branch Root River nr Fillmore, CSAH5 | mobilizing | 0.72 | **0.53** | **<0.01** | **0.32** | 121 | **0.60** | **<0.01** | **0.33** | 113 | 12.67 |
| Middle Fork Crow River nr Manannah, CSAH30 | mobilizing | 1.57 | **0.91** | **<0.01** | **0.28** | 70 | **0.83** | **<0.01** | **0.25** | 67 | -8.82 |
| Middle Fork Zumbro River nr Oronoco,5th St | mobilizing | 0.75 | **0.58** | **<0.01** | **0.50** | 127 | **0.62** | **<0.01** | **0.51** | 119 | 6.60 |
| Middle River at Argyle, MN | mobilizing | 0.49 | **0.16** | **<0.01** | **0.11** | 99 | **0.17** | **<0.01** | **0.10** | 96 | 3.79 |
| Minnesota River at Judson, CSAH42 | mobilizing | 0.79 | **0.52** | **<0.01** | **0.21** | 433 | **0.53** | **<0.01** | **0.19** | 419 | 1.36 |

| | | | | | | | | | | | |
|---|---|---|---|---|---|---|---|---|---|---|---|
| Minnesota River at Morton, MN | mobilizing | 0.77 | **0.26** | **<0.01** | **0.06** | **299** | **0.28** | **<0.01** | **0.07** | **292** | 9.18 |
| Minnesota River nr Lac qui Parle, MN | mobilizing | 0.9 | **0.31** | **0.01** | **0.05** | **165** | **0.35** | **<0.01** | **0.06** | **162** | 12.09 |
| Mississippi River at Grand Rapids, MN | mobilizing | 1.32 | **0.27** | **<0.01** | **0.06** | **184** | **0.42** | **<0.01** | **0.11** | **171** | 56.08 |
| Mississippi River nr Bemidji, CSAH11 | chemodynamic | 1.13 | -0.22 | 0.18 | 0.04 | 44 | -0.19 | 0.32 | 0.03 | 41 | -16.08 |
| Mississippi River nr Bemidji, MN | chemodynamic | 1.89 | -0.02 | 0.92 | 0.00 | 42 | -0.12 | 0.66 | 0.01 | 38 | 449.62 |
| Mud River nr Grygla, MN89 | mobilizing | 1.01 | **0.29** | **<0.01** | **0.17** | **157** | **0.40** | **<0.01** | **0.23** | **142** | 40.11 |
| Mustinka River nr Norcross, MN9 | mobilizing | 0.58 | **0.41** | **<0.01** | **0.20** | **91** | **0.62** | **<0.01** | **0.30** | **85** | 51.09 |
| Nemadji River nr Pleasant Valley, MN23 | mobilizing | 0.88 | **0.30** | **<0.01** | **0.31** | **68** | **0.28** | **<0.01** | **0.26** | **62** | -7.72 |
| North Fork Crow River nr Cokato, CSAH4 | mobilizing | 1.49 | **0.90** | **<0.01** | **0.19** | **74** | **1.02** | **<0.01** | **0.21** | **71** | 14.2 |
| North Fork Crow River nr Rockford, Farmington Ave | mobilizing | 1.05 | **0.33** | **<0.01** | **0.07** | **206** | **0.32** | **<0.01** | **0.06** | **202** | -4.36 |
| North Fork Whitewater River at Elba, Whitewater Dr | mobilizing | 0.99 | **0.93** | **<0.01** | **0.68** | **82** | **0.94** | **<0.01** | **0.68** | **80** | 1.08 |
| North Fork Zumbro River nr Mazeppa, CSAH7 | mobilizing | 0.90 | **0.58** | **<0.01** | **0.40** | **99** | **0.60** | **<0.01** | **0.39** | **95** | 2.88 |
| Otter Tail River at Breckenridge, CSAH16 | mobilizing | 3.26 | **1.21** | **<0.01** | **0.22** | **231** | **1.35** | **<0.01** | **0.24** | **220** | 11.46 |
| Otter Tail River nr Elizabeth, MN | chemodynamic | 3.18 | 0.15 | 0.32 | 0.01 | 70 | 0.21 | 0.23 | 0.02 | 63 | 41.57 |
| Pelican River nr Fergus Falls, MN210 | mobilizing | 1.27 | **0.49** | **0.01** | **0.09** | **77** | **0.60** | **<0.01** | **0.11** | **72** | 23.70 |
| Pine River nr Jenkins, CSAH15 | chemodynamic | 0.62 | 0.23 | 0.21 | 0.08 | 22 | 0.36 | 0.10 | 0.15 | 20 | 60.81 |
| Pipestone Creek nr Pipestone, CSAH13 | mobilizing | 0.55 | **0.60** | **<0.01** | **0.55** | **73** | **0.71** | **<0.01** | **0.56** | **66** | 18.10 |

| | | | | | | | | | | | |
|---|---|---|---|---|---|---|---|---|---|---|---|
| Pomme De Terre River at Appleton, MN | mobilizing | 1.07 | **0.95** | **<0.01** | **0.33** | **166** | **0.94** | **<0.01** | **0.33** | **164** | -0.47 |
| Pomme de Terre River nr Hoffman, CR76 | mobilizing | 1.94 | **0.91** | **<0.01** | **0.25** | **110** | **1.25** | **<0.01** | **0.32** | **103** | 38.26 |
| Poplar River nr Lutsen, 0.2mi US of MN61 | chemodynamic | 0.75 | 0.05 | 0.20 | 0.01 | 167 | 0.04 | 0.42 | 0.00 | 160 | -29.4 |
| Prairie River nr Taconite, MN | chemodynamic | 0.56 | -0.13 | 0.08 | 0.05 | 59 | **-0.18** | **0.05** | **0.08** | **53** | 33.85 |
| Rapid River at Clementson, MN11 | mobilizing | 0.51 | **0.04** | **0.05** | **0.01** | **309** | 0.01 | 0.66 | 0.00 | 296 | -74.51 |
| Red Lake River at Fisher, MN | mobilizing | 0.81 | **0.71** | **<0.01** | **0.34** | **383** | **0.77** | **<0.01** | **0.35** | **366** | 7.81 |
| Red Lake River at High Landing nr Goodridge, MN | mobilizing | 4.74 | **0.65** | **0.01** | **0.11** | **59** | **0.90** | **<0.01** | **0.17** | **53** | 37.65 |
| Red Lake River at Red Lake Falls, CR13 | mobilizing | 1.47 | **0.93** | **<0.01** | **0.42** | **121** | **1.00** | **<0.01** | **0.39** | **113** | 6.80 |
| Red River of the North nr Kragnes, CSAH26 | chemodynamic | 0.32 | -0.02 | 0.46 | 0.00 | 376 | 0.00 | 0.89 | 0.00 | 369 | -80.69 |
| Redwood River at Russell, CR15 | mobilizing | 0.61 | **0.28** | **<0.01** | **0.12** | **104** | **0.34** | **<0.01** | **0.13** | **97** | 21.16 |
| Redwood River nr Redwood Falls, MN | diluting | 0.66 | **-0.10** | **<0.01** | **0.02** | **327** | **-0.20** | **<0.01** | **0.10** | **316** | 96.5 |
| Rock River at Luverne, CR4 | mobilizing | 0.64 | **0.71** | **<0.01** | **0.52** | **194** | **0.71** | **<0.01** | **0.51** | **190** | 0.10 |
| Rum River at Anoka,headwater side of dam | mobilizing | 0.72 | **0.37** | **<0.01** | **0.14** | **171** | **0.38** | **<0.01** | **0.14** | **170** | 1.48 |
| S Br. Wild Rice River at CR27 nr Felton, MN | mobilizing | 0.32 | **0.29** | **<0.01** | **0.29** | **48** | **0.30** | **<0.01** | **0.26** | **44** | 2.91 |
| Sand Hill River at Climax, MN, US-75 | mobilizing | 0.41 | **0.50** | **<0.01** | **0.43** | **331** | **0.51** | **<0.01** | **0.43** | **322** | 2.12 |
| Sandhill River nr Fertile, 450th St SW | mobilizing | 0.70 | **0.47** | **<0.01** | **0.23** | **104** | **0.52** | **<0.01** | **0.23** | **94** | 9.61 |
| Sauk River nr St. Martin, CR12 | mobilizing | 1.27 | **0.83** | **<0.01** | **0.30** | **60** | **1.01** | **<0.01** | **0.37** | **56** | 21.66 |
| Second Creek nr Aurora, 0.6mi us of CSAH110 | mobilizing | 1.48 | **0.18** | **0.04** | **0.04** | **108** | 0.20 | 0.06 | 0.04 | 99 | 8.43 |

| Seven Mile Creek nr St. Peter, 0.6mi us of US169 | mobilizing | 0.56 | **0.41** | **<0.01** | **0.39** | **326** | **0.40** | **<0.01** | **0.33** | **304** | -2.53 |
|---|---|---|---|---|---|---|---|---|---|---|---|
| Shakopee Creek nr Benson, 20th Ave SW | mobilizing | 0.75 | **0.46** | **<0.01** | **0.13** | **135** | **0.76** | **<0.01** | **0.28** | **130** | 64.74 |
| Shell Rock River nr Gordonsville, CSAH1 | diluting | 1.69 | **-0.73** | **<0.01** | **0.29** | **192** | **-0.73** | **<0.01** | **0.28** | **189** | -0.47 |
| Sleepy Eye Creek nr Cobden, CR8 | mobilizing | 0.51 | **0.42** | **<0.01** | **0.44** | **118** | **0.43** | **<0.01** | **0.40** | **110** | 3.6 |
| Snake River above Warren, MN | mobilizing | 0.44 | **0.22** | **<0.01** | **0.12** | **78** | **0.21** | **<0.01** | **0.11** | **77** | -5.76 |
| Snake River nr Big Woods, MN220 | chemostatic | 0.26 | **0.12** | **<0.01** | **0.13** | **327** | **0.12** | **<0.01** | **0.13** | **325** | 0.48 |
| Snake River nr Pine City, MN | mobilizing | 0.57 | **0.23** | **<0.01** | **0.11** | **213** | **0.22** | **<0.01** | **0.10** | **206** | -4.32 |
| South Branch Buffalo River nr Glyndon, 28th Ave S | chemostatic | 0.28 | **0.20** | **<0.01** | **0.17** | **105** | **0.29** | **<0.01** | **0.24** | **98** | 49.53 |
| South Branch Middle Fork Zumbro River nr Oronoco,5th St | mobilizing | 0.59 | **0.49** | **<0.01** | **0.39** | **103** | **0.52** | **<0.01** | **0.40** | **100** | 6.68 |
| South Branch Root River at Lanesboro, Rochelle Ave N | mobilizing | 1.05 | **0.79** | **<0.01** | **0.47** | **127** | **0.80** | **<0.01** | **0.45** | **120** | 1.00 |
| South Branch Two Rivers at Hallock, MN175 | mobilizing | 0.46 | **0.21** | **<0.01** | **0.11** | **104** | **0.28** | **<0.01** | **0.14** | **100** | 33.96 |
| South Branch Two Rivers at Lake Bronson, MN | mobilizing | 0.46 | **0.17** | **0.01** | **0.07** | **89** | **0.17** | **0.02** | **0.06** | **86** | 1.82 |
| South Fork Crow River at Delano, Bridge Ave | mobilizing | 0.53 | **0.10** | **0.05** | **0.02** | **212** | **0.12** | **0.03** | **0.02** | **206** | 16.77 |
| South Fork Watonwan River nr Madelia, CSAH13 | mobilizing | 0.94 | **0.21** | **<0.01** | **0.08** | **123** | **0.31** | **<0.01** | **0.13** | **117** | 46.6 |
| South Fork Zumbro River nr Oronoco, CR121 | mobilizing | 0.86 | **0.18** | **<0.01** | **0.06** | **126** | **0.23** | **<0.01** | **0.09** | **120** | 31.70 |

| Site | Type | | | | | | | | | | |
|---|---|---|---|---|---|---|---|---|---|---|---|
| Split Rock Creek nr Jasper, 201st St | mobilizing | 0.52 | **0.60** | **<0.01** | **0.50** | **171** | **0.59** | **<0.01** | **0.47** | **165** | -2.05 |
| Spring Creek nr Hanley Falls, 480th St | chemodynamic | 0.49 | -0.07 | 0.26 | 0.01 | 106 | 0.02 | 0.77 | 0.00 | 99 | -133.05 |
| St. Francis River nr Big Lake, 164th St | chemodynamic | 0.78 | -0.17 | 0.27 | 0.02 | 61 | -0.16 | 0.32 | 0.02 | 56 | -2.19 |
| St. Louis River at Floodwood, CSAH8 | mobilizing | 0.99 | **0.24** | **<0.01** | **0.09** | **87** | **0.30** | **<0.01** | **0.12** | **82** | 21.11 |
| St. Louis River at Scanlon, MN | mobilizing | 1.08 | **0.24** | **<0.01** | **0.11** | **138** | **0.20** | **<0.01** | **0.06** | **127** | -16.9 |
| St. Louis River nr Forbes, US53 | mobilizing | 0.58 | **0.15** | **0.01** | **0.06** | **120** | **0.13** | **0.05** | **0.03** | **111** | -13.42 |
| Stony River nr Babbitt, Tomahawk Rd | chemodynamic | 0.9 | 0.04 | 0.70 | 0.00 | 76 | -0.13 | 0.33 | 0.01 | 70 | -419.25 |
| Straight River nr Faribault, MN | mobilizing | 0.50 | **0.34** | **<0.01** | **0.28** | **111** | **0.35** | **<0.01** | **0.28** | **109** | 2.89 |
| Sunrise River at Sunrise, CR88 | chemodynamic | 0.51 | -0.04 | 0.53 | 0.00 | 90 | -0.02 | 0.81 | 0.00 | 86 | -57.9 |
| Swan River nr Jacobson, CR438 | chemodynamic | 0.94 | -0.19 | 0.20 | 0.03 | 55 | -0.22 | 0.24 | 0.03 | 51 | 14.27 |
| Swan River nr Sobieski, MN238 | mobilizing | 1.06 | **0.52** | **<0.01** | **0.26** | **62** | **0.57** | **<0.01** | **0.27** | **59** | 9.49 |
| Tamarac River nr Florian, CSAH1 | mobilizing | 0.73 | **0.18** | **<0.01** | **0.14** | **111** | **0.26** | **<0.01** | **0.17** | **101** | 41.22 |
| Tamarac River nr Stephen, CSAH22 | mobilizing | 0.38 | **0.25** | **<0.01** | **0.29** | **289** | **0.24** | **<0.01** | **0.27** | **287** | -2.40 |
| Thief River downstream of CSAH 7, 6 mi E of Holt | mobilizing | 1.53 | **0.11** | **0.05** | **0.02** | **161** | 0.06 | 0.34 | 0.01 | 155 | -45.73 |
| Thief River nr Thief River Falls, MN | mobilizing | 0.83 | **0.14** | **<0.01** | **0.05** | **421** | **0.13** | **<0.01** | **0.04** | **410** | -1.92 |
| Turtle Creek at Austin, 43rd St | mobilizing | 0.92 | **0.58** | **<0.01** | **0.21** | **103** | **0.72** | **<0.01** | **0.24** | **94** | 23.05 |
| Twelvemile Creek nr Wheaton, CSAH14 | mobilizing | 0.32 | **0.15** | **0.02** | **0.07** | **84** | **0.33** | **<0.01** | **0.20** | **79** | 117.23 |
| Two Rivers nr Bowlus, 40th St | chemodynamic | 0.88 | -0.16 | 0.47 | 0.03 | 21 | -0.16 | 0.47 | 0.03 | 21 | 0 |

| | | | | | | | | | | | |
|---|---|---|---|---|---|---|---|---|---|---|---|
| Two Rivers nr Hallock, CSAH16 | mobilizing | 0.46 | **0.29** | **<0.01** | **0.35** | **303** | **0.27** | **<0.01** | **0.30** | **294** | -5.38 |
| Watonwan River nr Garden City, CSAH13 | mobilizing | 0.66 | **0.45** | **<0.01** | **0.38** | **360** | **0.42** | **<0.01** | **0.32** | **345** | -6.66 |
| Watonwan River nr La Salle, CSAH3 | mobilizing | 0.44 | **0.22** | **<0.01** | **0.12** | **140** | **0.30** | **<0.01** | **0.18** | **134** | 38 |
| Wells Creek nr Frontenac, US61 | mobilizing | 1.57 | **1.26** | **<0.01** | **0.59** | **56** | **1.26** | **<0.01** | **0.58** | **54** | -0.52 |
| West Branch Lac qui Parle River at Dawson, Diagonal St | mobilizing | 0.47 | **0.25** | **<0.01** | **0.12** | **64** | **0.42** | **<0.01** | **0.22** | **60** | 67.62 |
| West Branch Rum River nr Princeton, CR102 | chemodynamic | 0.33 | 0.15 | 0.12 | 0.10 | 25 | 0.15 | 0.12 | 0.10 | 25 | 0 |
| West Fork Des Moines River at Jackson, River St | chemodynamic | 1.57 | -0.04 | 0.59 | 0.00 | 165 | -0.05 | 0.56 | 0.00 | 160 | 16.04 |
| West Fork Des Moines River nr Avoca, CSAH6 | mobilizing | 0.69 | **0.60** | **<0.01** | **0.33** | **77** | **0.80** | **<0.01** | **0.41** | **72** | 34.59 |
| Whitewater River nr Beaver, CSAH30 | mobilizing | 1.28 | **0.82** | **<0.01** | **0.53** | **206** | **0.83** | **<0.01** | **0.53** | **201** | 0.52 |
| Wild Rice River at Hendrum, MN | mobilizing | 0.48 | **0.43** | **<0.01** | **0.36** | **402** | **0.46** | **<0.01** | **0.36** | **388** | 7.08 |
| Wild Rice River at Twin Valley, MN | mobilizing | 0.75 | **0.58** | **<0.01** | **0.49** | **57** | **0.55** | **<0.01** | **0.39** | **53** | -3.98 |
| Wild Rice River nr Mahnomen, CSAH25 | mobilizing | 0.73 | **0.22** | **0.04** | **0.10** | **43** | **0.36** | **0.03** | **0.13** | **38** | 60.21 |
| Yellow Bank River nr Odessa, CSAH40 | mobilizing | 0.52 | **0.55** | **<0.01** | **0.45** | **177** | **0.56** | **<0.01** | **0.44** | **173** | 1.26 |
| Yellow Medicine River nr Granite Falls, MN | mobilizing | 0.7 | **0.35** | **<0.01** | **0.25** | **257** | **0.34** | **<0.01** | **0.23** | **249** | -2.22 |
| Yellow Medicine River nr Hanley Falls, CR18 | mobilizing | 0.7 | **0.46** | **<0.01** | **0.34** | **119** | **0.48** | **<0.01** | **0.30** | **112** | 3.82 |
| Zumbro River at Kellogg, US61 | mobilizing | 1.03 | **0.59** | **<0.01** | **0.38** | **229** | **0.60** | **<0.01** | **0.37** | **220** | 2.57 |

**Table A6.** Conditional Permutation Importance value for predictor variables used in the random forest model.

| Predictor | Importance | Description |
|---|---|---|
| PctCrop2019CatRp100 | 0.009498 | % of AOI area classified as crop land use (NLCD class 82) |
| PctCrop2019WsRp100 | 0.009437 | % of AOI area classified as crop land use (NLCD class 82) |
| KffactCat | 0.009059 | Mean of STATSGO Kffactor raster within AOI. The Universal Soil Loss Equation (USLE) and represents a relative index of susceptibility of bare, cultivated soil to particle detachment and transport by rainfall |
| PermCat | 0.005716 | Mean permeability (cm/hour) of soils (STATSGO) within AOI |
| AgKffactWs | 0.005292 | Mean of STATSGO Kffactor raster on agricultural land (NLCD 2006) within AOI. The Universal Soil Loss Equation (USLE) and represents a relative index of susceptibility of bare, cultivated soil to particle detachment and transport by rainfall |
| PctMxFst2019WsRp100 | 0.005098 | % of AOI area classified as mixed deciduous/evergreen forest land cover (NLCD class 43) |
| PctUrbOp2019CatRp100 | 0.004543 | % of AOI area classified as developed, open space land use (NLCD class 21) |
| Precip_Minus_EVTWs | 0.004364 | This dataset represents surplus precipitation (mm): precipitation minus potential evaporation described in DOI: 10.1016/j.scitotenv.2020.137661 within individual, local NHDPlusV2 catchments and upstream, contributing watersheds. |
| ClayWs | 0.004223 | Mean % clay content of soils (STATSGO) within AOI |
| PctWdWet2019WsRp100 | 0.004041 | % of AOI area classified as woody wetland land cover (NLCD class 90) |
| PctGrs2019Ws | 0.003979 | % of AOI area classified as grassland/herbaceous land cover (NLCD class 71) |
| Pestic97Cat | 0.003880 | Mean pesticide use (kg/km2) in yr. 1997 within AOI. |
| MAST_2013 | 0.003805 | Predicted mean annual stream temperature (Jan-Dec) for year 2008, 2009, 2013, 2014 |
| Phos_Crop_UptakeWs | 0.003715 | Phosphorus uptake by crops in the watershed |
| FertWs | 0.003536 | Mean rate of synthetic nitrogen fertilizer application to agricultural land in kg N/ha/yr, within the AOI |
| WWTPAllDensWs | 0.003328 | Density (number/ km2) of all wastewater treatment plants per AOI |
| SiO2Cat | 0.003151 | Mean % of lithological silicon dioxide (SiO2) content in surface or near surface geology within AOI |
| PctHay2019Cat | 0.003151 | % of AOI area classified as hay land use (NLCD class 81) |
| RdCrsSlpWtdCat | 0.003115 | Density of roads-stream intersections (2010 Census Tiger Lines-NHD stream lines) multiplied by NHDPlusV21 slope within AOI (crossings*slope/square km) |
| PctImp2013CatRp100 | 0.003096 | Mean imperviousness of anthropogenic surfaces within AOI (NLCD) |
| Al2O3Cat | 0.003007 | Mean % of lithological aluminum oxide (Al2O3) content in surface or near surface geology within AOI |
| PctGlacLakeFineWs | 0.002988 | % of AOI area classified as lithology type: glacial lake sediment, fine-textured |
| PctCrop2019Ws | 0.002930 | % of AOI area classified as crop land use (NLCD class 82) |

| | | |
|---|---|---|
| CompStrgthCat | 0.002860 | Mean lithological uniaxial compressive strength (megaPascals) content in surface or near surface geology within AOI |
| NsurpCat | 0.002853 | Nitrogen Surplus (kg N / yr) per AOI - excluding biological N Fixation |
| K2OCat | 0.002668 | Mean % of lithological potassium oxide (K2O) content in surface or near surface geology within AOI |
| AgKffactCat | 0.002657 | Mean of STATSGO Kffactor raster on agricultural land (NLCD 2006) within AOI. The Universal Soil Loss Equation (USLE) and represents a relative index of susceptibility of bare, cultivated soil to particle detachment and transport by rainfall |
| NABD_NIDStorWs | 0.002568 | Volume all reservoirs (NID_STORA in NID) per unit area of AOI (cubic meters/square km) |
| PctMxFst2019Ws | 0.002464 | % of AOI area classified as mixed deciduous/evergreen forest land cover (NLCD class 43) |
| ElevWs | 0.002449 | Mean AOI elevation (m) |
| PctOw2019Cat | 0.002403 | % of AOI area classified as open water land cover (NLCD class 11) |
| PctDecid2019CatRp100 | 0.002028 | % of AOI area classified as deciduous forest land cover (NLCD class 41) |
| OmCat | 0.002027 | Mean organic matter content (% by weight) of soils (STATSGO) within AOI |
| PctCrop2019Cat | 0.001864 | % of AOI area classified as crop land use (NLCD class 82) |
| WaterInputCat | 0.001863 | Water Input (km2/cm): Ratio of the total area of irrigated land to precipitation within AOI |
| FertCat | 0.001784 | Mean rate of synthetic nitrogen fertilizer application to agricultural land in kg N/ha/yr, within the AOI |
| PctDecid2019Cat | 0.001709 | % of AOI area classified as deciduous forest land cover (NLCD class 41) |
| NO3_2008Ws | 0.001689 | Annual gradient map of precipitation-weighted mean deposition for nitrate ion concentration wet deposition for 2008 in kg of NO3/ha/yr, within AOI |
| Phos_FertCat | 0.001659 | Mean rate of synthetic nitrogen fertilizer application to agricultural land in kg N/ha/yr, within the AOI |
| MAST_2014 | 0.001589 | Predicted mean annual stream temperature (Jan-Dec) for year 2008, 2009, 2013, 2014 |
| Na2OCat | 0.001586 | Mean % of lithological sodium oxide (Na2O) content in surface or near surface geology within AOI |
| WaterInputWs | 0.001507 | Water Input (km2/cm): Ratio of the total area of irrigated land to precipitation within AOI |
| RockNWs | 0.001481 | N from rock weathering (kg/ km2) within AOI |
| Pestic97Ws | 0.001459 | Mean pesticide use (kg/km2) in yr. 1997 within AOI. |
| PctUrbMd2019CatRp100 | 0.001402 | % of AOI area classified as developed, medium-intensity land use (NLCD class 23) |
| MSST_2014 | 0.001387 | Predicted mean summer stream temperature (July-Aug) for years 2008, 2009, 2013, 2014 |
| MgOCat | 0.001343 | Mean % of lithological magnesium oxide (MgO) content in surface or near surface geology within AOI |
| MWST_2013 | 0.001315 | Predicted mean winter stream temperature (Jan-Feb) for year 2008, 2009, 2013, 2014 |

| | | |
|---|---|---|
| PctHbWet2019Cat | 0.001298 | % of AOI area classified as herbaceous wetland land cover (NLCD class 95) |
| Fe2O3Cat | 0.001270 | Mean % of lithological ferric oxide (Fe2O3) content in surface or near surface geology within AOI |
| PctShrb2019Ws | 0.001265 | % of AOI area classified as shrub/scrub land cover (NLCD class 52) |
| HydrlCondCat | 0.001220 | Mean lithological hydraulic conductivity (micrometers per second) content in surface or near surface geology within AOI |
| RdDensWs | 0.001211 | Density of roads (2010 Census Tiger Lines) within AOI (km/square km) |
| Phos_ManureCat | 0.001210 | Mean rate of manure application to agricultural land from confined animal feeding operations in kg N/ha/yr, within AOI |
| PctWdWet2019Ws | 0.001191 | % of AOI area classified as woody wetland land cover (NLCD class 90) |
| RdCrsWs | 0.001175 | Density of roads-stream intersections (2010 Census Tiger Lines-NHD stream lines) within AOI (crossings/square km) |
| NPDESDensWs | 0.001156 | Density of permitted NPDES (National Pollutant Discharge Elimination System) sites within AOI (sites/square km) |
| PctImp2008Ws | 0.001152 | Mean imperviousness of anthropogenic surfaces within AOI (NLCD) |
| PctHbWet2019CatRp100 | 0.001149 | % of AOI area classified as herbaceous wetland land cover (NLCD class 95) |
| MSST_2013 | 0.001115 | Predicted mean summer stream temperature (July-Aug) for years 2008, 2009, 2013, 2014 |
| NANIWs | 0.001108 | Net Anthropogenic Nitrogen within AOI: farm fertilizer + urban fertilizer + NOx deposition + CBNF + Human and Livestock food - crop N content - livestock N content |
| Precip8110Ws | 0.001107 | PRISM climate data - 30-year normal mean precipitation (mm): Annual period: 1981-2010 within AOI |
| MAST_2009 | 0.001104 | Predicted mean annual stream temperature (Jan-Dec) for year 2008, 2009, 2013, 2014 |
| SepticWs | 0.000991 | Density of septic systems per AOI - based on 1990 Census |
| Tmean8110Cat | 0.000952 | PRISM climate data - 30-year normal mean temperature (C°): Annual period: 1981-2010 within the AOI |
| RunoffCat | 0.000932 | Mean runoff (mm) within AOI |
| PctHbWet2019Ws | 0.000893 | % of AOI area classified as herbaceous wetland land cover (NLCD class 95) |
| SandCat | 0.000878 | Mean % sand content of soils (STATSGO) within AOI |
| SuperfundDensWs | 0.000855 | Density of Superfund sites within AOI (sites/square km) |
| PctGlacTilLoamCat | 0.000825 | % of AOI area classified as lithology type: glacial till, loamy |
| PctImp2013Cat | 0.000805 | Mean imperviousness of anthropogenic surfaces within AOI (NLCD) |
| NCat | 0.000726 | Mean % of lithological nitrogen (N) content in surface or near surface geology within AOI |
| BFIWs | 0.000721 | Base flow is the component of streamflow that can be attributed to ground-water discharge into streams. The BFI is the ratio of base flow to total flow, expressed as a percentage, within AOI |
| PctImp2006WsRp100 | 0.000699 | Mean imperviousness of anthropogenic surfaces within AOI (NLCD) |

| | | |
|---|---|---|
| ManureWs | 0.000695 | Mean rate of manure application to agricultural land from confined animal feeding operations in kg N/ha/yr, within AOI |
| TRIDensWs | 0.000673 | Density of TRI (Toxic Release Inventory) sites within AOI (sites/square km) |
| PctAlluvCoastWs | 0.000662 | % of AOI area classified as lithology type: alluvium and fine-textured coastal zone sediment |
| CaOCat | 0.000651 | Mean % of lithological calcium oxide (CaO) content in surface or near surface geology within AOI |
| SandWs | 0.000650 | Mean % sand content of soils (STATSGO) within AOI |
| PctImp2019Cat | 0.000636 | Mean imperviousness of anthropogenic surfaces within AOI (NLCD) |
| PctHbWet2019WsRp100 | 0.000595 | % of AOI area classified as herbaceous wetland land cover (NLCD class 95) |
| PctColluvSedWs | 0.000576 | % of AOI area classified as lithology type: colluvial sediment |
| PctBl2019Cat | 0.000569 | % of AOI area classified as barren land cover (NLCD class 31) |
| NH4_2008Ws | 0.000547 | Annual gradient map of precipitation-weighted mean deposition for ammonium ion concentration wet deposition for 2008 in kg of NH4/ha/yr, within AOI |
| PctGrs2019Cat | 0.000543 | % of AOI area classified as grassland/herbaceous land cover (NLCD class 71) |
| PctImp2016Ws | 0.000522 | Mean imperviousness of anthropogenic surfaces within AOI (NLCD) |
| PctUrbHi2019WsRp100 | 0.000522 | % of AOI area classified as developed, high-intensity land use (NLCD class 24) |
| PctImp2004CatRp100 | 0.000521 | Mean imperviousness of anthropogenic surfaces within AOI (NLCD) |
| Tmax8110Ws | 0.000520 | PRISM climate data - 30-year normal maximum temperature (C°): Annual period: 1981-2010 within the AOI |
| PctOw2019Ws | 0.000495 | % of AOI area classified as open water land cover (NLCD class 11) |
| KffactWs | 0.000492 | Mean of STATSGO Kffactor raster within AOI. The Universal Soil Loss Equation (USLE) and represents a relative index of susceptibility of bare, cultivated soil to particle detachment and transport by rainfall |
| RdDensCat | 0.000450 | Density of roads (2010 Census Tiger Lines) within AOI (km/square km) |
| PctImp2016WsRp100 | 0.000447 | Mean imperviousness of anthropogenic surfaces within AOI (NLCD) |
| MSST_2009 | 0.000422 | Predicted mean summer stream temperature (July-Aug) for years 2008, 2009, 2013, 2014 |
| CBNFCat | 0.000417 | Mean rate of biological nitrogen fixation from the cultivation of crops in kg N/ha/yr, within AOI |
| WtDepWs | 0.000408 | Mean seasonal water table depth (cm) of soils (STATSGO) within AOI |
| PctImp2008Cat | 0.000407 | Mean imperviousness of anthropogenic surfaces within AOI (NLCD) |
| Phos_Crop_UptakeCat | 0.000385 | Phosphorus update by crops in the catchment |
| WWTPMinorDensWs | 0.000353 | Density (number/ km2) of minor wastewater treatment plants per AOI |
| RockNCat | 0.000350 | N from rock weathering (kg/ km2) within AOI |

| | | |
|---|---|---|
| PctImp2013Ws | 0.000327 | Mean imperviousness of anthropogenic surfaces within AOI (NLCD) |
| Tmean8110Ws | 0.000292 | PRISM climate data - 30-year normal mean temperature (C°): Annual period: 1981-2010 within the AOI |
| NO3_2008Cat | 0.000290 | Annual gradient map of precipitation-weighted mean deposition for nitrate ion concentration wet deposition for 2008 in kg of NO3/ha/yr, within AOI |
| PctUrbOp2019WsRp100 | 0.000271 | % of AOI area classified as developed, open space land use (NLCD class 21) |
| PctImp2008WsRp100 | 0.000255 | Mean imperviousness of anthropogenic surfaces within AOI (NLCD) |
| SWs | 0.000224 | Mean % of lithological sulfur (S) content in surface or near surface geology within AOI |
| PctShrb2019CatRp100 | 0.000215 | % of AOI area classified as shrub/scrub land cover (NLCD class 52) |
| Tmax8110Cat | 0.000193 | PRISM climate data - 30-year normal maximum temperature (C°): Annual period: 1981-2010 within the AOI |
| Phos_ManureWs | 0.000193 | Mean rate of manure application to agricultural land from confined animal feeding operations in kg N/ha/yr, within AOI |
| PctAg2006Slp10Cat | 0.000163 | Mean % of lithological sulfur (S) content in surface or near surface geology within AOI |
| Fe2O3Ws | 0.000162 | Mean % of lithological ferric oxide (Fe2O3) content in surface or near surface geology within AOI |
| WsAreaSqKm | 0.000155 | Mean % of lithological sulfur (S) content in surface or near surface geology within AOI |
| PctUrbHi2019CatRp100 | 0.000138 | % of AOI area classified as developed, high-intensity land use (NLCD class 24) |
| PctUrbOp2019Cat | 0.000138 | % of AOI area classified as developed, open space land use (NLCD class 21) |
| NPDESDensWsRp100 | 0.000119 | Density of permitted NPDES (National Pollutant Discharge Elimination System) sites within AOI (sites/square km) |
| DamNrmStorWs | 0.000118 | Volume all reservoirs (NORM_STORA in NID) per unit area of AOI (cubic meters/square km) |
| SiO2Ws | 0.000107 | Mean % of lithological silicon dioxide (SiO2) content in surface or near surface geology within AOI |
| SuperfundDensWsRp100 | 0.000097 | Density of Superfund sites within AOI (sites/square km) |
| SCat | 0.000096 | Mean % of lithological sulfur (S) content in surface or near surface geology within AOI |
| CBNFWs | 0.000092 | Mean rate of biological nitrogen fixation from the cultivation of crops in kg N/ha/yr, within AOI |
| PctAlluvCoastCat | 0.000091 | % of AOI area classified as lithology type: alluvium and fine-textured coastal zone sediment |
| PctImp2001WsRp100 | 0.000074 | Mean imperviousness of anthropogenic surfaces within AOI (NLCD) |
| PctGlacLakeCrsCat | 0.000069 | % of AOI area classified as lithology type: glacial outwash and glacial lake sediment, coarse-textured |
| PctImp2001Ws | 0.000058 | Mean imperviousness of anthropogenic surfaces within AOI (NLCD) |
| PctWdWet2019CatRp100 | 0.000049 | % of AOI area classified as woody wetland land cover (NLCD class 90) |

| | | |
|---|---|---|
| PctColluvSedCat | 0.000046 | % of AOI area classified as lithology type: colluvial sediment |
| PctHydricWs | 0.000042 | % of AOI area classified as lithology type: hydric, peat and muck |
| PctConif2019CatRp100 | 0.000038 | % of AOI area classified as evergreen forest land cover (NLCD class 42) |
| PctGlacLakeCrsWs | 0.000014 | % of AOI area classified as lithology type: glacial outwash and glacial lake sediment, coarse-textured |
| PctGlacTilClayCat | 0.000000 | % of AOI area classified as lithology type: glacial till, clayey |
| DamNIDStorCat | 0.000000 | Volume all reservoirs (NID_STORA in NID) per unit area of AOI (cubic meters/square km) |
| MineDensCat | 0.000000 | Density of mines sites within AOI (mines/square km) |
| NABD_NrmStorCat | 0.000000 | Volume all reservoirs (NORM_STORA in NID) per unit area of AOI (cubic meters/square km) |
| PctGlacTilCrsCat | 0.000000 | % of AOI area classified as lithology type: glacial till, coarse-textured |
| PctGlacTilCrsWs | 0.000000 | % of AOI area classified as lithology type: glacial till, coarse-textured |
| PctHydricCat | 0.000000 | % of AOI area classified as lithology type: hydric, peat and muck |
| SuperfundDensCat | 0.000000 | Density of Superfund sites within AOI (sites/square km) |
| TRIDensCat | 0.000000 | Density of TRI (Toxic Release Inventory) sites within AOI (sites/square km) |
| WWTPMajorDensCat | 0.000000 | Density (number/ km2) of major wastewater treatment plants per AOI |
| DamDensCat | -0.000001 | Density of georeferenced dams within AOI (dams/ square km) |
| ElevCat | -0.000003 | Mean AOI elevation (m) |
| RckDepCat | -0.000010 | Mean depth (cm) to bedrock of soils (STATSGO) within AOI. |
| NABD_DensCat | -0.000011 | Density of georeferenced dams within AOI (dams/ square km) |
| DamNrmStorCat | -0.000043 | Volume all reservoirs (NORM_STORA in NID) per unit area of AOI (cubic meters/square km) |
| PctWaterWs | -0.000051 | % of AOI area classified as lithology type: water |
| NPDESDensCat | -0.000064 | Density of permitted NPDES (National Pollutant Discharge Elimination System) sites within AOI (sites/square km) |
| PctImp2011WsRp100 | -0.000079 | Mean imperviousness of anthropogenic surfaces within AOI (NLCD) |
| PctUrbLo2019Ws | -0.000088 | % of AOI area classified as developed, low-intensity land use (NLCD class 22) |
| ClayCat | -0.000090 | Mean % clay content of soils (STATSGO) within AOI |
| PctHay2019WsRp100 | -0.000098 | % of AOI area classified as hay land use (NLCD class 81) |
| CatAreaSqKm | -0.000113 | Mean % of lithological sulfur (S) content in surface or near surface geology within AOI |
| NABD_NIDStorCat | -0.000113 | Volume all reservoirs (NID_STORA in NID) per unit area of AOI (cubic meters/square km) |
| PctUrbLo2019CatRp100 | -0.000115 | % of AOI area classified as developed, low-intensity land use (NLCD class 22) |
| PctAg2006Slp20Ws | -0.000120 | Mean % of lithological sulfur (S) content in surface or near surface geology within AOI |
| WWTPMajorDensWs | -0.000130 | Density (number/ km2) of major wastewater treatment plants per AOI |

| | | |
|---|---|---|
| P2O5Cat | -0.000134 | Mean % of lithological phosphorous oxide (P2O5) content in surface or near surface geology within AOI |
| PctShrb2019WsRp100 | -0.000136 | % of AOI area classified as shrub/scrub land cover (NLCD class 52) |
| PctOw2019WsRp100 | -0.000158 | % of AOI area classified as open water land cover (NLCD class 11) |
| RdCrsSlpWtdWs | -0.000164 | Density of roads-stream intersections (2010 Census Tiger Lines-NHD stream lines) multiplied by NHDPlusV21 slope within AOI (crossings*slope/square km) |
| SepticCat | -0.000164 | Density of septic systems per AOI - based on 1990 Census |
| PctBl2019CatRp100 | -0.000189 | % of AOI area classified as barren land cover (NLCD class 31) |
| PctHay2019CatRp100 | -0.000195 | % of AOI area classified as hay land use (NLCD class 81) |
| PctImp2006CatRp100 | -0.000210 | Mean imperviousness of anthropogenic surfaces within AOI (NLCD) |
| CompStrgthWs | -0.000219 | Mean lithological uniaxial compressive strength (megaPascals) content in surface or near surface geology within AOI |
| PctUrbMd2019Ws | -0.000224 | % of AOI area classified as developed, medium-intensity land use (NLCD class 23) |
| MgOWs | -0.000235 | Mean % of lithological magnesium oxide (MgO) content in surface or near surface geology within AOI |
| PctConif2019Ws | -0.000236 | % of AOI area classified as evergreen forest land cover (NLCD class 42) |
| SN_2008Cat | -0.000237 | Annual gradient map of precipitation-weighted mean deposition for average sulfur & nitrogen wet deposition for 2008 in kg of S+N/ha/yr, within AOI |
| PctAg2006Slp20Cat | -0.000257 | Mean % of lithological sulfur (S) content in surface or near surface geology within AOI |
| PctUrbLo2019WsRp100 | -0.000281 | % of AOI area classified as developed, low-intensity land use (NLCD class 22) |
| PctGlacLakeFineCat | -0.000284 | % of AOI area classified as lithology type: glacial lake sediment, fine-textured |
| RdDensCatRp100 | -0.000293 | Density of roads (2010 Census Tiger Lines) within AOI (km/square km) |
| PctImp2013WsRp100 | -0.000297 | Mean imperviousness of anthropogenic surfaces within AOI (NLCD) |
| TRIDensWsRp100 | -0.000311 | Density of TRI (Toxic Release Inventory) sites within AOI (sites/square km) |
| MineDensWs | -0.000323 | Density of mines sites within AOI (mines/square km) |
| PctImp2019CatRp100 | -0.000369 | Mean imperviousness of anthropogenic surfaces within AOI (NLCD) |
| CaOWs | -0.000370 | Mean % of lithological calcium oxide (CaO) content in surface or near surface geology within AOI |
| WWTPMinorDensCat | -0.000371 | Density (number/ km2) of minor wastewater treatment plants per AOI |
| Al2O3Ws | -0.000384 | Mean % of lithological aluminum oxide (Al2O3) content in surface or near surface geology within AOI |
| NANICat | -0.000387 | Net Anthropogenic Nitrogen within AOI: farm fertilizer + urban fertilizer + NOx deposition + CBNF + Human and Livestock food - crop N content - livestock N content |
| PctImp2011Cat | -0.000388 | Mean imperviousness of anthropogenic surfaces within AOI (NLCD) |

| | | |
|---|---|---|
| PctUrbHi2019Ws | -0.000391 | % of AOI area classified as developed, high-intensity land use (NLCD class 24) |
| PctGrs2019WsRp100 | -0.000395 | % of AOI area classified as grassland/herbaceous land cover (NLCD class 71) |
| PctShrb2019Cat | -0.000403 | % of AOI area classified as shrub/scrub land cover (NLCD class 52) |
| NH4_2008Cat | -0.000420 | Annual gradient map of precipitation-weighted mean deposition for ammonium ion concentration wet deposition for 2008 in kg of NH4/ha/yr, within AOI |
| PctGlacTilClayWs | -0.000429 | % of AOI area classified as lithology type: glacial till, clayey |
| PctImp2001Cat | -0.000448 | Mean imperviousness of anthropogenic surfaces within AOI (NLCD) |
| RckDepWs | -0.000461 | Mean depth (cm) to bedrock of soils (STATSGO) within AOI. |
| HydrlCondWs | -0.000479 | Mean lithological hydraulic conductivity (micrometers per second) content in surface or near surface geology within AOI |
| Na2OWs | -0.000487 | Mean % of lithological sodium oxide (Na2O) content in surface or near surface geology within AOI |
| NABD_NrmStorWs | -0.000488 | Volume all reservoirs (NORM_STORA in NID) per unit area of AOI (cubic meters/square km) |
| PctImp2004Ws | -0.000492 | Mean imperviousness of anthropogenic surfaces within AOI (NLCD) |
| WWTPAllDensCat | -0.000542 | Density (number/ km2) of all wastewater treatment plants per AOI |
| PermWs | -0.000550 | Mean permeability (cm/hour) of soils (STATSGO) within AOI |
| PctMxFst2019CatRp100 | -0.000556 | % of AOI area classified as mixed deciduous/evergreen forest land cover (NLCD class 43) |
| BFICat | -0.000560 | Base flow is the component of streamflow that can be attributed to ground-water discharge into streams. The BFI is the ratio of base flow to total flow, expressed as a percentage, within AOI |
| PctImp2019WsRp100 | -0.000576 | Mean imperviousness of anthropogenic surfaces within AOI (NLCD) |
| NsurpWs | -0.000592 | Nitrogen Surplus (kg N / yr) per AOI - excluding biological N Fixation |
| MineDensWsRp100 | -0.000632 | Density of mines sites within AOI (mines/square km) |
| P2O5Ws | -0.000647 | Mean % of lithological phosphorous oxide (P2O5) content in surface or near surface geology within AOI |
| PctDecid2019WsRp100 | -0.000675 | % of AOI area classified as deciduous forest land cover (NLCD class 41) |
| PctImp2004Cat | -0.000691 | Mean imperviousness of anthropogenic surfaces within AOI (NLCD) |
| PctImp2011Ws | -0.000768 | Mean imperviousness of anthropogenic surfaces within AOI (NLCD) |
| PctConif2019Cat | -0.000774 | % of AOI area classified as evergreen forest land cover (NLCD class 42) |
| PctImp2011CatRp100 | -0.000784 | Mean imperviousness of anthropogenic surfaces within AOI (NLCD) |
| NWs | -0.000812 | Mean % of lithological nitrogen (N) content in surface or near surface geology within AOI |
| PctGlacTilLoamWs | -0.000859 | % of AOI area classified as lithology type: glacial till, loamy |

| | | |
|---|---|---|
| PctImp2008CatRp100 | -0.000863 | Mean imperviousness of anthropogenic surfaces within AOI (NLCD) |
| PctBl2019Ws | -0.000940 | % of AOI area classified as barren land cover (NLCD class 31) |
| InorgNWetDep_2008Ws | -0.000977 | Annual gradient map of precipitation-weighted mean deposition for inorganic nitrogen wet deposition from nitrate and ammonium for 2008 in kg of N/ha/yr, within AOI |
| Tile_density | -0.000989 | Tile density in the watershed in m2/km2 |
| PctMxFst2019Cat | -0.000993 | % of AOI area classified as mixed deciduous/evergreen forest land cover (NLCD class 43) |
| PctImp2004WsRp100 | -0.000997 | Mean imperviousness of anthropogenic surfaces within AOI (NLCD) |
| Tmin8110Ws | -0.001004 | PRISM climate data - 30-year normal minimum temperature (C°): Annual period: 1981-2010 within the AOI |
| MAST_2008 | -0.001057 | Predicted mean annual stream temperature (Jan-Dec) for year 2008, 2009, 2013, 2014 |
| DamNIDStorWs | -0.001058 | Volume all reservoirs (NID_STORA in NID) per unit area of AOI (cubic meters/square km) |
| Phos_FertWs | -0.001065 | Mean rate of synthetic nitrogen fertilizer application to agricultural land in kg N/ha/yr, within the AOI |
| PctConif2019WsRp100 | -0.001095 | % of AOI area classified as evergreen forest land cover (NLCD class 42) |
| PctImp2001CatRp100 | -0.001121 | Mean imperviousness of anthropogenic surfaces within AOI (NLCD) |
| RdDensWsRp100 | -0.001130 | Density of roads (2010 Census Tiger Lines) within AOI (km/square km) |
| PctUrbOp2019Ws | -0.001147 | % of AOI area classified as developed, open space land use (NLCD class 21) |
| WtDepCat | -0.001155 | Mean seasonal water table depth (cm) of soils (STATSGO) within AOI |
| Tmin8110Cat | -0.001190 | PRISM climate data - 30-year normal minimum temperature (C°): Annual period: 1981-2010 within the AOI |
| RunoffWs | -0.001264 | Mean runoff (mm) within AOI |
| PctDecid2019Ws | -0.001384 | % of AOI area classified as deciduous forest land cover (NLCD class 41) |
| PctWdWet2019Cat | -0.001398 | % of AOI area classified as woody wetland land cover (NLCD class 90) |
| PctImp2016CatRp100 | -0.001416 | Mean imperviousness of anthropogenic surfaces within AOI (NLCD) |
| SN_2008Ws | -0.001443 | Annual gradient map of precipitation-weighted mean deposition for average sulfur & nitrogen wet deposition for 2008 in kg of S+N/ha/yr, within AOI |
| RdCrsCat | -0.001469 | Density of roads-stream intersections (2010 Census Tiger Lines-NHD stream lines) within AOI (crossings/square km) |
| PctOw2019CatRp100 | -0.001521 | % of AOI area classified as open water land cover (NLCD class 11) |
| Precip8110Cat | -0.001556 | PRISM climate data - 30-year normal mean precipitation (mm): Annual period: 1981-2010 within AOI |
| PctImp2006Ws | -0.001586 | Mean imperviousness of anthropogenic surfaces within AOI (NLCD) |
| ManureCat | -0.001594 | Mean rate of manure application to agricultural land from confined animal feeding operations in kg N/ha/yr, within AOI |

| | | |
|---|---|---|
| PctUrbHi2019Cat | -0.001652 | % of AOI area classified as developed, high-intensity land use (NLCD class 24) |
| NABD_DensWs | -0.001661 | Density of georeferenced dams within AOI (dams/ square km) |
| PctImp2019Ws | -0.001693 | Mean imperviousness of anthropogenic surfaces within AOI (NLCD) |
| PctUrbLo2019Cat | -0.001734 | % of AOI area classified as developed, low-intensity land use (NLCD class 22) |
| Precip_Minus_EVTCat | -0.001799 | This dataset represents surplus precipitation (mm): precipitation minus potential evaporation described in DOI: 10.1016/j.scitotenv.2020.137661 within individual, local NHDPlusV2 catchments and upstream, contributing watersheds. |
| OmWs | -0.001820 | Mean organic matter content (% by weight) of soils (STATSGO) within AOI |
| PctAg2006Slp10Ws | -0.001835 | Mean % of lithological sulfur (S) content in surface or near surface geology within AOI |
| K2OWs | -0.001836 | Mean % of lithological potassium oxide (K2O) content in surface or near surface geology within AOI |
| PctUrbMd2019Cat | -0.001852 | % of AOI area classified as developed, medium-intensity land use (NLCD class 23) |
| PctImp2006Cat | -0.001861 | Mean imperviousness of anthropogenic surfaces within AOI (NLCD) |
| PctImp2016Cat | -0.001956 | Mean imperviousness of anthropogenic surfaces within AOI (NLCD) |
| CanalDensWs | -0.001980 | Density of NHDPlus line features classified as canal, ditch, or pipeline within the AOI (km/ square km) |
| InorgNWetDep_2008Cat | -0.002086 | Annual gradient map of precipitation-weighted mean deposition for inorganic nitrogen wet deposition from nitrate and ammonium for 2008 in kg of N/ha/yr, within AOI |
| PctUrbMd2019WsRp100 | -0.002287 | % of AOI area classified as developed, medium-intensity land use (NLCD class 23) |
| PctGrs2019CatRp100 | -0.002320 | % of AOI area classified as grassland/herbaceous land cover (NLCD class 71) |
| DamDensWs | -0.002323 | Density of georeferenced dams within AOI (dams/ square km) |
| PctBl2019WsRp100 | -0.002483 | % of AOI area classified as barren land cover (NLCD class 31) |
| CanalDensCat | -0.002813 | Density of NHDPlus line features classified as canal, ditch, or pipeline within the AOI (km/ square km) |
| MSST_2008 | -0.003001 | Predicted mean summer stream temperature (July-Aug) for years 2008, 2009, 2013, 2014 |
| PctHay2019Ws | -0.003454 | % of AOI area classified as hay land use (NLCD class 81) |