# Peer review of "Phosphorus transport in a hotter and drier climate: in-channel release of legacy phosphorus during summer low flow conditions"

_EGUsphere, 2024_

## Author Response (AR1)

**RC 1:**

In this work, Christine L et al. has evaluated the potential contribution of in-channel release of legacy P to bioavailable P transport in streams during summer low flow conditions across a land use gradient in Minnesota, USA. They found that elevated riverine concentrations of SRP during low flow conditions in late summer altered C-Q transport behavior for more than half (54%) of the gaged watersheds we studied, weakening what was otherwise more strongly mobilizing behavior during higher flow conditions and other times of year. These watersheds occurred almost exclusively in landscapes that were heavily modified by agricultural or urban land use. In general, the outcome of this work can be interesting for the scientific community. However, there are still some issues that need to be addressed.

**Author response:** We thank the reviewer for these comments. Our responses to the specific concerns identified by RC1 appear below:

1. In abstract: This section is too long, please just briefly state the significance of your research and the results of your research, the hypothesis can be placed in the introduction or other parts of the section.

**Author response:** In the revised manuscript, we have streamlined the abstract to reduce length.

2. Supplementary model uncertainty or error analysis is necessary.

**Author response:** We have provided additional text in results section 3.8 to which discusses model uncertainty and prediction error.

3. Conclusions: Similar issues to the abstract section, the content is too long, please revise it. Also, the main findings of this study should be stated followed by a description of the next steps. Here, the order seems to be reversed

**Author response:** In the revised manuscript we have streamlined and reorganized the conclusion, to more strongly state the main findings with a description of next steps.

4. Figure 7 is not very aesthetically pleasing, especially the colour scheme, and would like to improve it further

**Author response:** We have revised the color scheme for Figure 7, as suggested.

5. It is necessary to sum up the shortcomings of this study, which is important for further work in this area.

**Author response:** We have added a 'Limitations' section that summarizes the shortcomings of the study.

6. The discussion section simply summarises the findings of the study and then discusses them; there is no need to repeat the findings at great length. There are too many similar repetitions throughout the text, which takes up a lot of space, so please fix it!

**Author response:** In the revised draft,we have streamlined the discussion and sought to eliminate repeat mentions of study results.

7. The results of the study can be used to propose initial responses in the relevant regions.

**Author response:** Thank you for this comment. We have added some text to the Conclusion suggesting initial conservation responses in our study region, including the management of riparian buffer conditions.

**RC2:**

General comments:

In the manuscript entitled "Phosphorus transport in a hotter and drier climate: in-channel release of legacy phosphorus during summer low flow conditions," the authors assessed the potential for streambed sediment-bound phosphorus to be released to streams and rivers during summer low flow conditions. The authors used water quality data from across numerous sites, seasons, and years to first determine if dissolved phosphorus concentrations and patterns differed between seasons and then to model the drivers of the patterns observed in the phosphorus concentrations during late summer. This manuscript provides important information about what drives instream bioavailable phosphorus concentrations during late summer when algal blooms are most impactful on riverine and downstream ecosystems. While not measured directly in this study, the results imply that legacy phosphorus sources in the stream channel are negatively impacting water quality in the late summer. Overall, I think this manuscript is an important contribution to our understanding of potential factors influencing instream legacy phosphorus release.

**Author response:** We so appreciate the reviewer's detailed reading of our manuscript, and we thank them for their comments. Our responses to the specific concerns identified by RC2 appear below:

Specific comments

**Comment:** Line 86: Is summer of one of the 'times of year' when in-channel processes drive bioavailable phosphorus release? Please provide more details about this general statement.

**Author response:** Yes, summer is a time when in channel processes likely drive bioavailable P release; we have clarified this in the revised manuscript.

**Comment:** Lines 87-94: What are the important, main conclusions of these studies that you cite in this paragraph? Are any of the results from these studies relevant to the findings from your random forest model? Did you use any of these studies as a basis for the input variables for your random forest model?

**Author response:** Some of these findings we mention more specifically in the immediately preceding paragraph. We also address them in relation to the findings from our random forest model in the Discussion section.

**Comment:** Lines 175-176: Please add watershed areas to Table A3. This information would help the reader with the data interpretation.

**Author response:** We agree that this is a good idea, and have added watershed areas to Table A3 in the revised draft.

**Comment:** Line 184: What farm practice is 'swine finishing'?

**Author response:** Swine finishing is the final stage of pig farming where young pigs are fed until they reach market weight. We added text to the revised draft to clarify this.

**Comment:** Lines 199-200: What criteria were used to categorize the seasons? Was it air temperature? The breakdown of seasons is winter is five months long; spring is two months long; summer is four months long; and fall is only one month long.

**Author response:** We categorized seasons in approximate relation to the agricultural growing seasons in our study region, with spring corresponding to when crops are planted, summer corresponding to when crops are growing rapidly, fall corresponding to when dominant crops (corn, soybeans) are harvested, and winter corresponding to when crops are dormant. We divided winter into early winter when snow is accumulating and generally not melting, and late winter, which is associated with snowmelt. We divided summer into early summer when conditions are generally wetter and crops are experiencing rapid growth, and late summer when climate conditions are generally drier and warm season crops mature rapidly. We clarified these definitions in the revised text.

**Comment:** Lines 285-303: How many predictor variables did you start with? Did you use all 600 environmental metrics from StreamCat? What were the criteria for entering predictor variables in the model?

**Author response:** We used 253 predictor variables in the model, after excluding variables that did not provide useful information (i.e., were all 0s or had too many missing values), that didn't

match the timing of our dataset (i.e., land cover data from years outside of the phosphorus data we used), or weren't especially relevant (e.g., variables describing forest fire intensity). We added text in the revised manuscript to specify the number of input predictor variables we used in the model, and to clarify why we selected those attributes out of the total available.

**Comment:** Line 386: According to Table A2, early winter is when the R2 is 0.86 and late winter has a R2 of 0.44. Also, early winter only has five sites which may have inflated the R2 relative to the other time periods.

**Author response:** Thank you for catching this; we have revised the manuscript to link early winter to the correct R2. We have also added text to qualify the result given the small sample size.

**Comment:** Line 408: MC1 had relatively low SRP concentrations in early summer. Do you mean to reference WR1 instead?

**Author response:** Yes, thank you again for your attention to detail! We have revised the manuscript to correctly refer to WR1 as having high SRP in early summer.

**Comment:** Lines 426-427: According to Figure 5, it looks like only three or four higher WWTP influence sites had higher SRP concentrations.

**Author response:** We have corrected this to 3 sites with higher WWTP influence with higher SRP in late winter. We also rechecked all other results for all other seasons for this paragraph and they are correct.

**Comment:** Lines 582-584: Please provide a citation for the climate models that indicate that the Upper Midwest summers will be drier and hotter.

**Author response:** We have added the following citation, which notes likely increasing temperatures, decreasing summer precipitation and drier soils and increased frequency of droughts for the Midwest:

Wilson, A.B., J.M. Baker, E.A. Ainsworth, J. Andresen, J.A. Austin, J.S. Dukes, E. Gibbons, B.O. Hoppe, O.E. LeDee, J. Noel, H.A. Roop, S.A. Smith, D.P. Todey, R. Wolf, and J.D. Wood, 2023: Ch. 24. Midwest. In: Fifth National Climate Assessment. Crimmins, A.R., C.W. Avery, D.R. Easterling, K.E. Kunkel, B.C. Stewart, and T.K. Maycock, Eds. U.S. Global Change Research Program, Washington, DC, USA. https://doi.org/10.7930/NCA5.2023.CH24

**Comment:** Line 629: Are you referring to mean winter stream temperatures or mean annual stream temperatures?

**Author response:** We are referring to mean annual stream temperatures, and have corrected this in the revised text.

**Comment:** Lines 708-709: Through what processes does the deposition of fine clay sediment affect organic matter processing?

**Author response:** Fine clay sediments in streams can slow down organic matter processing by limiting light availability (thereby reducing photosynthesis and the supply of organic matter), smothering benthic habitats and reducing microbial activity in benthic environments, and altering chemical interactions. However, in responding to RC1 comments to streamline the manuscript, we have removed this text from the revised manuscript, as it provides additional information not necessary to interpret the findings.

**Comment:** Table 1: What was the detection limit for SRP? The maximum Spring SRP for the more impacted sites has two decimal points.

**Author response:** The detection limit was 0.001. We have specified this detection limit in the revised methods section. We have also edited the values in Table 1 to 3 significant digits to reflect this. We have also addressed the typo of two decimal points in Table 1.

**Comment:** Table 2: Lines 139, 179, and 401 all state that 10 farmed fields were sampled, yet 11 fields are listed in Table 2. Should the lines state 11 fields?

**Author response:** One farm (NOW1) had two tile outlets that drained two different areas of the same farm field. We have added text to the legend of Table 2 to clarify why there are 2 tile outlets listed for this farm.

**Comment:** Table A6: Please define the predictor variables like you did in Figure 9. Importance is misspelled in the table.

**Author response:** We have defined the predictor variables in a revised table as requested. The typo has also been fixed.

**Comment:** Figure 1: The map only displays eight farms. Is there more than one field per farm?

**Author response:** 1 farm location was hidden behind field site locations; we have moved it to the top layer so it is more visible. Two of the farm fields (known as RW1N and RW1S) are located in close proximity (but are not the same site), and so appear as one location on the map. We have noted this in the Figure legend.

**Comment:** Figures 2, 3, and 4: Recommend displaying the R2 values for each of the plots.

**Author response:** We have displayed R2 values for each of the panels in these 3 plots.

**Comment:** Figures 5 and A2: What order are the tile outlets? Is it the same order as in Figure 6? Also, recommend putting the number of gaged watersheds for each season as they are different.

**Author response:** Yes, the order is the same as Figure 6. We have added tile ID labels to the bottom of the Figure. We have also added the number of gages in each season as a label in the plot for each.

**Comment:** Figure 6: How were the 33 sites distributed across ditches, intermittent streams, and perennial streams?

**Author response:** We have added this information (# of sites in each category) to the Figure 6 legend.

**Comment:** Figure 8: The color dot for early summer is not on the figure.

**Author response:** The dot for early summer is on the figure, but it was very light in color and difficult to see. We have changed the color for this season to be more readily visible.

**Comment:** Figure 10 and Table 3: Great explanations and visualization of the effects of the predictor variables.

**Author response:** Thank you!

**Comment:** Figure A1: What does the dashed vertical line indicate?

**Author response:** The dashed line indicates the mean of all mean SRP values. We have defined this in the Figure legend.

Technical corrections

Line 48: Delete 'used'

**Author response:** deleted

Line 61: Should Keiser and Shapiro be 2018 or 2019?

**Author response:** 2019, we have corrected citations in the text.

Lines 80 and 82: Should King et al. by 2014 or 2015?

**Author response:** 2015, we have corrected citations in the text.

Line 248: Warrick et al. 2015 was not in the references section.

**Author response:** We have added this citation to the references section.

Line 319: Kuhn et al. 2000 was not in the references section.

**Author response:** We have added this citation to the references section.

Line 374: Table A1 lists that R2 for late summer as 0.06.

**Author response:** We have corrected this typo in the text.

Line 403: According to Table 2, mean SRP in winter is 0.131.

**Author response:** We have correct this to match the table.

Lines 590-591: Should Casquin et al. be 2020 or 2021?

**Author response**: 2020, we have corrected this in the text.

Line 594: Should Meybeck and Moatar be 2011 or 2012?

**Author response:** 2021, we have corrected this in the text

Lines 602, 632, 685, 739, and throughout manuscript: Adverbs that end in 'ly' like 'anthropogenically' are not hyphenated.

**Author response:** We have corrected these uses in the text.

Line 606: Delete 'he'.

**Author response:** deleted

Line 733: Should be 'in-channel'.

**Author response:** the text reads "in-channel"